

# Satellite-Derived Light Extinction Coefficient and its Impact on Thermal Structure Simulations in a 1-D Lake Model

Kiana Zolfaghari[1], Claude R. Duguay[1], Homa Kheyrollah Pour[1]

[1]Interdisciplinary Centre on Climate Change and Department of Geography & Environmental Management, University of
Waterloo, Waterloo, Canada

*Correspondence to*: Kiana Zolfaghari (kzolfagh@uwaterloo.ca)

**Abstract.** One essential optical parameter to specify in lake models is water clarity, which is parameterized based on the light extinction coefficient ($K_d$). A global constant value of $K_d$ is usually specified in lake models. One-dimensional (1-D) lake models are most often used as lake parameterization schemes in numerical weather prediction and regional climate models.
This study aimed to improve the performance of the 1-D Freshwater Lake (FLake) model using satellite-derived $K_d$ for Lake Erie. The CoastColour algorithm is applied to MERIS satellite imagery to estimate $K_d$ and evaluated against $K_d$ derived from Secchi disk depth (SDD) field-based measurements collected during Lake Erie cruises. A good agreement is found between field and satellite-derived $K_d$ (RMSE = 0.63 m$^{-1}$, MBE = -0.09 m$^{-1}$, I_a = 0.65) (in situ data was collected in 2004, 2005, 2008, 2011, 2012). The constant (0.2 m$^{-1}$) and satellite-derived $K_d$ values as well as radiation fluxes and meteorological station
observations are then used to run FLake at the location of a buoy where lake surface water temperature (LSWT) was measured in 2008. Results improved compared to using a constant $K_d$ value (0.2 m$^{-1}$) (lake-specific yearly average $K_d$ value: RMSE=1.54 °C, MBE= -0.08 °C; constant $K_d$ value: RMSE=1.76 °C, MBE= -1.26 °C). No significant improvement is found in FLake simulated LSWT when $K_d$ variations in time are considered using a monthly average. Therefore, results suggest that a time-independent, lake-specific, and constant satellite-derived $K_d$ value can reproduce LSWT with sufficient accuracy.
A sensitivity analysis is also performed to assess the impact of various $K_d$ values on the simulation of mean water column temperature (MWCT), mixed layer depth (MLD), water temperature isotherms as well as ice dates and thickness. Results show that FLake is sensitive to variations in $K_d$ to estimate the thermal structure of Lake Erie. Dark waters result in warmer spring and colder fall temperatures compare to clear waters. Dark waters always produce warmer MWCT, shallower MLD, longer ice cover duration, and thicker ice. The sensitivity of FLake to $K_d$ variations is more pronounced in the simulation of MWCT
and MLD. The model is particularly sensitive to $K_d$ values below 0.5 m$^{-1}$. This is the first study to assess the value of integrating $K_d$ from the satellite-based CoastColour algorithm into the FLake model. Satellite-derived $K_d$ is found to be a useful input parameter for simulations with FLake and possibly other lake models, and with potential for applicability to other lakes where $K_d$ is not commonly measured.

Keywords: Water clarity, extinction coefficient, MERIS, CoastColour, FLake, Lake Erie, lake water temperature



## 1 Introduction

There has been significant progress made in recent years in the representation of lakes in regional climate models (RCM) and numerical weather prediction (NWP) models. Lakes are known to be an important land surface component affecting weather and climate, especially in lake-rich regions of the northern hemisphere (Eerola et al., 2010; Martynov et al., 2012; Samuelsson

et al., 2010). They can influence the atmospheric boundary layer by modifying the air temperature, wind and precipitation. Therefore, consideration of lakes in NWP/RCM is essential (Kheyrollah Pour et al., 2012, 2014b; Martynov et al., 2010). In order to account for lakes in NWP/RCM, a description of energy exchanges between lakes and the atmosphere is required (Eerola et al., 2010). Lake Surface Water Temperature (LSWT) is one of the key variables when investigating lake-atmosphere energy exchanges (Kheyrollah Pour et al., 2012). There are various approaches to obtaining LSWT and integrating it in NWP

models, such as through climatic observations, assimilation and/or lake parameterization schemes (Eerola et al., 2010; Kheyrollah Pour et al., 2014a). Currently, LSWT is broadly modelled in NWP models using one-dimensional (1-D) lake models as lake parameterization schemes (Martynov et al., 2012). For instance, the 1-D Freshwater Lake (FLake) model performs adequately for various lake sizes, shallow to relatively deep (artificially limited to 40-60 m depth (Kourzeneva et al., 2012a)), located in both temperate and warm climate regions (Kourzeneva, 2010; Martynov et al., 2010, 2012; Mironov et al.,

2010, 2012; Samuelsson et al., 2010; Kourzeneva et al., 2012a; Kourzeneva et al., 2012b).

One of the optical parameters required as input in the FLake model is water clarity. This variable is considered as an apparent optical property and is parameterized using the light extinction coefficient ($K_d$) to describe the absorption of shortwave radiation within the water body as a function of depth (Heiskanen et al., 2015). A global constant value of $K_d$ is usually used to run lake models, including FLake. For example, Martynov et al. (2012) coupled FLake in the Canadian Regional Climate

Model (CRCM) by specifying a $K_d$ value equal to 0.2 m$^{-1}$ (Martynov, pers. comm., 2015) for all North American Lakes, including Lake Erie for years 2005-2007. Heiskanen et al. (2015) evaluated the sensitivity of two 1-D lake models, LAKE and FLake, to seasonal variations and the general level of $K_d$ for simulating water temperature profiles and turbulent fluxes of heat and momentum in a small boreal Finnish lake. Modelled values were compared to those measured for the lake during the ice-free period of 2013. The study found a critical threshold for $K_d$ (0.5 m$^{-1}$) in 1-D lake models. Heiskanen et al. (2015) concluded

that for too clear waters ($K_d < 0.5$ m$^{-1}$), the model is much more sensitive to $K_d$. The study recommends a global mapping of $K_d$ to run the FLake model for regions with clear waters ($K_d < 0.5$ m$^{-1}$) for future use in NWP models. The authors also suggest that this global mapping can be time-independent (i.e. with a constant value per lake) (Heiskanen et al., 2015), and this can be derived from satellite imagery. Potes et al. (2012) used empirically derived water clarity from space-borne Medium Resolution Imaging Spectrometer (MERIS) measurements to test the sensitivity of FLake to this parameter. The sensitivity analysis was

conducted using two $K_d$ values, representing the expected extreme water clarity cases for their study (1.0 m$^{-1}$ for clear water and 6.1 m$^{-1}$ for turbid water). The importance of lake optical properties was evaluated based on the evolution of LSWT and heat fluxes. Their results show that water clarity is an essential parameter affecting the simulated LSWT. The daily mean LSWT range increased from 1.2 ℃ in clear water to 2.4 ℃ in turbid water (Potes et al., 2012). Water clarity measurements





are included in water quality monitoring programs; however, global measurements of clarity are not yet available. Satellite remote sensing can provide water clarity observations to the modelling communities at higher spatial and temporal resolutions, to fill the gap of field measurements.

In recent years, a number of algorithms have been devised to retrieve different water optical parameters, including water clarity, from satellite observations for coastal (ocean) and lake waters (Attila et al., 2013; Binding et al., 2007, 2015; Olmanson et al., 2013; Potes et al., 2012; Wu et al., 2009; Zhao et al., 2011). Turbid inland and coastal waters are optically more complex compared to open ocean, and large optical gradients exist. There is more than only one component (phytoplankton species, various dissolved and suspended matters with non-covarying concentrations) in coastal waters and lakes that determines the variability of water-leaving reflectance. Considering this complexity, the development of algorithms for coastal waters and lakes is more challenging. MERIS, which operated from March 2002 to April 2012, collected data from the European Space Agency's (ESA) Envisat satellite. The spatial resolution and spectral bands settings were carefully selected in order to meet the primary objectives of the mission; addressing coastal monitoring from space. The best possible signal-to-noise ratio, additional channels to measure optical signatures as well as the relatively high spatial resolution of 300 m are some of the specific instrument characteristics (Ruescas et al., 2014). In 2010, ESA launched the CoastColour project to fully exploit the potential of MERIS instrument for remote sensing of coastal zone waters. CoastColour (CC) is providing a global dataset of MERIS full resolution data of coastal zones that are processed with the best possible regional algorithms to produce water-leaving reflectance and optical properties (Ruescas et al., 2014).

The objectives of this study are to: 1) evaluate satellite-derived $K_d$ values for a large lake in the Great Lakes region; 2) apply the evaluated satellite-derived $K_d$ in FLake model to investigate the improvement of model performance to reproduce LSWTs. Three different values of $K_d$ are used in the simulations: yearly average, monthly average, and a constant value to demonstrate the impact of a time-independent, lake-specific $K_d$ value in simulating LSWT; and 3) understand the sensitivity of the FLake model to $K_d$ values based on simulated LSWT, mean water column temperature (MWCT), mixed layer depth (MLD), and water temperature isotherms during the ice-free season on Lake Erie (from April to November). The impact of $K_d$ variations on ice dates (freeze-up, break-up, and duration) and ice thickness is also evaluated.

## 2 Data and Methods

### 2.1 Study Site and Station Observations

Lake Erie (42° 11′N, 81° 15′W; Fig. 1) is a large shallow temperate freshwater lake covering a surface area of 25,700 km². The lake is characterized by three basins: shallow western, central, and deep eastern basins with maximum depths of 19 m, 25 m, and 64 m, respectively. Lake Erie is monomictic with occasional dimictic years (Bootsma & Hecky, 2003). It is the shallowest and smallest by volume of the Laurentian Great Lakes (Daher, 1999). These characteristics make Lake Erie unique from the other Great Lakes.



The meteorological forcing variables required for FLake model runs include solar (shortwave) and longwave irradiance, air temperature, air humidity, wind speed, and cloudiness. Mean daily air temperature, wind speed and water temperature measurements were obtained from the National Data Buoy Center (NDBC) of NOAA, station 45005 (2003-2012). The station location is shown in Fig. 1 (41°40' N, 82°23' W, and depth: 12.6 m). Air temperature is measured 4 m above the water surface

and anemometer height is 5 m above the water surface to measure the wind speed, whereas the water surface is at 173.9 m above mean sea level. Water temperature is also measured at 0.6 m below the water line. The NDBC station was selected to perform simulations with FLake, since water temperature observations collected at the buoy station can be used to evaluate the model output. The other meteorological forcing variables required for model simulations at the NDBC station were obtained from nearby stations. Air humidity, and cloudiness were available in a daily format from EC-Ontario Climate Center

(OCC) for the Windsor station (climate ID: 6139525) (2003-2012). The location of this station is shown in Fig. 1, which is a near-shore station close to the NDBC station. The distance between OCC and NDBC stations is less than 9.5 km. Incoming radiation fluxes data was supplied by the National Water Research Institute (NWRI), Environment Canada (EC), from a station located in the western basin of Lake Erie (see Fig. 1). Daily shortwave irradiance measurements were available only for 2004 and 2008. Therefore, a daily time series of solar irradiance for the entire study period (2003-2012) was completed for the

NDBC station using solar irradiance model data (see Sect. 2.2). Longwave irradiance was measured only in 2008 at the NWRI-EC station. An empirical equation (see Sect. 2.2) was therefore employed to obtain longwave irradiance for the full period of study (2003-2012).

FLake requires information on water transparency (downward light $K_d$) as input for model runs. MERIS satellite imagery was used to derive $K_d$ for the NDBC station during the study period. The method is described in details in Sect. 2.3. Available

Secchi disk depth (SDD) field measurements were used to estimate lake water clarity. SDD data was provided by EC and utilized in this study to evaluate the satellite-derived $K_d$. Research cruises on board the Canadian Coast Guard Ship *Limnos* visited Lake Erie at a total of 89 distributed stations in five different years (September 2004; May, July, and September 2005; May and June 2008; July and September 2011; and February 2012). The location of stations is shown in Fig. 1.

## 2.2 Shortwave and Longwave Irradiance

The SUNY model, a satellite solar irradiance model, has been developed to exploit Geostationary Operational Environmental Satellites (GOES) for deriving solar irradiance using cloud, albedo, elevation, temperature, and wind speed observations (Kleissl et al., 2013). The basic principles of solar-irradiance modelling based on inputs from geostationary satellites and atmospheric models are described in Kleissl et al. (2013). Data from these sources are used to generate site and time specific high-resolution maps of solar irradiance with the SUNY model. The daily mean solar irradiance data for the present study was

obtained from the second version of the SUNY model (Version 2.4), available in SolarAnywhere® (https://www.solaranywhere.com). The model provides a gridded data set with a spatial resolution of one tenth of a degree (ca. 10 km). The solar irradiance data was extracted from a tile corresponding to the NWRI station (see Fig. 1) for 2004 and 2008, when observations were available for evaluation, and also for FLake model run on Lake Erie for the full study period (2003-





2012). As shown in Fig. 2, there is a strong agreement ($R^2 = 0.93$) between model-derived and measured solar irradiance at the station. The SUNY model slightly underestimates observations by 2.18 Wm$^{-2}$ (N = 362, RMSE = 21.58 Wm$^{-2}$, MBE = -2.18 Wm$^{-2}$, I_a = 0.88; see Sect. 2.5 for details).

Longwave irradiance was computed on a daily basis using the equation of Maykut and Church (1973), as implemented in the Canadian Lake Ice Model (CLIMo) (Duguay et al., 2003):

$$E = \sigma T^4 (0.7855 + 0.000312 G^{2.75})$$   Eq. (1)

where $T$ is the air temperature (°K) and $G$ is the cloudiness in tenth from meteorological stations.

Longwave irradiance calculated from Eq. 1 was evaluated against observations from the NWRI-EC station, only available in 2008 (Fig. 3). The two datasets are highly correlated ($R^2 = 0.74$) with the equation underestimating measured irradiance by 0.86 Wm$^{-2}$ (N = 194, RMSE = 17.74 Wm$^{-2}$, MBE = -0.86 Wm$^{-2}$, I_a = 0.76). Model-derived incoming shortwave and longwave
fluxes were used as input in FLake model simulations for subsequent analyses over the 2003-2012 period.

### 2.3 Satellite-Derived Extinction Coefficient

MERIS operated on-board the ESA Envisat polar-orbiting satellite until April 2012. The sensor was a push-broom imaging spectrometer which measured solar radiation reflected from the Earth's surface high spectral and radiometric resolutions with a dual spatial resolution (300 m and 1200 m). Measurements were obtained in the visible and near-infrared part of the
electromagnetic spectrum (across the 390 nm to 1040 nm range) in 15 spectral bands during daytime, whenever illumination conditions were suitable, and with a full spatial resolution of 300 m at nadir, with a 68.5° field-of-view. MERIS scanned the Earth with a global coverage of every 2-3 days.

In this study, a total of 326 full resolution archived MERIS images encompassing the NDBC station in Lake Erie (see Fig. 1) were acquired from CC (Version 2) products through the Calvalus on-demand processing service for the period of 2003-2012.
CC Level2W products are the result of in-water processing algorithms to derive optical parameters from the water leaving reflectance. These parameters include inherent optical properties (IOPs), concentrations of water constituents, and other optical water parameters such as spectral vertical $K_d$. The IOP parameters are first derived applying two different inversion algorithms: neural network (NN) and Quasi Analytical Algorithm (QAA). The derived IOPs are then converted to estimate constituents' concentrations and apparent optical properties (AOP), including diffuse $K_d$ for different spectral bands applying Hydrolight
simulations (Ruescas et al., 2014).

The diffuse $K_d$ product in CC MERIS L2W data was evaluated against SDD in situ data collected during *Limnos* cruises. The CC-derived diffuse $K_d$ values were extracted for pixels on the same day and location as the *Limnos* cruise stations. The satellite-





derived $K_d$ values were then extracted for the pixel at the geographic location of the NDBC station. A valid pixel expression was defined in all pixel extraction steps that excluded pixels with properties listed in Table 1.

## 2.4 FLake Model and Configuration

The FLake model is a self-similar parametric representation (assumed shape) of the temperature structure in the four media of the lake including water column, bottom sediments, and in the ice and snow. The water column temperature profile is assumed to have two layers: a mixed layer with constant temperature and a thermocline that extends from the base of mixed layer to the lake depth. The shape of thermocline temperature is parameterized using a fourth-order polynomial function of depth that also depends on a shape coefficient $C_T$. The value of $C_T$ lies between 0.5 and 0.8 so that the thermocline can neither be very concave nor very convex. FLake has an optional scheme for the representation of bottom sediments layer, which is based on the same parametric concept (De Bruijn et al., 2014; Martynov et al., 2012). The system of prognostic equations for parameters is described in Mironov (2008).

The prognostics ordinary differential equations are solved to estimate the thermocline shape coefficient, the mixed layer depth, bottom, mean and surface water column temperatures, and also parameters related to the bottom sediment layers (Martynov et al., 2012; Mironov, 2008; Mironov et al., 2010). The mixed layer depth is calculated considering the effects of both convective and mechanical mixing, also accounting for volumetric heating which is through the absorption of net shortwave radiation (Thiery et al., 2014). The non-reflected shortwave radiation is absorbed after penetrating the water column in accordance with the Beer-Lambert law (Martynov et al., 2012; Mironov, 2008; Mironov et al., 2010).

Stand-alone FLake simulations were conducted for the NDBC station. The setup conditions of NDBC buoy station (height of the wind measurement: 5 m, height of the air temperature measurements: 4 m, depth of the water temperature measurements: 0.6 m), the measured meteorological parameters and model-derived irradiance, as well as the geographic location and depth of this site (41°40' N, 82°23' W, and depth: 12.6 m) were used to configure the FLake model. A fetch value of 100 km was used to run all simulations. It was found that there is only little sensitivity to modifications in this parameter for Lake Erie. The same result found for Lake Kivu in Thiery et al. (2014). The bottom sediments module was switched off in all simulations and the zero bottom heat flux condition is adopted. The ability of FLake to reproduce the observed temperature variations using different $K_d$ values was tested by comparing the simulated LSWT to the corresponding in situ observations in the NDBC station. Also, the model sensitivity to variations in water clarity was assessed studying the LSWT, MWCT, MLD, isotherms, ice phenology, and ice thickness.

## 2.5 Accuracy Assessment

To assess the model outputs, three statistical indices were calculated: the root mean square error (RMSE), the mean bias error (MBE), and the index-of-agreement ($I\_a$). RMSE is a comprehensive metric that combines the mean and variance of model errors into a single statistic (Moore et al., 2014). The MBE is calculated as the modelled values minus the in situ observations. Therefore, a positive (negative) value of this error shows an overestimation (underestimation) of the parameter of interest. $I\_a$





is a descriptive measure of model performance. It is used to compare different models and also modelled against observed parameters. I_a was originally developed by Willmott in the 1980s (Willmott, 1981) and a refined version of it was presented by Willmott et al. (2012). The refined version, which was adopted in this study, is dimensionless and bounded by -1.0 (worst performance) and 1.0 (the best possible performance). These statistical indices are considered as robust measures of model

performance (e.g. Hinzman et al., 1998; Kheyrollah Pour et al., 2012; Willmott and Wicks, 1980).

## 3 Results and Discussion

### 3.1 Satellite-Derived $K_d$

#### 3.1.1 Evaluation of CoastColour $K_d$

The assessment of the satellite-derived $K_d$ retrieval reliability highly depends on the comparison with independent in situ SDD

measurements. The relation between $K_d$ and SDD was established by the pioneer study of Poole and Atkins (1929):

$$SDD \times K_d = K \hspace{5cm} \text{Eq. (2)}$$

where $K$ is a constant value of 1.7 (Poole and Atkins, 1929). Studies have found a high variability of this constant depending on the type of the lake considered (Koenings and Edmundson, 1991). Armengol et al. (2003) also show that $K_d$ and SDD are negatively correlated and they developed an empirical relation between these two parameters using Eq. (2).

In this study, applying a cross validation approach, an empirical relation based on Eq. (2) was developed between in situ measured SDD and CC-derived $K_d$. SDD measurements were conducted 117 times during cruises on Lake Erie from 2004 to 2012. These spatially-distributed measurements have minimum, maximum, mean, and standard deviation values of 0.2, 11, 3.69, and 2.68 m, respectively. CC L2W satellite products were acquired on the same day as the in situ measurements. Applying defined flags produced 49 data pairs (matchup dataset) of CC observations of $K_d$ and SDD in situ data that were collected on

the same day and location.

The matchup dataset was divided into training and testing data in 100 iterations. In each iteration, the data used for the equation's training and evaluation were kept independent, where 70% of the sample was used for equation calibration and 30% for evaluation. Ordinary least square regression was used in the calibration step of each iteration to relate the in situ measurements of SDD to the CC-derived $K_d$. Locally tuned equations were derived from this step and applied on SDD

observations to predict $K_d$ in testing matchup data. The statistical parameters of the model performance were derived between the estimated $K_d$ from SDD observations and the paired CC-derived values. These steps were repeated for 100 iterations; and the final statistical indices, slope and power of the locally tuned equation was reported as the average of the ones derived over all iterations.

Results from the above procedure show that $K_d$ and SDD are strongly correlated with $R^2 = 0.78$. The extinction coefficient can

be derived from equation $K_d = 1.64 \times SDD^{-0.76}$. There is a good agreement between the satellite-derived $K_d$ and the corresponding ones estimated from in situ measured SDD (N = 49, RMSE = 0.63 m$^{-1}$, MBE = -0.09 m$^{-1}$, I_a = 0.65) (Fig. 4).



Arst et al. (2008) obtained a similar regression formula between SDD and $K_d$ for the boreal lakes in Finland and Estonia representing different types of water, expanding from oligotrophic to hypertrophic. SDD is a suitable characteristic to describe water transparency for small values of $K_d$. However, for high values of $K_d$ (ranging above 4 m$^{-1}$), Arst et al. (2008) and Heiskanen et al. (2015) suggest that SDD is unable to describe any changes in $K_d$. Fig. 4 also shows that SDD cannot describe

the scatter of $K_d$ for values above 4 m$^{-1}$. Therefore, the estimation of $K_d$ from SDD should be used with caution, motivating the investigation on the potential of integrating satellite-based estimations of $K_d$ into lake models.

### 3.1.2 Spatial and Temporal Variations in $K_d$

This section describes how the CC satellite observations can explain the spatial and temporal variations of $K_d$. Spatial variations of $K_d$ derived from the CC algorithm are shown in Fig. 5 for a selected day (3 September 2011). This particular day of 2011 is

selected as the lake experienced its largest algal bloom in its recorded history in that year, before the new recent record of 2015 (Michalak et al., 2013; NOAA, 2015). The bloom was expanding from the western basin into the central basin. Algal bloom is one of the factors affecting the water clarity of Lake Erie (NOAA, 2015). Other parameters include the concentrations of suspended and dissolved matters in the lake. The western basin is the shallowest region of the lake; and therefore is the most vulnerable to sediment re-suspension that also results in reducing water clarity. The map shows that Lake Erie experienced

different levels of clarity in various locations with an average $K_d$ value of $0.90 \pm 0.80$ m$^{-1}$ over the entire lake. The NDBC station is also shown on the map as a reference (with $K_d = 0.87$ m$^{-1}$ on 3 September 2011).

Since fully cloud-free MERIS satellite images for consecutive months were only available in 2010, four months (May-August 2010) are selected to illustrate variations in $K_d$ on a monthly-basis for one year (Fig. 6). $K_d$ over the full lake during May, June, July, and August has average values of $0.82 \pm 0.85$ m$^{-1}$, $0.72 \pm 1.10$ m$^{-1}$, $0.73 \pm 1.20$ m$^{-1}$, $0.78 \pm 0.55$ m$^{-1}$, respectively. The western

basin is always experiencing the lowest levels of water clarity in comparison to other regions of the lake, with a maximum $K_d$ in May. This can be the result of a spring algal bloom, and also wind-driven re-suspension of sediments. $K_d$ at the NDBC station varies between $0.68$ m$^{-1}$, $0.62$ m$^{-1}$, $0.66$ m$^{-1}$, and $0.85$ m$^{-1}$ from May to August 2010, respectively.

Two MERIS images with full coverage of Lake Erie were only available in May of two consecutive years (2008 and 2009). Hence, the MERIS images of May 2008 and May 2009 were selected to show variations in $K_d$ between the two years. Although

the images are for the same month of the year, $K_d$ still varies across the lake (Fig. 7). In May 2008 an average value of $0.77 \pm 0.49$ m$^{-1}$ is estimated for the entire lake, while in May 2009 the average value is $0.90 \pm 0.93$ m$^{-1}$. Comparing the estimated maps for the two years suggests that the spring bloom in 2009 was stronger than the one in 2008 for the western basin. However, algal bloom in all basins of Lake Erie for the complete year of 2008 was recorded as the third largest that the lake experienced before the occurrence of the breaking record blooms in 2011 and 2015 (Michalak et al., 2013; NOAA, 2015). $K_d$ value estimated for

the NDBC station is $0.69$ and $0.62$ m$^{-1}$ in May 2008 and 2009, respectively.

Fig. 8 depicts variations of $K_d$ for the NDBC station during the full study period (2003-2012). In the shallow section of Lake Erie, re-suspension of bottom sediments is the most important factor that leads to higher water clarity. Therefore, the highest $K_d$ values are related to the turn-over times in spring and fall. The results from applying the CC algorithm on MERIS satellite





imagery show that the maximum value of $K_d$ is 3.54 m$^{-1}$, estimated in April 2003. A minimum value of 0.58 m$^{-1}$ is estimated in June 2007. The average value of $K_d$ during the study period is 0.90 m$^{-1}$ with a standard deviation of 0.38 m$^{-1}$. Hence, these values, identified as the average, the lower, and the upper limits of clarity at the NDBC station were used to carry out a sensitivity analysis with FLake (see Sect. 3.2.2).

## 3.2 FLake Model Results

### 3.2.1 Improvement of LSWT Simulations with Satellite-Derived $K_d$

Martynov et al. (2012) focused on 2005 to 2007 to run FLake at the NDBC station using a constant value of 0.2 m$^{-1}$ for $K_d$. They simulated the lake using both realistic and excessive depths of 20 and 60 m, respectively, for a grid tile corresponding to the NDBC station. They showed that applying a more realistic lake depth parameterization improved the performance of the model to reproduce the observed surface temperature. In this section, $K_d$ values were derived from the CC algorithm for different months during the same years (2005-2007) as in Martynov et al. (2012).

Table 2 displays the average $K_d$ values for each month of these years. The monthly averaged values are only focused on the months of the year when both LSWT observations and CC–derived $K_d$ values were available. The average value of $K_d$ in these months in each year is considered as the average value of $K_d$ for that year.

Fig. 9 shows the results of different LSWT FLake simulations at the NDBC station. The model was run first applying $K_d = 0.2$ m$^{-1}$ from Martynov et al. (2012) using both the real lake depth at the station (12.6 m: CRCM-12.6) and also a tile depth corresponding to the station in their study (20 m: CRCM-20). Then, simulations using the yearly average CC-derived $K_d$ for each year of study are plotted (Avg). The $K_d$ values derived from the monthly average of each year were used to simulate the surface water temperature and produce a merged LSWT product. Results of the merged product are also plotted (Merged).

Comparing LSWT in situ observations with the modelled values in Fig. 9 demonstrate that in Avg and Merged simulations for 2005-2007, surface temperature in spring (April-June) is modelled warmer and in summer-fall (July-November) colder than in situ observations (spring: MBE $_{Avg}$ = 1.31 ℃, MBE $_{Merged}$ = 1.25 ℃; summer-fall: MBE $_{Avg}$ = -1.27 ℃; MBE $_{Merged}$ = -1.37 ℃). CRCM-12.6 and CRCM-20 are reproducing a colder LSWT in average with maximum under-prediction in July-August (for 2005-2007: -2.93 ℃ <MBE$_{July-August}$<-0.99 ℃). Simulation with a larger depth (CRCM-20) tends to more slowly gain (lose) heat in spring (fall), compared to all other simulations.

The performance of each simulation is summarized in Table 3 during the period of data availability. For all years, the average and merged simulations perform better than simulations using $K_d$ (0.2 m$^{-1}$) as in Martynov et al. (2012), with improvement in RMSE and MBE for both real depth and tile depth. In all three years, LSWT simulated from the $K_d$ value employed in Martynov et al. (2012) results in an underestimation (CRCM-12.6: MBE= -1.52 ℃, -0.98 ℃, -1.08 ℃; CRCM-20: MBE= -1.54 ℃, -1.09 ℃, -1.35 ℃; during years 2005, 2006, and 2007, respectively). In 2005, the average of $K_d$ for the year demonstrates a better performance compared to the merged results; contrary to the results of 2007. However, for the merged results in 2006, the MBE is improved compared to the simulation using the average $K_d$; whereas its performance decreases in terms of RMSE.





The extent of $K_d$ variations in each month might not be captured by the available MERIS images due to cloud coverage in MERIS images or the absence of any satellite overpass. Therefore, the merged results cannot always perform better than the year average, which can be more representative of $K_d$ variations. Considering the months of September-November into the calculations of MBE for 2005 can be the reason of underestimating LSWT in this year for both Avg and Merged simulations compared to two other years (2006-2007). Turbid waters in these months simulate colder LSWT.

Fig. 10 illustrates the scatterplots of simulated LSWT for all four different runs including three years of data (2005-2007), in comparison with the corresponding in situ observations. All simulated results are in a high agreement with in situ measurements. CRCM simulations (both depths of 12.6 and 20 m) under-predict LSWT with MBE values of -1.26 ℃ and -1.37 ℃, respectively. The under-prediction of these model runs is particularly stronger for LSWT above 12℃ which can be explained by the $K_d$ value used, since both depths considered in CRCM runs are as affected. However, the CRCM-20 simulation tends to produce the coldest LSWT (the most under-prediction; MBE = -1.37 ℃). This can be explained by the lake depth value considered for the model run. This shows clearly that applying a realistic lake depth and $K_d$ value will improve model results and therefore the parameterization schemes.

Fig. 10-a and -b show that the resulting LSWT from yearly average (Ave) and monthly average (Merged) $K_d$ are not significantly different, whereas simulations with yearly average $K_d$ reproduces LSWT with improved RMSE and MBE values compared to monthly average (Avg: RMSE=1.54 ℃, MBE=-0.08 ℃; Merged: RMSE=1.57 ℃, MBE=-0.14 ℃). It is possible that the extent of $K_d$ variations is best represented by the yearly average value. Therefore, using a constant annual open water season value for $K_d$ could be sufficient to simulate LSWT in 1-D lake models with relatively high accuracy. The time-dependent (monthly average) $K_d$ does not improve simulation results for Lake Erie ($K_d$ ranging from 0.58 to 3.54 m$^{-1}$ with average value of 0.90 m$^{-1}$ during open water seasons of 2002-2012). However, comparing results from Fig. 10-a and –c shows improvement in LSWT simulations when a lake-specific value of $K_d$ is used (Avg: RMSE=1.54 ℃, MBE=-0.08 ℃; CRCM-12.6: RMSE=1.76 ℃, MBE= -1.26 ℃). Under-prediction of LSWT decreases when the yearly-average CC-derived $K_d$ values are used, rather than a generic constant value (0.2 m$^{-1}$). Heiskanen et al. (2015) suggest that the effect of $K_d$ seasonal variations on LSWT simulations are not significant for lakes with $K_d$ values higher than 0.5 m$^{-1}$ (e.g. Lake Erie). Therefore, a lake-specific, time-independent, and constant value of $K_d$ can be used in 1-D lake models when the $K_d$ values are higher than 0.5 m$^{-1}$.

Martynov et al. (2012) conclude that applying a more realistic lake depth parameterization improves the FLake model performance. Using the realistic lake depth (12.6 m) at the NDBC station slightly improves the model performance in reproducing LSWT compared to simulations employing the corresponding tile depth (20 m) (CRCM-12.6: RMSE=1.76 ℃, MBE= -1.26 ℃; CRCM-20: RMSE=1.88 ℃, MBE= -1.37 ℃) (Fig. 10-c and –d).

### 3.2.2 Sensitivity of FLake to $K_d$ Variations

The sensitivity of FLake to LSWT, MWCT, MLD, isotherm, ice phenology and thickness simulation using different values of $K_d$ is investigated in this section for year 2008. As indicated previously (Sect. 2.1), shortwave irradiance measurements were





available in that year and longwave irradiance was also measured from May to October 2008. Therefore, longwave irradiance for the other months of 2008 was modelled as described in Sect. 2.2 to fill the temporal gaps. Fig. 11 presents simulation results for LSWT and MWCT using the lowest, average, and highest values of $K_d$ observed in the study period (minimum $K_d$=0.58 $m^{-1}$, average $K_d$ =0.90 $m^{-1}$, maximum $K_d$ =3.54 $m^{-1}$). The water temperature simulation from CRCM-12.6 (realistic depth at station) simulation is also plotted.

In the case of extreme clear water ($K_d = 0.2$ $m^{-1}$; CRCM-12.6), LSWT shows smoother variations during the open water season in 2008 as opposed to the most turbid water simulation (maximum or Max) which displays more abrupt LSWT variations (Fig. 11). This is because solar radiation is absorbed faster in turbid waters. It penetrates less deeply and warms up the shallow surface layer causing thinner mixing depth. This shallow layer exchanges heat faster with the atmosphere, resulting in sudden surface water temperature variations as opposed to clear waters. Also, the maximum turbid water simulation shows warmer LSWT in spring and colder LSWT in fall compared to the results of the more clear water simulation. In spring (the start of heating season), darker surface waters absorb heat faster than clear water because of existing particles in water. However, in fall the loss of energy to the atmosphere is also faster due to the shallow mixing depth. On average, the most turbid water simulation (Max) resulted in 0.09 ℃ higher LSWT compared to the average (Avg) simulation, whereas the least turbid water (minimum or Min) simulation produced on average 0.02 ℃ colder LSWT during 2008. CRCM-12.6 simulation with $K_d$ value of 0.2 resulted in a larger difference compared to Avg simulation, 0.55 ℃ colder LSWT. The comparison of the simulated LSWT results show that FLake is not significantly sensitive to LSWT when $K_d$ value varies in the range of our Min to Max $K_d$. However, the sensitivity increases rapidly for $K_d$ values less than our Min (0.58 $m^{-1}$). Rinke et al. (2010) conclude that the thermal structure of lakes is particularly sensitive to changes in $K_d$ when its value is below 0.5 $m^{-1}$. More recently, Heiskanen et al. (2015) confirmed the critical threshold of $K_d$ (ca. 0.5 $m^{-1}$). They suggest that the response of 1-D lake models to $K_d$ variations is nonlinear. The models are much more sensitive if the water is estimated to be too clear. Heiskanen et al. (2015) recommend to use a value of $K_d$ that is too high rather than too low in lake simulations, if the clarity of lake is not known exactly.

For both clear and dark waters, LSWT is warmer than the MWCT, due to being exposed to more intense solar radiation. Shortwave radiation is attenuated as it reaches a greater depth, particularly in turbid waters. In the extreme clear water simulation, the MWCT is on average 0.99º C colder than LSWT, whereas for the most turbid water the average difference is much higher equal to 4.82º C.

The MWCT in the most turbid condition (Max) is less than for all other clear water simulations. This is because the lower layers in dark waters accumulate less heat during the heating season as opposed to clear waters which results in less heat storage and lower water column temperature in turbid waters (Heiskanen et al., 2015; Potes et al., 2012). The solar radiation penetrates less deeply and is absorbed by the surface layer, thereby heating it; where the surface layer transfers the energy faster to the atmosphere, resulting in a colder water column in turbid waters. The MWCT decreases by 0.94 $^0$C (increases by 0.63 $^0$C) when $K_d$ changes from its average to its maximum (minimum) value during the study period. The increase in MWCT



is even larger when $K_d$ changes from its average to 0.2 m$^{-1}$ (2.25 $^0$C). Therefore, $K_d$ variations have a larger impact on MWCT than on LSWT, and the largest difference is when $K_d$ is estimated to be extremely clear.

Fig. 12 shows variations of the MLD in 2008 derived from simulations using the Min, Ave, and Max $K_d$, and CRCM-12.6 simulation. All simulations show two turnover (complete mixing) events, spring and fall. The highest depth of spring mixing is at the same time for all simulations; however, fall mixing occurs at different dates for each simulation. CRCM-12.6 reaches its maximum MLD (fall turnover) at the end of summer (August 28), while the other three runs show that the fall turnover takes place in late fall, before ice forms. The Min simulation reaches its highest MLD in early November (November 3), earlier than the Avg and Max simulations (November 21). As a result, the water column in clear water reaches the temperature of maximum density ($4^0$C) much faster than turbid water and therefore the turnover happens earlier.

In the turbid water simulation (Max), the MLD is shallower than the other simulations (an average difference of 4.94 m in 2008 between two simulations with extreme $K_d$ values). In turbid waters, solar radiation does not penetrate as far beyond the water surface as opposed to clear waters; and it will get absorbed by the particles in water. Therefore, clear waters have a deeper mixed layer when the solar radiation can penetrate further and distribute to a larger volume in the water column. CRCM-12.6 produces a MLD of 3.47 m deeper compared to Avg simulation, whereas the Min (Max) simulations result in MLD of 1.15 m (1.47 m) deeper (shallower) compared to the Avg simulation. Hence, clear water simulates deeper MLD; and the effect of $K_d$ on the MLD is larger when the $K_d$ value is estimated to be too clear. The MLD is influenced by the water column thermal structure. Fig. 13 displays the simulated isotherms derived from using different $K_d$ values. Comparing isotherms for dark and clear waters confirms the results presented in Fig. 12. It demonstrates that the mixed layer in dark waters is not only shallower as opposed to the clear waters, but also warmer. The reason is that solar radiation is mostly absorbed at the upper layers in turbid water. Thus, the radiation is used to warm up a thinner layer in dark waters leading to higher temperatures. Fig. 13 also shows that the deepening of the thermocline layer in clear waters is monotonic; whereas in dark waters it is slower, as the heat transfer in dark waters is slower between water layers to stabilize the temperatures in different layers.

Fig. 14 depicts the monthly average of temperature profiles in 2008 for different values of $K_d$. A warmer epilimnion at the beginning of the heating season occurs in dark waters, whilst temperature in the epilimnion reduces later in fall compared to clear waters. There are a number of factors determining the epilimnion temperature in lakes, including the radiation fluxes (sensible heat, latent heat, and longwave radiation), and cooling effects from the water below. Persson and Jones (2008) conclude that for colored waters (turbid), the combination of these heating and cooling effects leads to a warmer epilimnion initially. However, a lower temperature in the epilimnion is followed due to the gradual lessening of the radiative absorption and increased effect of cooling from the layers below. Fig. 14 supports observations by Persson and Jones (2008) and Heiskanen et al. (2015) that the depth of the thermocline layer is always deeper in clear waters due to the faster heat distribution between different underneath layers, resulting in a colder temperature but thicker and deeper epilimnion. However, the extreme clear water simulation reproduces a warmer hypolimnion as opposed to the other ones, due to the fact that light penetration in clear waters warms up the lower layers (Heiskanen et al., 2015).




Fig. 15 shows the impact of $K_d$ variations on lake ice phenology and thickness in winter 2008 (January-March). Freeze-up corresponds to the earliest date that the NDBC station is completely covered by ice, and the earliest date the station is completely free of floating ice is defined as break-up. The Avg simulation reproduces similar ice phenology as the Max simulation, whereas Min and CRCM-12.6 result in the similar break-up/freeze-up dates. The break-up in CRCM-12.6 and Min simulations are on March 23, one day earlier than Max and Avg simulations and freeze-up occurs on January 24, two days after Max and Avg simulations. CRCM-12.6 and Min simulations reproduce 1.28 and 1.27 cm thinner ice than Avg simulation in 2008, respectively. The most turbid water (Max) reproduces 0.21 cm thicker ice in 2008 compared to the Avg simulation. The ice sheet forms later in clear waters (CRCM-12.6 and Min) and disappears earlier compared to dark waters (Max and Avg), resulting in a shorter ice cover duration (3 days) and hence thinner ice in clear water simulations. Lake morphological properties determine ice cover as well as climatic factors. Among morphological aspects, lake depth is the most important factor that can impact the ice cover by influencing the amount of heat storage in the water and hence the time needed for the lake to cool and ultimately freeze (Brown and Duguay, 2010). For a given depth and climatic condition, however, the amount of heat storage is determined by water clarity. Dark waters store more heat in a shallower depth. Therefore, in the winter time the heat can be transferred faster to the atmosphere through the lake surface, resulting in an earlier freeze-up as mentioned in Heiskanen et al (2015) that freeze-up occurs earlier in more turbid waters. However, as shown by simulations with 12.6 m, ice phenology is minimally affected by $K_d$ value in FLake. For a larger depth or in a different model, the impact of $K_d$ values in ice onset should be investigated.

## 4 Summary and Conclusion

Spatial and temporal variations of $K_d$ in Lake Erie were derived from the globally available satellite-based CC product during open water seasons 2002-2012. The CC product was evaluated against SDD in situ measurements. CC-derived $K_d$ values, modelled incoming radiation flux data, in addition to complementary meteorological observations during the study period, were used to force the 1-D FLake model. The model was run for a selected site (NDBC buoy station) on Lake Erie, a large shallow temperate freshwater lake.

FLake was run with the range of clarity values acquired from satellite observations. Results were compared to a previous study which assumed a constant $K_d$ value due to the lack of data. Results clearly showed that applying satellite-derived $K_d$ values improves FLake model simulations using a derived yearly average value as well as monthly averaged values of $K_d$. Although $K_d$ varies in time, a time-invariant (constant) annual value is sufficient for obtaining reliable estimates of LSWT with FLake for Lake Erie. It was also shown that the model is very sensitive to variations in $K_d$ when the value is less than 0.5 m$^{-1}$.This finding is in agreement with the study of Rinke et al. (2010) and the recent study of Heiskanen et al. (2015) who determined that the impact of seasonal variations of $K_d$ on the simulated thermal structure is small, for a lake with $K_d$ values larger than 0.5 m$^{-1}$. The studies suggest that the response of 1-D lake models to $K_d$ variations is nonlinear. The models are much more sensitive if the water is estimated to be too clear. Heiskanen et al. (2015) recommend to use a value of $K_d$ that is too high rather





than too low in lake simulations, if the clarity of lake is not known exactly. Results of our study showed that the sensitivity to $K_d$ variations was more pronounced in simulation results for MWCT and MLD compared to LSWT.

Results of this study has important implications for understanding the thermal regime of lakes and shows that the transparency of lakes can impact physical processes by influencing changes in seasonal mixing regime. Integrating lake specific $K_d$ values

can improve the performance of 1-D lake models. However, field measurements of $K_d$ are not widely available. This study demonstrates that satellite observations are a reliable data source to provide lake models with global estimates of $K_d$ with high spatial and temporal resolutions. The globally available CC product can be used as a source to fill the gaps in $K_d$ in situ observations, and improve the performance of parameterization schemes and, as a result, further improve the NWP and climate models. Although MERIS is no longer active, the Ocean and Land Colour Instrument (OLCI) to be operated on the ESA

Sentinel-3 satellite (launched on February 16, 2016) will provide continuity of MERIS-like data. OLCI has MERIS heritages and improves upon it with an additional six spectral bands.

**Author Contribution**

The presented research is the direct result of a collaboration with the listed co-authors. All materials in composition of the research article is the sole production of the primary investigator listed as first author. Dr. Claude R. Duguay and Dr. Homa

Kheyrollah Pour supported this research through comments and advice related to the FLake model. The manuscripts were edited for content and composition by the co-authors.

**Acknowledgment**

The authors would like to thank Dr. Caren Binding (Environment and Climate Change Canada) for providing the optical in situ data of Lake Erie, Dr. Ram Yerubandi (Environment and Climate Change Canada) for providing the meteorological station

data for Lake Erie, and Dr. Andrey Martynov for providing advice related to running the FLake model. Financial assistance was provided through a Discovery Grant from the Natural Sciences and Engineering Research Council of Canada (NSERC) to Claude Duguay.

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





Table 1 Flags of excluded pixels

| Level 1 | Level 1P | Level 2 |
|---|---|---|
| Glint_risk | Land | AOT560_OOR (Aerosol optical thickness at 550 nm out of the training range) |
| Suspect | Cloud | TOA_OOR (Top of atmosphere reflectance in band 13 out of the training range) |
| Land_ocean | Cloud_ambigious | TOSA_OOR (Top of standard atmosphere reflectance in band 13 out of the training range) |
| Bright | Cloud_buffer | Solzen (Large solar zenith angle) |
| Coastline | Cloud_shadow | NN_WLR_OOR (Water leaving reflectance out of training range) |
| Invalid | Snow_ice | NN_CONC_OOR (Water constituents out of training range) |
| | MixedPixel | NN_OOTR (Spectrum out of training range) |
| | | C2R_WHITECAPS (Risk of white caps) |





Table 2 CC-derived average values of Kd for each month (2005-2007). The values correspond to the time of year when water LSWT observations, as well as the CC derived Kd values, are available.

| Year | Apr. | May | June | July | Aug. | Sep. | Oct. | Nov. | Avg. |
|------|------|------|------|------|------|------|------|------|------|
| 2005 | -- | 0.69 | 0.62 | 0.63 | 0.79 | 1.07 | 0.92 | 0.97 | 0.81 |
| 2006 | 0.82 | 0.70 | 0.62 | 0.65 | 0.77 | -- | -- | -- | 0.71 |
| 2007 | 0.86 | 0.72 | 0.64 | 0.65 | 0.76 | -- | -- | -- | 0.73 |





Table 3 Simulated LSWT compared to in situ observations (2005 – 2007). Period corresponds to the time of year when LSWT and $K_d$ values were available.

| Period | $K_d$ | RMSE | MBE | I_a |
|---|---|---|---|---|
| 2005 May-Nov | Avg2005 | 1.69 | -0.86 | 0.87 |
| | Merged | 1.76 | -0.95 | 0.86 |
| | CRCM-12.6 | 1.88 | -1.52 | 0.85 |
| | CRCM-20 | 2.12 | -1.54 | 0.83 |
| 2006 Apr-Aug | Avg2006 | 1.40 | 0.59 | 0.89 |
| | Merged | 1.42 | 0.54 | 0.89 |
| | CRCM-12.6 | 1.50 | -0.98 | 0.89 |
| | CRCM-20 | 1.47 | -1.09 | 0.89 |
| 2007 Apr-Aug | Avg2007 | 1.37 | 0.62 | 0.90 |
| | Merged | 1.35 | 0.57 | 0.91 |
| | CRCM-12.6 | 1.78 | -1.08 | 0.86 |
| | CRCM-20 | 1.80 | -1.35 | 0.87 |





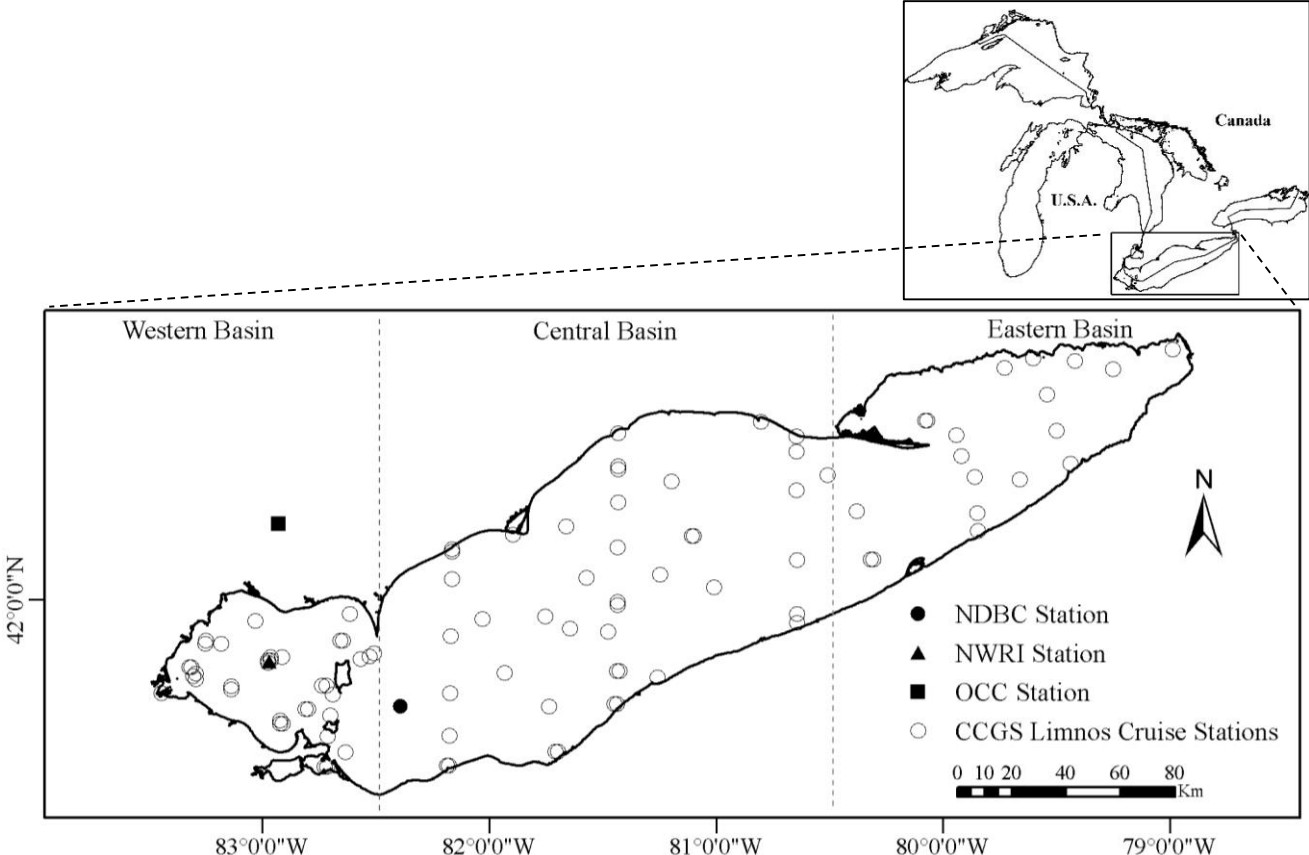

Fig. 1 Maps showing Lake Erie in Laurentian Great Lakes and the location of stations where different parameters were measured.





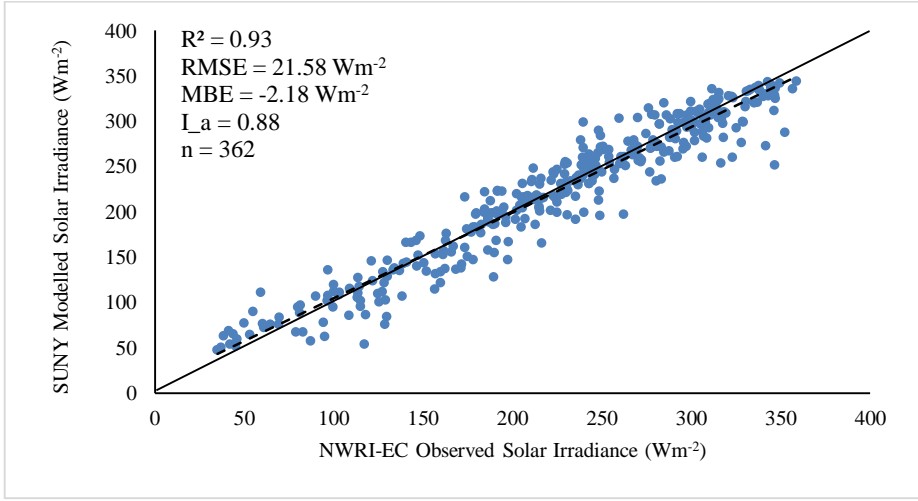

Fig. 2 Scatter plot of NWRI-EC and SUNY mean daily solar irradiance (data from 2004 and 2008). The obtained statistical indices are included. The dashed line shows the best-fit line. Solid line corresponds to 1:1 relationship.





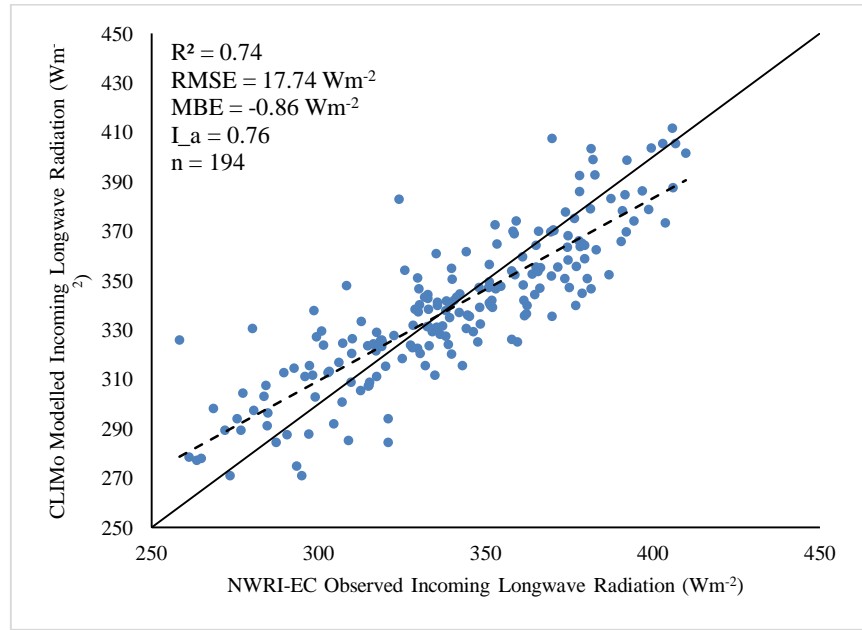

Fig. 3 Scatter plot of NWRI-EC and CLIMo mean daily incoming longwave radiation (data from 2008). The obtained statistical indices are included. The dashed line shows the best-fit line. Solid line corresponds to 1:1 relationship.





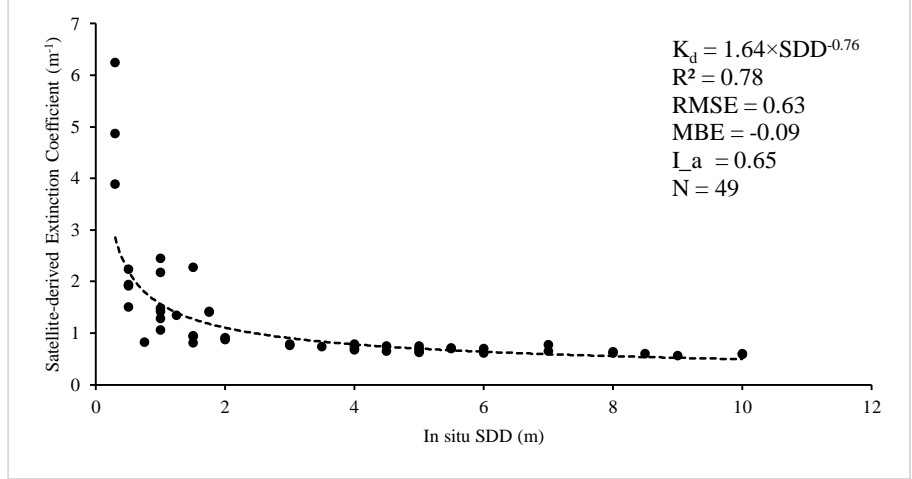

Fig. 4 Relation between satellite-derived $K_d$ and in situ SDD matchups.



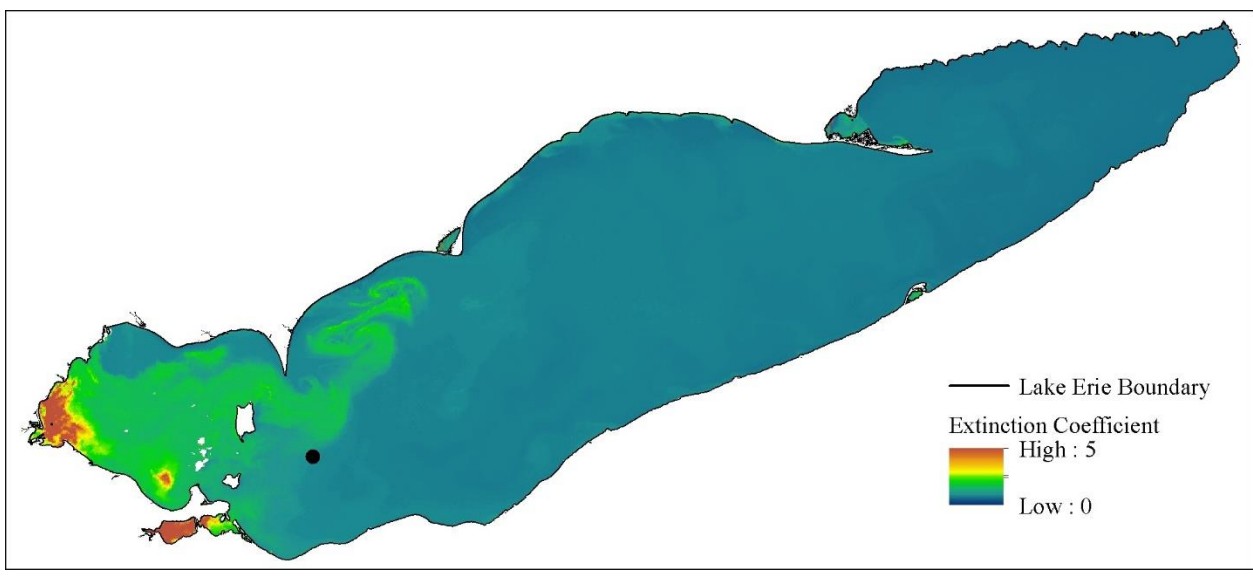

Fig. 5 Spatial variation of satellite-derived $K_d$ in Lake Erie, on 3 September 2011. Location of NDBC station is shown on the map as a solid dot.





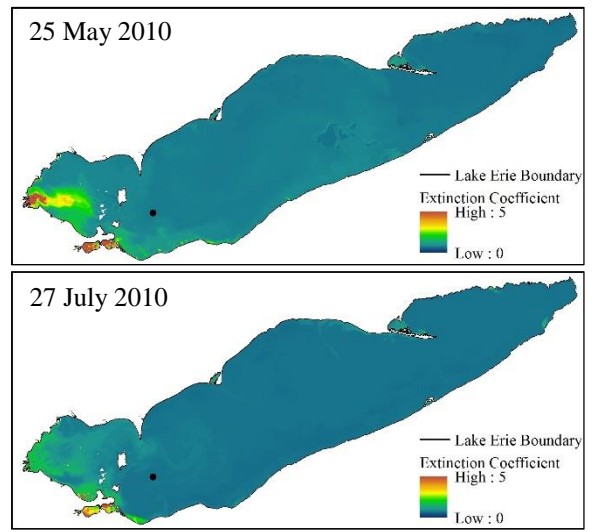
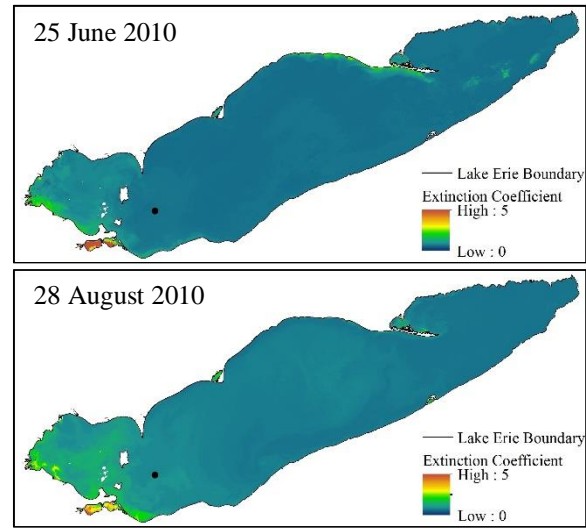

Fig. 6 Temporal and spatial variation of satellite-derived $K_d$ in Lake Erie for different months of a year: May- August 2010.
Location of NDBC station is shown on the map as a solid dot.





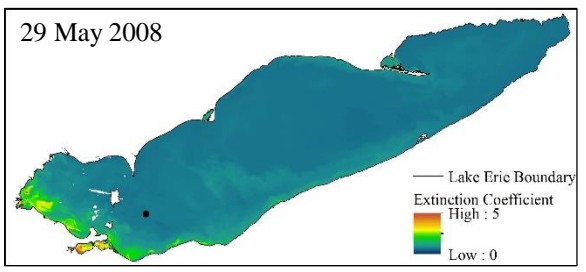
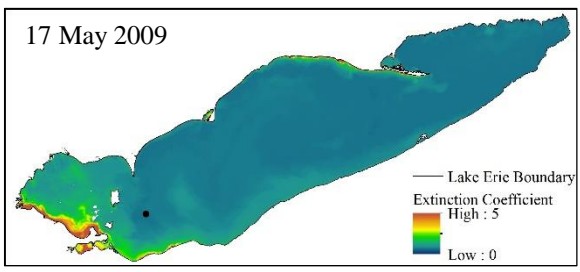

Fig. 7 Temporal and spatial variation of $K_d$ in Lake Erie during May of two consecutive years: 2008 and 2009. Location of NDBC station is shown on the map as a solid dot.



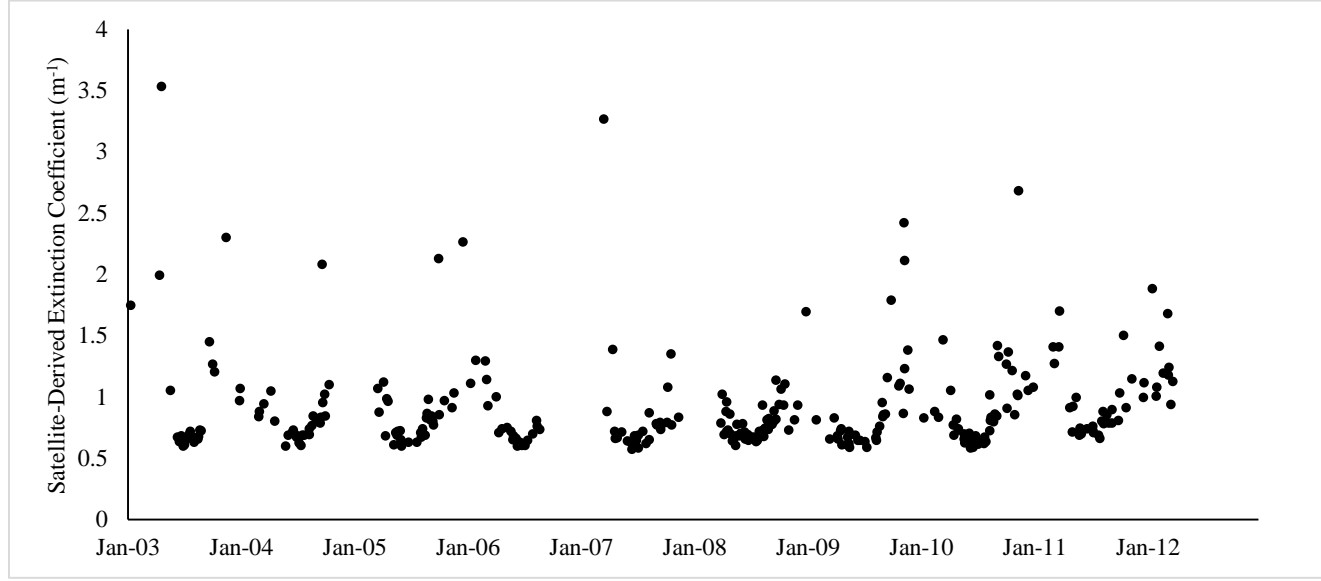

Fig. 8 Variations of CC-derived Kd for the selected location during the study period (2003-2012).





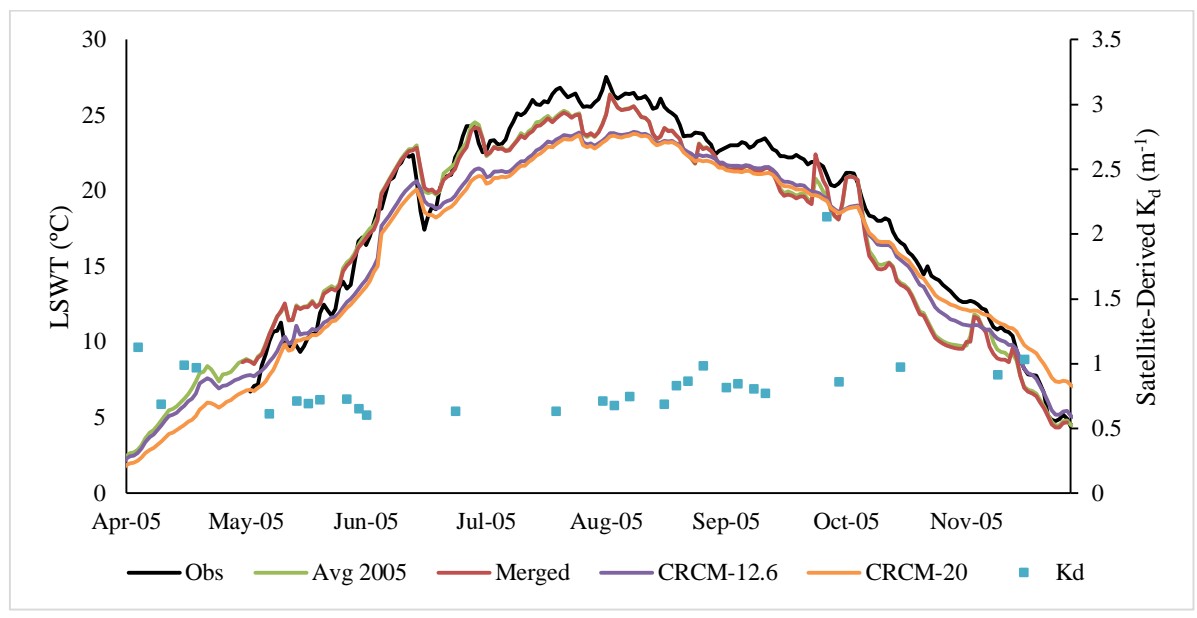

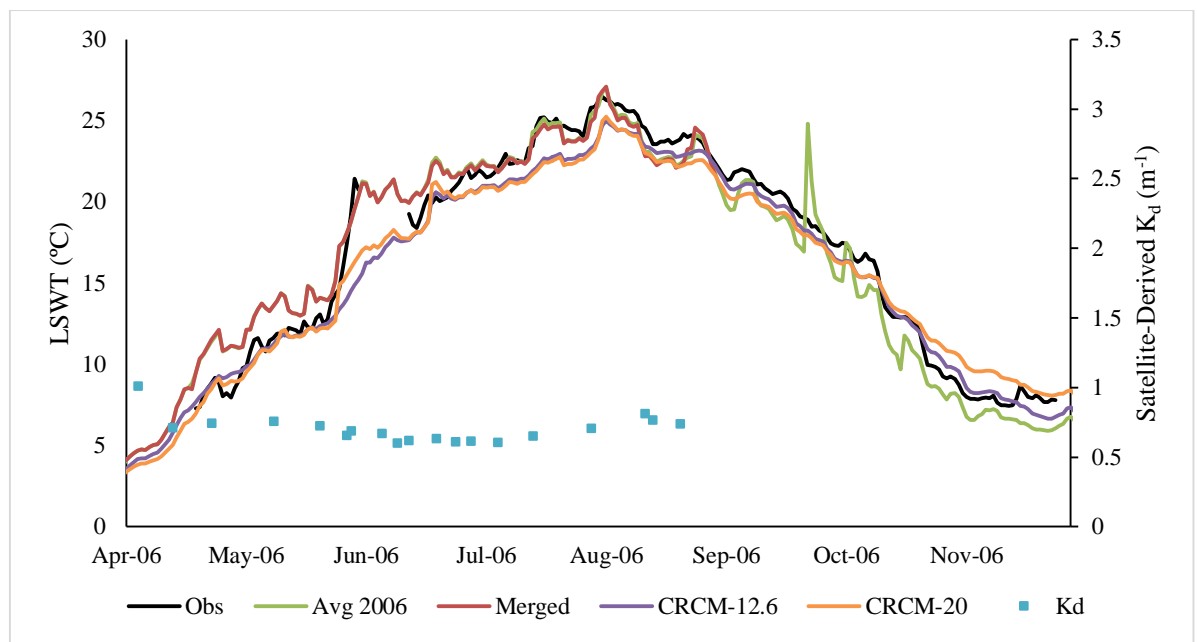



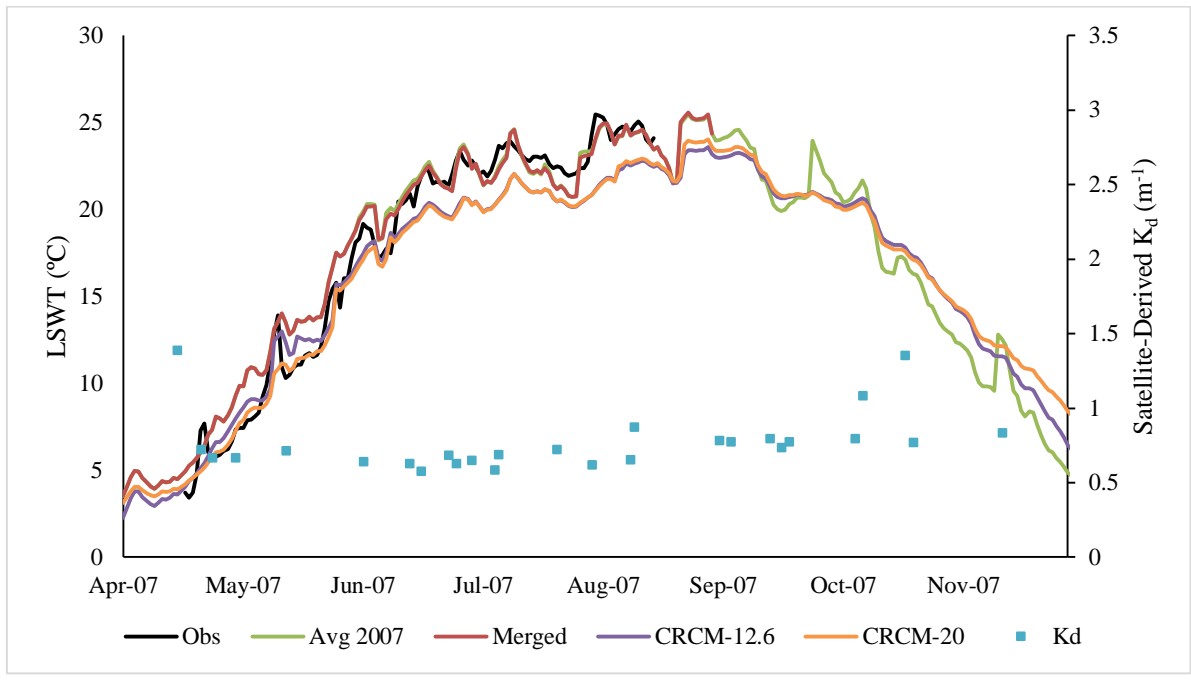

Fig. 9 LSWT simulation results for 2005 - 2007; from: CRCM-12.6, CRCM-20, CC-derived average for $K_d$ during selected month of each year (0.81, 0.71, and 0.73 m$^{-1}$; respectively), and the merged simulations based on each month average $K_d$. The corresponding observations for LSWT, and CC-derived $K_d$ values are also plotted.







Fig. 10 Modelled (y-axis) versus observed (x-axis) LSWT for yearly average, merged, CRCM-12.6, and CRCM-20 simulations during the ice-free seasons in 2005-2007. A linear fit (dashed line) and its coefficients are shown on the plot. The statistics related to the regression of parameters, and a 1:1 relationship (solid line) are also shown.







Fig. 11 LSWT and MWCT simulation results in 2008, when using the lowest (Min), average (Avg), and the highest (Max) Kd values. Results from the CRCM-12.6 simulation is also plotted.





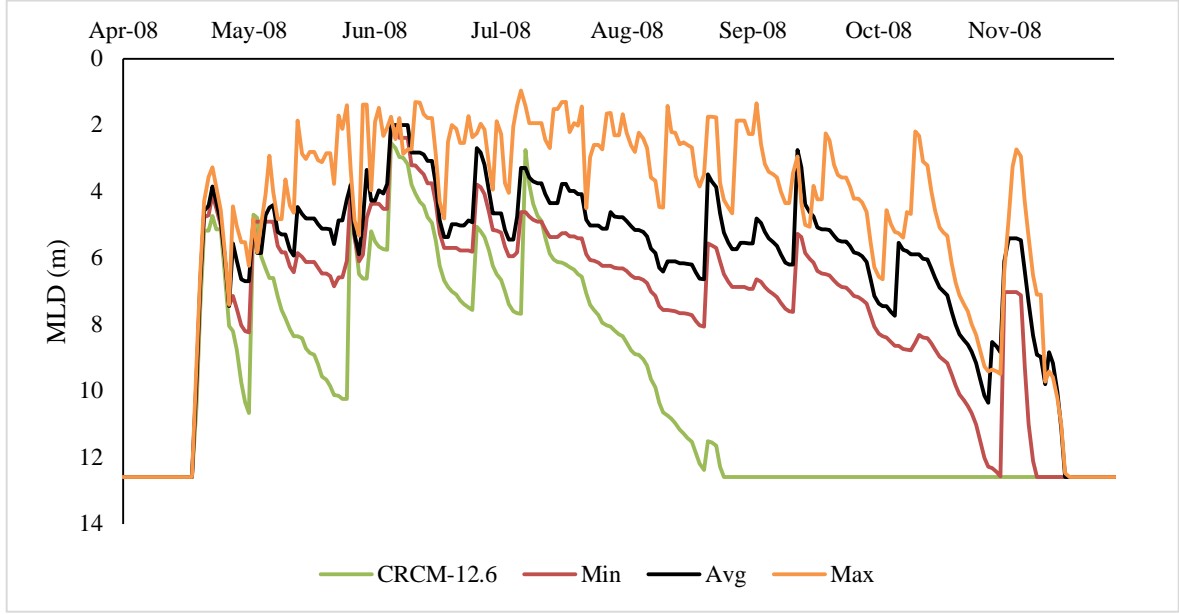

Fig. 12 MLD simulation results for the lowest (Min), average (Avg), and the highest (Max) $K_d$ values in 2008. CRCM-12.6 results are also plotted.





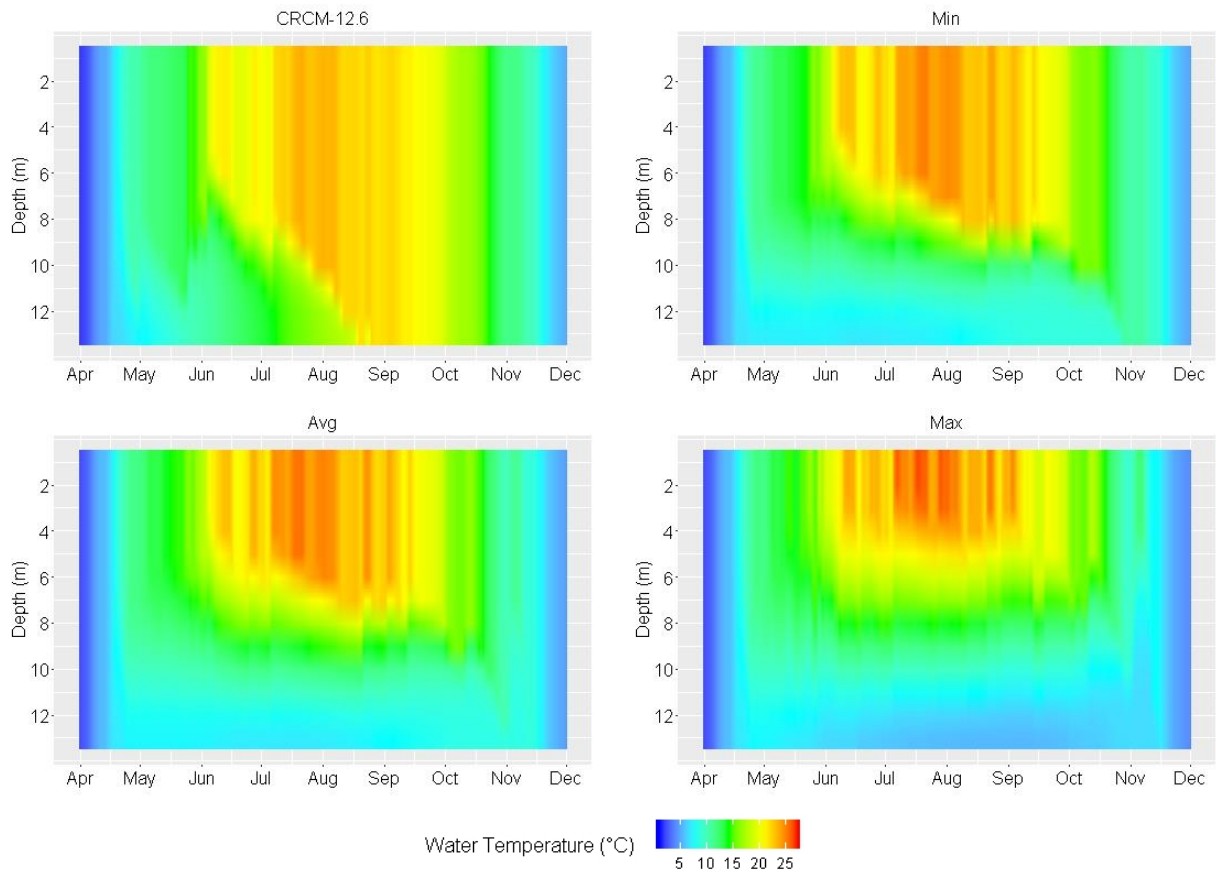

Fig. 13 Isotherms in open water period 2008 for CRCM-12.6 simulation is shown. Results for the lowest (Min), average (Avg), and the highest (Max) $K_d$ values are also shown.



Fig. 14 Monthly average temperature profile for CRCM-12.6, Min, Avg, and Max simulations in 2008.





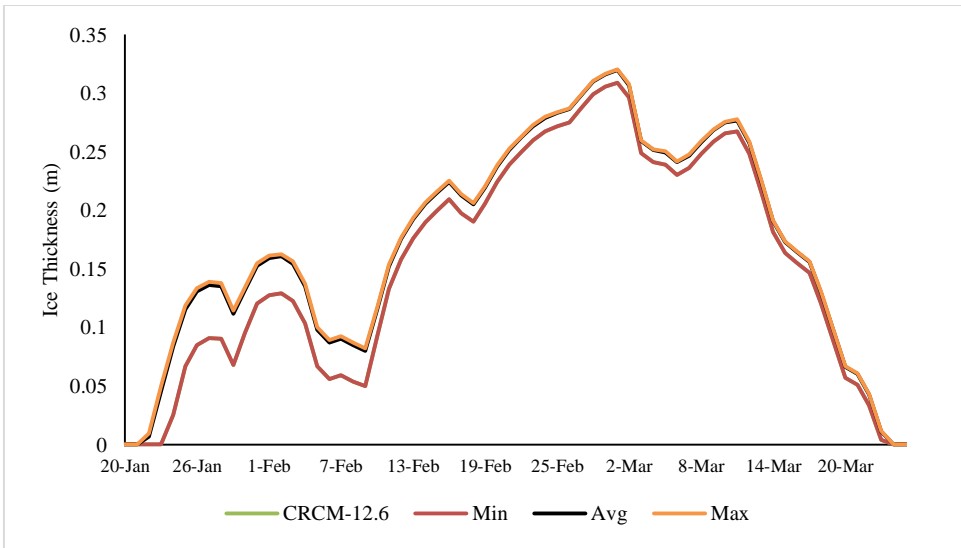

Fig. 15 Ice thickness during 2008 for CRCM-12.6 simulation and the lowest (Min), average (Avg), and the highest (Max) $K_d$ values are shown.

