# Peer review of "Satellite-Derived Light Extinction Coefficient and its Impact on Thermal Structure Simulations in a 1-D Lake Model"

_Hydrology and Earth System Sciences, 2016_

## Referee Comment (RC1) · Anonymous Referee #1 · 23 Mar 2016

Overall review of submitted paper:

This paper is about the lake water transparency (light extinction coefficient Kd) for the freshwater Lake Erie. Satellite-based lake water transparency values were compared with in-situ Secchi disk depths (SDD). Next, the 1D Flake model was run for several water transparency values and model results were compared with lake water surface temperature (LWST) measurements. It is a clearly written paper. I therefore recommend this paper for publication after minor revision. My remarks are summarized below:

Quality of model results (1):

The model results are compared with (Martynov, 2012) in which a light extinction coefficient Kd of 0.2 m-1 was used. This corresponds to a SSD of 8.5 m (see Eq. 2), which is not a very common value for SSD. Also, the Flake model results appear to be very sensitive for Kd values less than 0.5 m-1. Potes et al. (2012) used a Kd of 1.0 m-1 for clear water. Why didn't the authors choose the more common SSD value of Potes for a comparison with their model results? This would also have been more in line with the Kd for the NDBC station with a minimum value of 0.58 m 1 and an average value is 0.9 m-1 over the period of 2003 to 2012. It is not very difficult to improve the results of (Martynov, 2012) because a rather unrealistic Kd value of 0.2 m-1 was applied in that paper. A reference value of 1.0 m-1 of Potes would probably have resulted in comparable results.

Quality of model results (2):

In this paper only a comparison with LWST is conducted. As stated on page 2, this is 'one of the key variables' for modeling thermal structures in lake-atmosphere models. Why didn't the authors compare with other key variables, such as the thermal stratification? Are CTD-measurements available at buoy stations in Lake Erie? A comparison of computed isotherms with measured isotherms (cf. Fig. 13) may significantly improve the impact of this paper.

Issues of less importance:

- (page 7) Relation between Kd and SSD; The relation in Eq. (2) is applied. However, at the end of this page is stated that the extinction coefficient can be derived from the equation Kd =1.64 * SSSˆ(-0.76), which is a different one. This is confusing. Which equation is used?

- (page 9/Fig. 9) Flake model depth; It is confusing that two model depths (12.6 and 20 m) are applied. Is a depth of 12.6 m applied in the simulations with varying Kd values applied, because this is the actual depth? I suggest to remove all results for the 20 m depth simulations, also because the results are quite similar to CRCM-12.6.

- (Figures 5 to 7) Contour interval; The interval is between 0 and 5. As a result, the interesting range of approximately 0.5 to 1.5 is not clearly visible in these figures.

- (Fig. 9) Thickness of lines; In Figure 9 for 2007 the observations are not visible for September to December 2007. This is caused by the thickness of the lines. Please use another order of the shown time series so that the measurements become visible.

- (general remark) It is beyond the scope of this paper, but why is 1D modeling applied? With the current computing power of off-the-shelf computers, 3D modeling of lakes like Lake Erie is (easily) feasible. Then, for example, horizontal circulation and the non-equidistant bed level can be taken into account.

Please also note the supplement to this comment:
http://www.hydrol-earth-syst-sci-discuss.net/hess-2016-82/hess-2016-82-RC1-supplement.pdf

**Supplement:**

**Review of paper "Satellite-Derived Light Extinction Coefficient and its Impact on Thermal Structure Simulations in a 1-D Lake Model",**

**by Kiana Zolfaghari, Claude R. Duguay and Homa Kheyrollah Pour**

**Manuscript number: hess-2016-82**

**23 March 2016**

**Overall review of submitted paper:**
This paper is about the lake water transparency (light extinction coefficient $K_d$) for the freshwater Lake Erie. Satellite-based lake water transparency values were compared with in-situ Secchi disk depths (SDD). Next, the 1D Flake model was run for several water transparency values and model results were compared with lake water surface temperature (LWST) measurements. It is a clearly written paper. I therefore recommend this paper for publication after minor revision. My remarks are summarized below:

*Quality of model results (1)*
The model results are compared with (Martynov, 2012) in which a light extinction coefficient $K_d$ of 0.2 m$^{-1}$ was used. This corresponds to a SSD of 8.5 m (see Eq. 2), which is not a very common value for SSD. Also, the Flake model results appear to be very sensitive for $K_d$ values less than 0.5 m$^{-1}$. Potes et al. (2012) used a $K_d$ of 1.0 m$^{-1}$ for clear water. Why didn't the authors choose the more common SSD value of Potes for a comparison with their model results? This would also have been more in line with the $K_d$ for the NDBC station with a minimum value of 0.58 m$^{-1}$ and an average value is 0.9 m$^{-1}$ over the period of 2003 to 2012. It is not very difficult to improve the results of (Martynov, 2012) because a rather unrealistic $K_d$ value of 0.2 m$^{-1}$ was applied in that paper. A reference value of 1.0 m$^{-1}$ of Potes would probably have resulted in comparable results.

*Quality of model results (2)*
In this paper only a comparison with LWST is conducted. As stated on page 2, this is 'one of the key variables' for modeling thermal structures in lake-atmosphere models. Why didn't the authors compare with other key variables, such as the thermal stratification? Are CTD-measurements available at buoy stations in Lake Erie? A comparison of computed isotherms with measured isotherms (cf. Fig. 13) may significantly improve the impact of this paper.

Issues of less importance:
- (page 7) *Relation between $K_d$ and SSD;* The relation in Eq. (2) is applied. However, at the end of this page is stated that the extinction coefficient can be derived from the equation $K_d = 1.64 \times SDD^{-0.76}$. This is confusing. Which equation is used?
- (page 9/Fig. 9) *Flake model depth;* It is confusing that two model depths (12.6 and 20 m) are applied. Is a depth of 12.6 m applied in the simulations with varying $K_d$ values applied, because this is the actual depth? I suggest to remove all results for the 20 m depth simulations, also because the results are quite similar to CRCM-12.6.

- (Figures 5 to 7) *Contour interval;* The interval is between 0 and 5. As a result, the interesting range of approximately 0.5 to 1.5 is not clearly visible in these figures.
- (Fig. 9) *Thickness of lines;* In Figure 9 for 2007 the observations are not visible for September to December 2007. This is caused by the thickness of the lines. Please use another order of the shown time series so that the measurements become visible.
- (*general remark*) It is beyond the scope of this paper, but why is 1D modeling applied? With the current computing power of off-the-shelf computers, 3D modeling of lakes like Lake Erie is (easily) feasible. Then, for example, horizontal circulation and the non-equidistant bed level can be taken into account.

---

## Referee Comment (RC2) · Anonymous Referee #2 · 25 Mar 2016

Reviewer comments to Satellite-Derived Light Extinction Coefficient and its Impact on Thermal Structure Simulations in a 1-D Lake Model by Zolfaghari et al. 2016

General comments

This manuscript deals with water optical properties which are acquired by remote sensing, and which are then used as input to 1-D lake modeling. This approach is important and needed addition to current efforts of incorporating lakes and reservoirs to weather prediction models. Water clarity is an important factor in defining lake heat budget and thermal stratification and thus is a significant parameter for processes in the air-water interface. With millions lakes of different sizes around the world, comprehensive direct measurements of water clarity are not possible, highlighting the need for indi-

rect estimates of water optical properties. Satellite measurements show great promise in mapping light extinction coefficient, a parameter to define water clarity, and to my knowledge this study is the first one to incorporate lake modeling to water clarity defined from satellite observations. This is an important study with wide interest in scientist from different fields and therefore I find this study appropriate for HESS. However, there are a few major points which prevents me from recommending this manuscript for publication as it is.

The major topic of this manuscript is that satellite-derived light extinction coefficient, Kd, represents well the in situ measurements of Kd, that it can be used as an input to lake modeling, and that it enhances the performance of FLake as compared to the current approach of using constant Kd of 0.2 m-1. These points should be emphasized. Currently, the manuscript seems unbalanced, with much of the focus given to topics not strictly the main theme of this manuscript. Also some restructuring is needed so that the reader does not get distracted from the main focus. In general, the manuscript would benefit from reducing the amount of figures.

Since this is the first approach in combining satellite-derived Kd to lake modeling, it would be of value to describe the strengths and the weaknesses of this approach. How easy and accurate method this is for the modeling community in general; can this be used without in situ measurements of Kd or should there always be e.g. Secchi disk measurements for validation; what are the next steps needed for applying this method in broader context.

Specific comments

In regards of restructuring, I will give here an example how the figures (and related discussion) could be rearranged. After the map, I suggest to first show satellite-derived Kd at the site of FLake modeling (current Fig. 8), which is the main input parameter under focus here. For the estimated solar irradiance and incoming long-wave radiation (Figs. 2 and 3), the figures add only little to the reported statistics and thus these two figures

could be removed. After showing the satellite-derived Kd, it would be logical to show their validation (current Fig. 4). Then the results of models against measurements, i.e. current Figs. 9 and 10. Current Figs. 11, 12, 13 and 14 basically show the same data in different forms. I suggest either to combine Figs. 11 and 12 and show only this, or show only Fig 13. Lastly, current Figs. 5 and 6 could be shown, which would then lead to the discussion of the strengths of satellite-derived Kd and possible future studies (see also the related comment later). This is only a suggestion for restructuring, it could also be done otherwise.

Page 4, line 9. Air humidity, which is used as FLake input, is taken from land 10 km away from the lake site. Air humidity is important for modelled latent heat fluxes. Could the authors briefly state their opinion how well the measured air humidity represent that over lake, and how or if this affects the modeled results.

Page 4, lines 19-21. Sentences starting with "Available Secchi disk" and "SDD data was" are repetitive and should be merged to one sentence. Also, it could be mentioned here that the SDD data comes from Limnos cruises.

Page 6, lines 1-2. Please clarify this sentence. Does this basically mean that only the pixels which were not rejected according to the criteria in Table 1 were used?

Page 7, Chapter 3.1.1. A lot of space is dedicated for this, and therefore a justification could be given in the first sentence. E.g. "Validating the satellite-derived Kd with in situ observations is important because. . ." And in the end of the chapter the outcome of the evaluation, e.g. "For these reasons, we deem the satellite-derived Kd correct and thus were confident in using them in the modelling." Also, in Chapter 3.1.1. or later, the authors could discuss whether this kind of validation is always needed with satellite observations, what are the implications, etc.

Page 8, Chapter 3.1.2. This chapter seems interesting but out of place. These results are not further elaborated, and therefore I suggest to move them to the end of the Discussion. This way the authors could show what benefits remote sensing of Kd would

bring (spatial and temporal variability, which is not achieved well with manual sampling; perhaps good input for 2D and 3D modeling), which would lead to the discussion of possible next studies. This way also the key input parameter, Kd at the NDBC station, would be shown earlier.

Page 9, Chapter 3.2.1. If the satellite-derived Kd has been validated sufficiently well and it produces better simulations, what would be needed for the simulations to match the measured LSWT more accurately? This would lead to suggestions for future re-search.

Page 9, Paragraph starting 'Fig. 9 shows the results...'. I suggest to first describe the observed behavior in the temperatures and then discuss how the modelled behaviors compare to these.

Page 10, lines 17-18. This is quite strong statement and probably not true for all lakes. Lakes are very heterogeneous, be more specific which type of lakes is meant here.

Page 10, Chapter 3.2.2. This chapter needs the most restructuring. E.g. the paragraph on page 11, lines 25-28, could be removed. The two first sentences are basic limnological knowledge and the last sentence does not really lead the story further. In this chapter, the theme of light penetration and absorption is discussed in many places, e.g. on page 11, lines 8-9, lines 11-12 and lines 29-34, page 12, lines 11-13 and lines 19-20. Remove excess repetition. The last paragraph on page 12 (starting Fig. 14 depicts...) repeats what is said earlier and is not the main focus of this study, therefore I suggest to remove that paragraph. The last paragraph of Chapter 3.2.2. discusses about modeled ice cover. This seems a bit out of scope and there really seems to be no ice measurements against which to validate modeling. For this reason, I suggest to either remove this paragraph or significantly shorten it.

Page 11, lines 11-12. The authors seem to mix two concepts here. Darker water color is related to dissolved substances, such as colored dissolved organic matter, not to particulate matter.

Page 11, lines 13-14. The authors over-simplify the underlying mechanisms for LSWT behavior. The loss of energy to the atmosphere is related to the surface water temperature (and wind), not only in fall but throughout the open-water season. However, the mechanism how mixed layer depth affects the rate of heat loss needs more explaining.

Page 11, lines 18-23. Tie these results from the literature more tightly to the findings in this study, e.g. by writing whether this study supports or opposes previous findings. Also, the sentence on lines 21-23 (starting 'Heiskanen et al...') could be removed either from here or from the Summary.

Page 12, paragraph starting with 'Fig. 12 shows'. Here full mixing is described in very atypical way on several occasions, e.g. by 'highest depth of mixing' and 'reaches maximum MLD'. I suggest to describe these occasions either by discussing of overturn, of full mixing or similar.

Page 12, lines 8-9. If this is the reason for earlier overturn in simulations with clearer water, how the authors then explain the results shown in Figs. 11 and 13 where it is evident that there is full mixing in the beginning of September in CRCM-12.6 simulation with temperatures of about 20 deg C? Fig. 14 also shows that the clearer the water, the higher the water temperatures in Oct and Nov. Note that in addition to convection, mixing is related to wind forcing and density gradient in the water column.

Page 12, lines 16-17. MLD is not influenced by the thermal structure, but it is part of the thermal structure. I would remove this sentence.

Page 12, lines 13-14. Fig. 13 is essentially the same data as in Fig. 12 and therefore one cannot be used to confirm the results of the other.

Page 12, lines 20-22. Deepening of the thermocline is related e.g. to wind forcing and thus it cannot be suggested that thermocline deepening in clear waters is monotonic. Also, it is not clear what is meant with 'stabilize the temperatures'. I suggest to remove this sentence.

Technical corrections

Page 6, line 23. The same result 'was' found for...

Page 8, line 32. 'leads to higher water clarity'. The authors must mean lower water clarity.

Page 9, line 18. The sentence starting 'The Kd values' and the sentence after that could be merged and rephrased. E.g. 'The monthly-averaged Kd were used to simulate the surface water temperature and produce a merged LSWT (Merged)."

Page 9, line 20. Comparing LSWT in situ observations (Obs) with...

Page 9, line 21. How can the authors compare measured and modelled surface temperatures in April when there seems to be little or no measured LSWT during April, at least according to Fig 9?

Page 10, lines 2-3. Rephrase. Do the authors mean that the annual average of Kd can occasionally be closer to the actual Kd than the monthly-averaged Kd? This same topic is also mentioned in lines 16-17, and at least to me it is unclear how yearly average value (i.e. one single number) can represent the extent of Kd variations (i.e. how big is the range).

Page 10, line 10. Please clarify what is specifically meant with 'are as affected'.

Page 10, lines 11-12. 'This can be explained by...'. Be more specific in telling how lake depth explains this.

Page 10, lines 12-13. This should be self-evident if the model is any good, and therefore I suggest to remove this sentence.

Page 11, line 9. '... causing thinner mixing depth (Fig. 12)'

Page 11, line 35. Change 'when Kd changes...' e.g. to 'when maximum (minimum) Kd is used instead of its average value...'

Page 12, line 1. Similar comment as previous. This is a bit misleading wording since it gives the idea that Kd changes naturally, whereas what is meant that different Kd is used as an input.

Page 13, line 27. Write open the abbreviation 'LSWT' here. It is not typical abbreviation and not clear for those who only read Summary and conclusions.

Page 14, line 3. Change 'has' to 'have'.

Comments to figures

Fig. 1. It would be of interest to see the main river inlets and outlets. This way it would be easier to assess how much river inflow possibly affects modeling results.

Fig. 5. Remove 'Lake Erie boundary' from the legend, it is not needed. Also make the color bar much larger. Same comments for other similar figures.

Fig. 8. It would be of interest to see the SDD at this location (or from the nearest location where those exist) together with these CC-derived Kd. These could be marked to the same graph with secondary y-axis.

Fig. 10. It would be interesting to see the performance for each year separately. This could shown by plotting each year with different color. Also, it is more standard to show these kind of scatter plots as box plots (both axes of same length).

Fig. 11. The measured LSWT should be shown. Otherwise, it is impossible to say which simulation performs the best. Use a) and b) for these two graphs. Also in the legend, the Kd values could be shown for each model run.

Fig. 15. Model run CRCM-12.6. is not visible. If the resolution can not be increased, describe in the caption where the line is.

[Figure]

---

## Referee Comment (RC3) · Anonymous Referee #3 · 3 Apr 2016

General comments

The manuscript presents a study on the use of light extinction coefficient values derived from MERIS satellite imagery in the FLake 1-D lake model. FLake is the most widely used lake scheme in numerical weather prediction (NWP) models. To take advantage of the coupling of lake schemes to NWP and climate models it is necessary to have data on the lakes transparency. As they are not observations in an in-situ operational way, the most promising strategy is to use satellite images. Therefore this study deals with a current scientific issue, that really fit in the HESS scope. As far as I my English allow, the manuscript is well written. The study is original and contains new results which are worth to be published. In my opinion the manuscript requires major revision

before being accepted. Please, consider the following comments, with different levels of importance:

Detailed comments:

pag. 2 line 3 land → continental

line 12: In the first mention to the FLake model, a reference to the the model may be given. This: Mironov, D. V., 2008: Parametrization of lakes in numerical weather prediction. Description of a lake model. COSMO Technical Report, No. 11, Deutscher Wetterdienst, Offenbach am Main, Germany, 41 pp. or this: Mironov, D., E. Heise, E. Kourzeneva, B. Ritter, N. Schneider, and A. Terzhevik, 2010: Implementation of the lake parametrisation scheme FLake into the numerical weather prediction model COSMO. Boreal Env. Res., 15, 218-230.

line 13 "artificially limited to 40-60 m depth". Is not artificial. Flake is not able to simulated deep lakes, as it consider only two layers. To lakes deeper than 40 meters it will be necessary to consider one third layer below the termocline.

page4 line 3. Only 1 station for all the lake?

page 5 line 10: T at which station, at which level?

line 15 – Why not use data from analysis (From ECMWF, for example)

Section 2.3 In my opinion, this section should identify the time periods in which Kd MERIS images were available. This information exists in a dispersed form in section 3.1.2..

line 26 – 31. It is not clear which Kd is used: in a spectral band (which ?), broadband ?

page 6 Section 2.4: In this section, the set-up of the simulations should be clearly presented, namely: - How were the FLake prognostic variables initialized? - The model integration period (start and end) - The temporal resolution of the forcing and the time step of the simulations.

lines 14/11 In this paragraph the ice parametrization used should also be introduced as it was activated. (maybe also the snow scheme).

line 18: what means "The setup conditions"?

line 19/20- depth of the water temperature measurements: why is it included here? The water temperature is not a forcing parameter...

line 21: why use the local depth (12.6 m) and not an averaged depth, maybe of the western basin? line 21: "to configure" means force initialize, or both?

page 7 line 11 (Eq. 2) and line 30 – Equation I don't understand the process to adjust a relation between Kd and SDD. The equation Eq 2 indicate an inverse proportionality, but the expression obtained is not linear..

page 8 line 8: "can explain"? or can detect?

line16: Kd = 0.87 m-1: is a satellite-derived value or was in-situ measured? By the way, are there any in situ measurements on the selected day?

line 18: "on a monthly-basis for one year" but only four months were considered.

Figure 6 shows only particular days and not average values, but the text refers to monthly average values. This question is valid also for the discussion of the results presented in Figure 7. It should be indicated how the averaged values were calculated.

Figures 5, 6 and 7. With the chosen color scale, most of the field is in the same color. I think it would be better to use a color scale with higher resolution especially in lower values (could be a non-linear scale, possibly logarithmic). Also, the color scale should have a more detailed legend.

line 21/22 and 30: the values of Kd are satellite-derived values or were measured in-situ?

line 23 "full coverage of Lake Erie were only available in May of two consecutive years

(2008 and 2009)", but figure 6 show a map for May 2010. Contradiction?

page 9 Neither in the section 2.4, nor here, the start of the simulations were indicated. line 18 and 20: Which depth were used in the Avg and Merged simulations?

In the Figure 9, results for 2007 there are plotted values for Kd for fall. Why are this values not shown in table 2 and not used in the merged simularion?

Analysis of figure 9: Some explanation for the strange spike that occurred in mid-September 2006 in the avg simulation? A less pronounced effect also occurred in midpage 10 line 3. It seems strange to defend that year average can be more representative of Kd variations than monthly averages...

line 5 "Turbid waters in these months simulate colder LSWT". this statement must be explained. In my opinion, the reason is not on the fact that during those months the waters are turbid, but because the water was more turbid before, during spring and summer, reducing the heating of deep water. This should be discussed further, in particular by analyzing the evolution of deep temperature and column mean temperature (Flake variables)

line 11 "This can be explained" → "this is due to".

line12 The conclusion that "a realistic lake depth and Kd value will improve model results" is obviously correct, but, specially concerning the depth, could not be demonstrated using only the two depth values considered.

Figure 10 a and b: An hypothesis: If different colours were used for spring and summer-fall it may be possible to see and than discuss two different behaviours

Caption of figure 10: the means of a,b,c,d should be indicated

line 16/17 I do not agree with:"It is possible that the extent of Kd variations is best represented by the yearly average value". Maybe the problem is that the errors in the

determination of monthly mean values may result in a worse simulation...

line 24-26: I would be less categorical, adding for example at the beginning of the statement something like: "In the absence of reliable values of the temporal evolution of Kd, ….."

Considering that 12.6 m is the realistic lake depth must be better justified. see my comment (page 6 line 21)

line 33: The sensitivity of FLake to LSWT, MWCT, MLD, isotherm, ice phenology and thickness???

page 11 lines 2/4: which depth were used in the sensitivity imulations?

lines 4/5 More than the depth, what is important is to indicate the value of Kd in the RCM-12.6 simulation.

line 7 (maximum or Max) → Max will be enough.

line 8 The world faster does not seem the most appropriate in: "solar radiation is absorbed faster in turbid waters".What happens is that the radiation is more absorbed in the water surface layer, as explained by the authors afterward.

line 9 "This shallow layer exchanges heat faster with the atmosphere", is correct but should be explained, In my opinion the main reason has to do with the fact that the as the surface water temperature is higher the sensible and latent heat fluxes increase.

line 12/13 "However, in fall the loss of energy to the atmosphere is also faster due to the shallow mixing depth" This will not be the main reason. In my opinion the main reason has to do with the fact that the deep (and the mean) water temperature is lower.

line 14: "least turbid water" The use of the world "least" here can be confusing, as it is less turbid than what considered in the CRCM-12.6 simulation

line 15: "Min" is enough.

line 18: "FLake is not significantly sensitive to LSWT" It is not correct in terms of English

line 25: Please delete the sentence:"For both clear and dark waters, LSWT is warmer than the MWCT, due to being exposed to more intense solar radiation.". The reason is the density! (for water temperatures over 4°C)

lines 25/28. In this discussion it will be interesting to compare also with the FLake deep temperature.

page 12 line4 "two turnover". In my opinion the first period without stratification should not be identified as a turnover. As I can imagine, the Flake were initialized with a constant temperature profile.

line 8/9 "As a result, the water column in clear water reaches the temperature of maximum density (4°C) much faster than turbid water ..."?? is not what we can see in Figure 11 (bottom) and in Figure 14!

lines 10 / l5 The average values over the whole period does not seem to be relevant in this discussion.

line 10 "In the turbid" → "In the more turbid"

line 13 "distribute to" "be absorbed in" or "distribute energy to"

line 16: We can not say that"The MLD is influenced by the water column thermal structure". The MLD is itself a parameter used by Flake to characterize the water thermal structure...

Caption of figure 13. Please, improve the wording... The 4 individual figures should have the same caption.

line 19: "but also warmer" is not valid for the whole period. I think it will be more correct to say something like: "warmer in spring and summer, and colder in fall"

line 19/ 20: the sentences: "The reason is that solar radiation is mostly absorbed at the

upper layers in turbid water. Thus, the radiation is used to warm up a thinner layer in dark waters leading to higher temperatures." are correct but the argument is repeated some times on the text. In my opinion, it will be better to explain in a more integrated way, based on physics of course, the differences between clear and turbid waters.

line 21: "shows that the deepening of the thermocline layer in clear waters is monotonic". I can not see this. Can you be more precise.

line 29: before the "increased effect of cooling from the layers below" it should be noted that as the surface temperature of the turbid lakes is higher, the radiative losses to the atmosphere are greater. So, during the heating period, a turbid lake as a whole, loses more energy by radiation and therefore stores less energy..

page 13 (before Summary and Conclusions) It is difficult to analyze the discussion contained in this page without knowing the details about the initialization of the simulations. And about observations? When occurred the break-up and the freeze-up?

line 13. "Dark waters store more heat in a shallower depth." The sentence may be misunderstood. First consider change "depth" by "layer". But if one consider the whole water column, dark waters store less heat. "Therefore, in the winter time". In my opinion the "in the winter time" should be deleted, as this is also valid in summer and autumn (and may be more important during these seasons)

page 16 line 21. Arkady Terzhevik should be added to the list of co-authors

---

## Author Comment (AC1) · 21 May 2016

**We would like to thank all the reviewers for their constructive comments which helped improve the manuscript. Our replies to comments are covered below.**

Overall review of submitted paper: This paper is about the lake water transparency (light extinction coefficient Kd) for the freshwater Lake Erie. Satellite-based lake water transparency values were compared with in-situ Secchi disk depths (SDD). Next, the 1D Flake model was run for several water transparency values and model results were compared with lake water surface temperature (LWST) measurements. It is a clearly written paper. I therefore recommend this paper for publication after minor revision. My remarks are summarized below:

Quality of model results (1):

The model results are compared with (Martynov, 2012) in which a light extinction coefficient Kd of 0.2 m-1 was used. This corresponds to a SSD of 8.5 m (see Eq. 2), which is not a very common value for SSD. Also, the Flake model results appear to be very sensitive for Kd values less than 0.5 m-1. Potes et al. (2012) used a Kd of 1.0 m-1 for clear water. Why didn't the authors choose the more common SSD value of Potes for a comparison with their model results? This would also have been more in line with the Kd for the NDBC station with a minimum value of 0.58 m 1 and an average value is 0.9 m-1 over the period of 2003 to 2012. It is not very difficult to improve the results of (Martynov, 2012) because a rather unrealistic Kd value of 0.2 m-1 was applied in that paper. A reference value of 1.0 m-1 of Potes would probably have resulted in comparable results.

We agree that it is not very difficult to improve Martynov et al. (2012) using realistic Kd value; however, the challenge would be to extract this realistic value, which is one of the main point of our manuscript. Using Martynov et al. (2012) was very useful for us to use as they applied FLake model on Lake Erie and the same station of our study (NDBC station). Therefore results of that study are compared to ours to show how coupling satellite observation with lake modeling can improve results rather than using a generic constant value for Lake Erie NDBC station.

Quality of model results (2):

In this paper only a comparison with LWST is conducted. As stated on page 2, this is 'one of the key variables' for modeling thermal structures in lake-atmosphere models. Why didn't the authors compare with other key variables, such as the thermal stratification? Are CTD-measurements available at buoy stations in Lake Erie? A comparison of computed isotherms with measured isotherms (cf. Fig. 13) may significantly improve the impact of this paper.

We agree with the reviewer's comment but unfortunately CTD measurements for NDBC station were not available for Lake Erie.

Issues of less importance:

1- (page 7) Relation between Kd and SSD; The relation in Eq. (2) is applied. However, at the end of this page is stated that the extinction coefficient can be derived from the equation Kd =1.64 * SSS^(-0.76), which is a different one. This is confusing. Which equation is used?

Thanks for your comment. Equation $K_d = 1.64 \times SDD^{-0.76}$ is the reformed version of equation $SDD \times K_d = K$ (a general format of relationship between SDD and K$_d$), where the constant value of $K$ is calculated and replaced.

2- (page 9/Fig. 9) Flake model depth; It is confusing that two model depths (12.6 and 20 m) are applied. Is a depth of 12.6 m applied in the simulations with varying Kd values applied, because this is the actual depth? I suggest to remove all results for the 20 m depth simulations, also because the results are quite similar to CRCM-12.6.

Thanks for your suggestion. The purpose of using 20m depth was comparing our results with the previous study and find out if results can be improved on Lake Erie by keeping everything constant with updating only Kd values. Therefore tile depth of 20 m was used in simulations to exactly reproduce the simulations of Martynov et al. 2012. We also compared results of CRCM-12.6 and CRCM-20 simulations to demonstrate the effect of depth on reproducing lake parameters.

3- (Figures 5 to 7) Contour interval; The interval is between 0 and 5. As a result, the interesting range of approximately 0.5 to 1.5 is not clearly visible in these figures.

Thanks for suggesting this. The corrections have been made in the new version of manuscript.

4- (Fig. 9) Thickness of lines; In Figure 9 for 2007 the observations are not visible for September to December 2007. This is caused by the thickness of the lines. Please use another order of the shown time series so that the measurements become visible.

The observations in the period of Sep-Dec 2007 were not available after August.

5- (general remark) It is beyond the scope of this paper, but why is 1D modeling applied? With the current computing power of off-the-shelf computers, 3D modeling of lakes like Lake Erie is (easily) feasible. Then, for example, horizontal circulation and the non-equidistant bed level can be taken into account. Please also note the supplement to this comment:
http://www.hydrol-earth-syst-sci-discuss.net/hess-2016-82/hess-2016-82-RC1supplement.pdf

The 1D FLake Model is commonly used in the forecasting models and also applied on Great Lakes (Martynov A., L. Sushama and R. Laprise (2010), Simulation of temperate freezing lakes by onedimensional lake models: performance assessment for interactive coupling with regional climate models Boreal Env. Res. 15 143–164).

More complex 3-D lake models are now starting to be used to reproduce large lake properties. For example, Environment Canada has recently implemented a fully coupled 3-D atmosphere-lake modelling system to represent the complex air-lake interaction over the Great Lakes region (Dupont, F., Chittibabu, P., Fortin, V., Rao, Y. R., and Lu, Y. 2012. Assessment of a NEMO-based hydrodynamic modelling system for the Great Lakes. Water Quality Research Journal of Canada, 47, 198–214.).

The contribution of satellite-derived water clarity in improving simulations with more complex 3-D lake models such as NEMO could form the main body of a follow-up paper.

---

## Author Comment (AC2) · 23 May 2016

**We would like to thank all the reviewers for their constructive comments which helped improve the manuscript. Our replies to comments are covered below.**

General comments:

This manuscript deals with water optical properties which are acquired by remote sensing, and which are then used as input to 1-D lake modeling. This approach is important and needed addition to current efforts of incorporating lakes and reservoirs to weather prediction models. Water clarity is an important factor in defining lake heat budget and thermal stratification and thus is a significant parameter for processes in the air-water interface. With millions lakes of different sizes around the world, comprehensive direct measurements of water clarity are not possible, highlighting the need for indirect estimates of water optical properties. Satellite measurements show great promise in mapping light extinction coefficient, a parameter to define water clarity, and to my knowledge this study is the first one to incorporate lake modeling to water clarity defined from satellite observations. This is an important study with wide interest in scientist from different fields and therefore I find this study appropriate for HESS. However, there are a few major points which prevents me from recommending this manuscript for publication as it is.

The major topic of this manuscript is that satellite-derived light extinction coefficient, Kd, represents well the in situ measurements of Kd, that it can be used as an input to lake modeling, and that it enhances the performance of FLake as compared to the current approach of using constant Kd of 0.2 m-1. These points should be emphasized. Currently, the manuscript seems unbalanced, with much of the focus given to topics not strictly the main theme of this manuscript. Also some restructuring is needed so that the reader does not get distracted from the main focus. In general, the manuscript would benefit from reducing the amount of figures.

Since this is the first approach in combining satellite-derived Kd to lake modeling, it would be of value to describe the strengths and the weaknesses of this approach. How easy and accurate method this is for the modeling community in general; can this be used without in situ measurements of Kd or should there always be e.g. Secchi disk measurements for validation; what are the next steps needed for applying this method in broader context.

Thanks for your comment. Strength: Integrating lake specific Kd values can improve the performance of 1-D lake models. However, field measurements of Kd are not widely available. This study demonstrates that satellite observations are a reliable data source to provide lake models with global estimates of Kd with high spatial and temporal resolutions.

The globally available CC product can be easily used as a source to fill the gaps in Kd in situ observations, and improve the performance of parameterization schemes and, as a result, further improve the NWP and climate models.

Weakness: in situ data of SDD or water clarity are always required for the method development, calibration, and validation.

Next steps: investigate the potential of Sentinel-3 to provide lake modeling community with the water clarity information. Also study the resulted improvement in the performance of NWP and climate models.

These points are mentioned in page 14, lines 5-11.

Specific comments

In regards of restructuring, I will give here an example how the figures (and related discussion) could be rearranged. After the map, I suggest to first show satellite-derived Kd at the site of FLake modeling (current Fig. 8), which is the main input parameter under focus here. For the estimated solar irradiance and incoming long-wave radiation (Figs.2 and 3), the figures add only little to the reported statistics and thus these two figures could be removed. After showing the satellite-derived Kd, it would be logical to show their validation (current Fig. 4). Then the results of models against measurements, i.e. current Figs. 9 and 10. Current Figs. 11, 12, 13 and 14 basically show the same data in different forms. I suggest either to combine Figs. 11 and 12 and show only this, or show only Fig 13. Lastly, current Figs. 5 and 6 could be shown, which would then lead to the discussion of the strengths of satellite-derived Kd and possible future studies (see also the related comment later). This is only a suggestion for restructuring, it could also be done otherwise.

Thanks for the suggestion. The authors, however, would like to keep the current format. This study is based on using satellite-derived shortwave radiation. Longwave radiation is also estimated. Therefore these two parameters need to be evaluated (Figures 2 and 3). Other input of FLake model, the most important one, is water clarity which is also derived from satellite observations. Before any further discussing the potential of combining these observations into the model, first the evaluation was conducted (Figure 4). After that, Figures 5-7 show the extend of spatial and temporal variations of water clarity for the entire water body. Finally, Figure 8 shows how water clarity has changed in the station of interest (NDBC) during the study period. Based on the variations (min, max, average values of Kd at this station), different simulations were designed.

Figure 9 shows how observations and different simulations compare during the study period. The figure demonstrate if there is any specific timing that the difference between simulations and observations is more highlighted; whereas Figure 10 is for statistical evaluation purpose.

Figure 11-14 might have overlaps in the demonstrated information; however, they are presented in this study for different purposes. Figure 11 show LSWT and MWCT; Figure 12: MLD; Figure 13: timing, depth, and temperature of different thermal layers (epilimnion, thermocline) also temperature of MLD; Figure 14: average temperature and depth of each thermal layer.

Page 4, line 9. Air humidity, which is used as FLake input, is taken from land 10 km away from the lake site. Air humidity is important for modelled latent heat fluxes. Could the authors briefly state their opinion how well the measured air humidity represent that over lake, and how or if this affects the modeled results.

As the reviewer mentioned, the humidity is important for modelled latent hear flux and would be different over land and over lake. Since warm air hold more moisture than cold air, the percentage of humidity must change with change in air temperature. We expect that the humidity decreases as temperature increases. Large temperature and humidly differences can lead to a large sensible and latent heat fluxes. Water is a

good absorber of the energy but the land absorb much faster the energy from the sun. Water heats up much slowly than land and therefore, the air above land will have higher temperature and therefore less humidity. On the other hand, lack of in-situ data over lakes and the distance of the stations from the shoreline (less than 81km in our case, 10 km was recognized as a mistake in the manuscript) is one of the main limitation of lake studies. In this study all the model forcing comes from the station on land such as air temperature and humidity, therefore in this way the rate of differences between air temperature and humidity kept constant.

Page 4, lines 19-21. Sentences starting with "Available Secchi disk" and "SDD data was" are repetitive and should be merged to one sentence. Also, it could be mentioned here that the SDD data comes from Limnos cruises.

Thanks for your comment. Page 4 line 20 ("Available Secchi disk depth (SDD) field measurements were used to estimate lake water clarity.") has been removed.

Page 6, lines 1-2. Please clarify this sentence. Does this basically mean that only the pixels which were not rejected according to the criteria in Table 1 were used?

Yes.

Page 7, Chapter 3.1.1. A lot of space is dedicated for this, and therefore a justification could be given in the first sentence. E.g. "Validating the satellite-derived Kd with in situ observations is important because. . ." And in the end of the chapter the outcome of the evaluation, e.g. "For these reasons, we deem the satellite-derived Kd correct and thus were confident in using them in the modelling." Also, in Chapter 3.1.1. or later, the authors could discuss whether this kind of validation is always needed with satellite observations, what are the implications, etc.

In the first sentence of this section, line 9, the main focus of that section is mentioned: the reliability of satellite-derived Kd values is highly dependent on comparison of them with independent in situ SDD measurements. So in situ observations are always required for validation of satellite-derived data.

End of this section is closed with the reason and motivation of using satellite-based water clarity measurements, when in situ SDD data are not always describing Kd values (only small values of Kd are described using SDD).

Page 8, Chapter 3.1.2. This chapter seems interesting but out of place. These results are not further elaborated, and therefore I suggest to move them to the end of the Discussion. This way the authors could show what benefits remote sensing of Kd would bring (spatial and temporal variability, which is not achieved well with manual sampling; perhaps good input for 2D and 3D modeling), which would lead to the discussion of possible next studies. This way also the key input parameter, Kd at the NDBC station, would be shown earlier.

Thanks for the suggestion. This section described how Kd value changes spatially and temporally over the full lake; and it ends with the variations of Kd in the location of NDBC station. This is to demonstrate how variant Kd value could be over the lake and over a period of time, demonstrating the shortage of in situ observations to cover these changes temporally and spatially and highlighting the motivation of using remote sensing observations to overcome these concerns. After highlighting this important role that remote sensing observations can play in coupling with lake models, the study continued showing the results of

integrating satellite-derived water clarity with FLake. Therefore the authors would prefer to keep the current format of the manuscript.

Page 9, Chapter 3.2.1. If the satellite-derived Kd has been validated sufficiently well and it produces better simulations, what would be needed for the simulations to match the measured LSWT more accurately? This could lead to suggestions for future research.

Thanks for the suggestion. Section 3.2.1 discuss the improvement of modeling using the satellite-derived Kd values. The next section study the sensitivity of FLake to the variations of Kd, and if it is necessary to consider the temporal variations (monthly basis) of Kd in simulation or a constant-lake specific value is sufficient in the modeling for Lake Erie.

Therefore if this comment is suggesting to consider the temporal variations of Kd in simulations, this has been already considered and tested for the range of Kd values in Lake Erie.

Page 9, Paragraph starting 'Fig. 9 shows the results. . .'. I suggest to first describe the observed behavior in the temperatures and then discuss how the modelled behaviors compare to these.

The temperature changes in three years, 2005-2007, have a normal fluctuation, increasing from spring to summer and decreasing toward winter. Therefore, the authors did not find it necessary to add to the manuscript.

Page 10, lines 17-18. This is quite strong statement and probably not true for all lakes. Lakes are very heterogeneous, be more specific which type of lakes is meant here.

These sentences are for explaining why results of two simulations of Avg and Merged are comparable, while Avg simulation are producing lower MBE. The statement starts with "it is possible". Therefore it is only a potential reason for such results in Lake Erie, and not a generalized rule for all lakes. However, modification to the sentence has been applied to clearly demonstrate the point.

Page 10, Chapter 3.2.2. This chapter needs the most restructuring. E.g. the paragraph on page 11, lines 25-28, could be removed. The two first sentences are basic limnological knowledge and the last sentence does not really lead the story further. In this chapter, the theme of light penetration and absorption is discussed in many places, e.g. on page 11, lines 8-9, lines 11-12 and lines 29-34, page 12, lines 11-13 and lines 19-20. Remove excess repetition. The last paragraph on page 12 (starting Fig. 14 depicts. . .) repeats what is said earlier and is not the main focus of this study, therefore I suggest to remove that paragraph. The last paragraph of Chapter 3.2.2. discusses about modeled ice cover. This seems a bit out of scope and there really seems to be no ice measurements against which to validate modeling. For this reason, I suggest to either remove this paragraph or significantly shorten it.

Thanks for your comment. Although in situ measurements are not available; however, the sensitivity of FLake to Kd variations to reproduce ice phenology and thickness is investigated and is one of the scopes of this paper.

The repetition has been removed in the new version of manuscript.

Page 11, lines 11-12. The authors seem to mix two concepts here. Darker water color is related to dissolved substances, such as colored dissolved organic matter, not to particulate matter.

Attenuation of light in dark waters is high; and this could be because of the existence of dissolved (absorption) or suspended matters (scattering).

Page 11, lines 13-14. The authors over-simplify the underlying mechanisms for LSWT behavior. The loss of energy to the atmosphere is related to the surface water temperature (and wind), not only in fall but throughout the open-water season. However, the mechanism how mixed layer depth affects the rate of heat loss needs more explaining.

The mixed layer depth (MLD) affects the speed of losing energy to the atmosphere throughout the year. But in these sentences the reason of faster loss of energy in fall is explained. This is because MLD in fall is shallower than in spring, therefor loss of energy is also faster.

Considering MLD to explain the reason is combining the effect of both temperature and wind.

Page 11, lines 18-23. Tie these results from the literature more tightly to the findings in this study, e.g. by writing whether this study supports or opposes previous findings. Also, the sentence on lines 21-23 (starting 'Heiskanen et al. . .') could be removed either from here or from the Summary.

In line 18, the result of our study is that the sensitivity of the model increases from Min to CRCM-12.6 simulation (Kd decreasing from 0.58 to 0.2). The statements after this lines (lines 19-23) are discussing other studies which support the finding of our study. This finding is that FLake is more sensitive to Kd values less than 0.5.

Page 12, paragraph starting with 'Fig. 12 shows'. Here full mixing is described in very atypical way on several occasions, e.g. by 'highest depth of mixing' and 'reaches maximum MLD'. I suggest to describe these occasions either by discussing of overturn, of full mixing or similar.

Lake turnover is the process of lake's water turning over from top to the bottom, which is full mixing. The maximum/highest possible depth of mixing at NDBC station is 12 meter, so when MLD reaches this depth turnover happens. This is the reason for describing turnovers using terms such as 'highest depth of mixing' and 'reaches maximum MLD'.

Page 12, lines 8-9. If this is the reason for earlier overturn in simulations with clearer water, how the authors then explain the results shown in Figs. 11 and 13 where it is evident that there is full mixing in the beginning of September in CRCM-12.6 simulation with temperatures of about 20 deg C? Fig. 14 also shows that the clearer the water, the higher the water temperatures in Oct and Nov. Note that in addition to convection, mixing is related to wind forcing and density gradient in the water column.

CRCM-12.6 has the clearest water compared to other simulations, therefore the water column reaches the same temperature in its layers earlier than other simulations, leading to earlier turnover. Figures 11 and 13 and 14 support this statement.

Page 12, lines 16-17. MLD is not influenced by the thermal structure, but it is part of the thermal structure. I would remove this sentence.

The sentence has been removed.

Page 12, lines 13-14. Fig. 13 is essentially the same data as in Fig. 12 and therefore one cannot be used to confirm the results of the other.

"confirms" has been changed to "also represents".

Page 12, lines 20-22. Deepening of the thermocline is related e.g. to wind forcing and thus it cannot be suggested that thermocline deepening in clear waters is monotonic. Also, it is not clear what is meant with 'stabilize the temperatures'. I suggest to remove this sentence.

Figure 13 shows that in the simulation related to the most clear waters (CRCM-12.6), deepening of thermocline is faster, with a monotonic speed, as opposed to the dark waters.

Convection that transfer heat between layers continues until temperature is fixed (stabilized) in all layers.

Technical corrections

Page 6, line 23. The same result 'was' found for. . .

It has been corrected

Page 8, line 32. 'leads to higher water clarity'. The authors must mean lower water clarity.

Thanks for catching this mistake. It has been corrected

Page 9, line 18. The sentence starting 'The Kd values' and the sentence after that could be merged and rephrased. E.g. 'The monthly-averaged Kd were used to simulate the surface water temperature and produce a merged LSWT (Merged)."

The authors would prefer to keep the sentences separate. This is to more emphasize on the steps taken to produce the merged product. First simulations were run using the monthly averages, and then a merged LSWT is produced.

Page 9, line 20. Comparing LSWT in situ observations (Obs) with. . .

It has been added.

Page 9, line 21. How can the authors compare measured and modelled surface temperatures in April when there seems to be little or no measured LSWT during April, at least according to Fig 9?

Observations for 2006 and 2007 starts from 19 and 18 April, respectively.

Page 10, lines 2-3. Rephrase. Do the authors mean that the annual average of Kd can occasionally be closer to the actual Kd than the monthly-averaged Kd? This same topic is also mentioned in

lines 16-17, and at least to me it is unclear how yearly average value (i.e. one single number) can represent the extent of Kd variations (i.e. how big is the range).

Yes. Satellite images are not always available (due to cloud cover or shortage of temporal resolution) to cover the actual variations of Kd in the station on the lake. Having a yearly average has a higher chance of capturing potential variations in Kd value and calculate the average of them; therefore is a better and closer representative of the actual Kd value. The statement has been rephrased.

Page 10, line 10. Please clarify what is specifically meant with 'are as affected'.

It means that no matter which depth we use, the actual depth at station or a tile depth, the large under-prediction is happening for these two simulations of CRCM-12.6 and CRCM-20 (MBE for both is above 1ºC); especially for temperatures above 12 ºC.

Page 10, lines 11-12. 'This can be explained by. . .'. Be more specific in telling how lake depth explains this.

CRCM-12.6 and CRCM-20 only differ in the depth used as input in the simulations. Therefore, if CRCM-20 has the most under-prediction compared to all other simulations (including CRCM-12.6), it is related to the input depth. Clarification has been added to the manuscript.

Page 10, lines 12-13. This should be self-evident if the model is any good, and therefore I suggest to remove this sentence.

The authors would prefer to keep the statement to emphasize on this and other studies results.

Page 11, line 9. '. . . causing thinner mixing depth (Fig. 12)'

It has been corrected.

Page 11, line 35. Change 'when Kd changes. . .' e.g. to 'when maximum (minimum) Kd is used instead of its average value. . .

It has been corrected.

Page 12, line 1. Similar comment as previous. This is a bit misleading wording since it gives the idea that Kd changes naturally, whereas what is meant that different Kd is used as an input.

It has been corrected.

Page 13, line 27. Write open the abbreviation 'LSWT' here. It is not typical abbreviation and not clear for those who only read Summary and conclusions.

It has been corrected.

Page 14, line 3. Change 'has' to 'have'.

It has been corrected.

Comments to figures

Fig. 1. It would be of interest to see the main river inlets and outlets. This way it would be easier to assess how much river inflow possibly affects modeling results.

It has been added to the figure.

Fig. 5. Remove 'Lake Erie boundary' from the legend, it is not needed. Also make the color bar much larger. Same comments for other similar figures.

It has been corrected.

Fig. 8. It would be of interest to see the SDD at this location (or from the nearest location where those exist) together with these CC-derived Kd. These could be marked to the same graph with secondary y-axis.

There is no SDD in situ measurements at NDBC station. According to Fig. 1, the nearest locations with SDD observations are within about 20 km distance from NDBC station. However, water optical properties changes in spatial scale smaller than this distance. Therefore showing SDD values for those station are not a good approximate for SDD at NDBC station.

Fig. 10. It would be interesting to see the performance for each year separately. This could shown by plotting each year with different color. Also, it is more standard to show these kind of scatter plots as box plots (both axes of same length).

The performance of each year separately is shown in Table 3.

Fig. 11. The measured LSWT should be shown. Otherwise, it is impossible to say which simulation performs the best. Use a) and b) for these two graphs. Also in the legend, the Kd values could be shown for each model run.

(a) and b) has been used in the new manuscript. Kd values has been shown in the legend of new manuscript.

However, because this figure is related to sensitivity analysis, there is no need to show the observations. Section 3.2.1, which is more related to the accuracy assessment and improvement of simulation results, has shown observations.

Fig. 15. Model run CRCM-12.6. is not visible. If the resolution can not be increased, describe in the caption where the line is.

Description of the figure was given in the body of manuscript (page 13 lines 33-4), however it has been also added in the caption.

---

## Author Comment (AC3) · 24 May 2016

**We would like to thank all the reviewers for their constructive comments which helped improve the manuscript. Our replies to comments are covered below.**

General comments

The manuscript presents a study on the use of light extinction coefficient values derived from MERIS satellite imagery in the FLake 1-D lake model. FLake is the most widely used lake scheme in numerical weather prediction (NWP) models. To take advantage of the coupling of lake schemes to NWP and climate models it is necessary to have data on the lakes transparency. As they are not observations in an in-situ operational way, the most promising strategy is to use satellite images. Therefore this study deals with a current scientific issue that really fit in the HESS scope. As far as my English allow, the manuscript is well written. The study is original and contains new results which are worth to be published. In my opinion the manuscript requires major revision before being accepted. Please, consider the following comments, with different levels of importance:

Detailed comments:

pag. 2 line 3 land → continental

It has been corrected, thanks.

line 12: In the first mention to the FLake model, a reference to the model may be given. This: Mironov, D. V., 2008: Parametrization of lakes in numerical weather prediction. Description of a lake model. COSMO Technical Report, No. 11, Deutscher Wetterdienst, Offenbach am Main, Germany, 41 pp. or this: Mironov, D., E. Heise, E. Kourzeneva, B. Ritter, N. Schneider, and A. Terzhevik, 2010: Implementation of the lake parametrisation scheme FLake into the numerical weather prediction model COSMO. Boreal Env. Res., 15, 218-230.

Mironov et al. (2010) is already added. Mironov (2008) has been added to the new manuscript as well.

line 13 "artificially limited to 40-60 m depth". Is not artificial. Flake is not able to simulated deep lakes, as it consider only two layers. To lakes deeper than 40 meters it will be necessary to consider one third layer below the termocline.

By using the word "Artificially", we mean for deeper lakes (depth more than 40 m), which also form hypolimnion, an artificial depth of 40 m is used in simulation rather than the actual lake depth to only reproduce lake properties in epilimnion and thermocline layer.

page4 line 3. Only 1 station for all the lake?

Only one station is selected as the purpose of this study is to investigate how satellite-derived lake water clarity can improve a 1-D lake model such as FLake in comparison with NDBC station observations, which has lake surface water temperature in situ observations available to evaluate the performance of modeling while employing different values of Kd.

page 5 line 10: T at which station, at which level?

Page 4 line 2-5 describe the location of station and the level that air temperature measurements were conducted.

line 15 – Why not use data from analysis (From ECMWF, for example)

The ECMWF data are not used because of the of the resolution differences. Authors preferred the modeling methods used in this study. The modeling also achieved acceptable accuracies.

Section 2.3 In my opinion, this section should identify the time periods in which Kd MERIS images were available. This information exists in a dispersed form in section 3.1.2..

Section 2 includes a description of the sources of data used in this study. Subsection 2.3 also has information about the satellite derived water clarity and how this data is produced and extracted in general. Section 3 cover results of applying the considered methods to derive information for Lake Erie NDBC station. Therefore, subsection 3.1.2 describe water clarity information derived from satellite observation on Lake Erie and specifically for NDBC station. This section ends with a time series of satellite-derived Kd values at NDBC station. The reader can find out from this graph, that how often and at what time of year, the satellite-derived Kd values were available at NDBC station.

line 26 – 31. It is not clear which Kd is used: in a spectral band (which ?), broadband ?

The Kd average value in the visible part of spectrum was used. This information has been added to the new version of manuscript.

page 6 Section 2.4: In this section, the set-up of the simulations should be clearly presented, namely: - How were the FLake prognostic variables initialized? - The model integration period (start and end) - The temporal resolution of the forcing and the time step of the simulations.

It has been added to the new manuscript.

lines 14/11 In this paragraph the ice parametrization used should also be introduced as it was activated. (maybe also the snow scheme).

It has been added to the new manuscript.

line 18: what means "The setup conditions"?

The condition that the observation at buoy station are collected, is setup condition; and include the parameters summarized in the same line in brackets.

line 19/20- depth of the water temperature measurements: why is it included here? The water temperature is not a forcing parameter...

"z_Tw_m" is one of the inputs in FLake model and is the depth of water temperature measurements.

line 21: why use the local depth (12.6 m) and not an averaged depth, maybe of the western basin?

The exact depth of NDBC station is used in simulations to remove the effect of depth on simulation results and only focus on the influence of water clarity.

line 21: "to configure" means force initialize, or both?

The parameters mentioned in the bracket are constant and used to force the FLake model.

page 7 line 11 (Eq. 2) and line 30 – Equation I don't understand the process to adjust a relation between Kd and SDD. The equation Eq 2 indicate an inverse proportionality, but the expression obtained is not linear.

Eq 2 is a general format of the relationship between these two parameters, where K is a constant value. This relationship was based on a pioneer study. After that, there were other studies that derived an empirical relationship between two parameters with similar inverse relationship. The relationship derived in our study is also empirical and specifically derived using data collected for Lake Erie; also it is still inverse and in a similar format as the general equation that introduced by the pioneer study.

page 8 line 8: "can explain"? or can detect?

"explain" has been changed to "detect".

line16: Kd = 0.87 m-1: is a satellite-derived value or was in-situ measured? By the way, are there any in situ measurements on the selected day?

The map is derived from satellite observation and as it is mentioned in the manuscript, Kd value of 0.87 m-1 is for NDBC station shown on the map.

There are no in situ SDD measurements in NDBC station during that period of time of our study.

line 18: "on a monthly-basis for one year" but only four months were considered.

Fig. 6 aim to show the monthly variations of Kd. Therefore, because MERIS images were available for four consecutive months in 2010, images in this period (May-August 2010) were selected.

"for one year" has been changed to "one selected year".

Figure 6 shows only particular days and not average values, but the text refers to monthly average values. This question is valid also for the discussion of the results presented in Figure 7. It should be indicated how the averaged values were calculated.

The average values mentioned in line 19 in page 8 are average value for the full lake in a specific day. Therefore those are spatially-averaged, not temporally. The same explanation is valid for Fig. 7. It has been clarified in the manuscript.

Figures 5, 6 and 7. With the chosen color scale, most of the field is in the same color. I think it would be better to use a color scale with higher resolution especially in lower values (could be a non-linear scale, possibly logarithmic). Also, the color scale should have a more detailed legend.

Will be corrected in the new version of manuscript.

line 21/22 and 30: the values of Kd are satellite-derived values or were measured in-situ?

The values are extracted from the maps and therefore are satellite-derived Kd values.

line 23 "full coverage of Lake Erie were only available in May of two consecutive years (2008 and 2009)", but figure 6 show a map for May 2010. Contradiction?

This sentence means that the only month, in two consecutive years, with full coverage MERIS images were only available in May and not any other month. Therefore May of these two years were selected to show variations of Kd. The selected two years could be potentially 2009 and 2010; however, because map of May 2010 was already shown, authors preferred to compare the maps of May 2008 and 2009.

page 9 Neither in the section 2.4, nor here, the start of the simulations were indicated. line 18 and 20: Which depth were used in the Avg and Merged simulations?

All simulations are in the actual depth of 12.6, unless otherwise is mentioned, which is for CRCM-20 simulation. It has been clarified in the new version of manuscript.

In the Figure 9, results for 2007 there are plotted values for Kd for fall. Why are this values not shown in table 2 and not used in the merged simulation?

As it is mentioned in the caption of Table 2, only values that have both LSWT observations and satellite-derived Kd available, are used in the merged simulations. In fall 2007, there are no in situ LSWT (observation in black line).

Analysis of figure 9: Some explanation for the strange spike that occurred in mid September 2006 in the avg simulation? A less pronounced effect also occurred in mid

We are not sure if we understood the comment correctly!

page 10 line 3. It seems strange to defend that year average can be more representative of Kd variations than monthly averages...

Monthly averages are calculated based on satellite-derived Kd values, which might not be available due to cloud coverage in MERIS images. However, there are more MERIS images available in the longer period of one year that can potentially catch the actual variations of Kd value, rather than only a few images (or even none) in a month. Therefore yearly-average Kd could be potentially closer to the actual Kd value.

line 5 "Turbid waters in these months simulate colder LSWT". this statement must be explained. In my opinion, the reason is not on the fact that during those months the waters are turbid, but because the water was more turbid before, during spring and summer, reducing the heating of deep water. This should be discussed further, in particular by analyzing the evolution of deep temperature and column mean temperature (Flake variables)

This statement is explaining the negative MBE in 2005 as opposed to 2006 and 2007 simulations the difference in calculating MBE for years of 2005 - 2007 is taking months of Sep-Nov into calculations. On the other hand the Kd value for same months in year 2005-2007 are in the same range. Therefore the underestimation of LSWT in 2005 cannot be related to more turbid waters before, in spring and summer.

line 11 "This can be explained" → "this is due to".

It has been corrected in the new version.

line12 The conclusion that "a realistic lake depth and Kd value will improve model results" is obviously correct, but, specially concerning the depth, could not be demonstrated using only the two depth values considered.

This conclusion is based on having two simulation of CRCM-12.6 and CRCM-20 that only depth is changing; and between these two simulations, CRCM-12.6 is reproducing LSWT more closely to the observations. Page 9 line 9-10 also mentioned that the study by Martynov et al. (2012) also had the same conclusion regarding depth.

Figure 10 a and b: A hypothesis: If different colors were used for spring and summer fall it may be possible to see and then discuss two different behaviors

Thanks for your suggestion. Colors will be added to the figures to investigate the potential seasonal behavior.

Caption of figure 10: the means of a,b,c,d should be indicated

The means has been added in the new version of manuscript.

line 16/17 I do not agree with:"It is possible that the extent of Kd variations is best represented by the yearly average value". Maybe the problem is that the errors in the determination of monthly mean values may result in a worse simulation...

The possible error in the monthly mean values of Kd is due to the fact that the monthly variations might not be captured by limited MERIS images. However, there are more MERIS images available in a year that can capture the actual variation of Kd value; and the average value is derived based on a larger sample of Kd values; therefore has a higher chance to be close to the actual Kd value.

line 24-26: I would be less categorical, adding for example at the beginning of the statement something like: "In the absence of reliable values of the temporal evolution of Kd, . . .."

It has been added in the new version of manuscript.

Considering that 12.6 m is the realistic lake depth must be better justified. see my comment (page 6 line 21).

We are not sure if we understood the comment clearly.

line 33: The sensitivity of FLake to LSWT, MWCT, MLD, isotherm, ice phenology and thickness???

It has been rephrased in the new version of manuscript.

page 11 lines 2/4: which depth were used in the sensitivity simulations?

The real depth is used for all simulations, except for CRCM-20.

lines 4/5 More than the depth, what is important is to indicate the value of Kd in the RCM-12.6 simulation.

The setup condition for both CRCM simulations are mentioned in the beginning in page 9 line 15-17.

line 7 (maximum or Max) → Max will be enough.

The simulations name is abbreviated to use further in the manuscript.

line 8 The world faster does not seem the most appropriate in: "solar radiation is absorbed faster in turbid waters".What happens is that the radiation is more absorbed in the water surface layer, as explained by the authors afterward.

Solar radiation is both absorbed faster and also more in turbid waters compared to clear ones. This is the reason for abrupt fluctuations in LSWT for turbid waters shown in Fig. 11. As it is more illustrated further in the same paragraph.

line 9 "This shallow layer exchanges heat faster with the atmosphere", is correct but should be explained, In my opinion the main reason has to do with the fact that the as the surface water temperature is higher the sensible and latent heat fluxes increase.

Thanks for the comment, this discussion will be expanded in the revised version of the manuscript.

line 12/13 "However, in fall the loss of energy to the atmosphere is also faster due to the shallow mixing depth" This will not be the main reason. In my opinion the main reason has to do with the fact that the deep (and the mean) water temperature is lower.

We are not sure if we understood the comment correctly!

line 14: "least turbid water" The use of the world "least" here can be confusing, as it is less turbid than what considered in the CRCM-12.6 simulation

What means by "least turbid" is further explained in the bracket. The least turbid water simulation in this study is shown in Min simulation.

line 15: "Min" is enough

The simulations name is abbreviated to use further in the manuscript.

line 18: "FLake is not significantly sensitive to LSWT" It is not correct in terms of English

It has been corrected in the new version of manuscript.

line 25: Please delete the sentence:"For both clear and dark waters, LSWT is warmer than the MWCT, due to being exposed to more intense solar radiation.". The reason is the density! (for water temperatures over 4∘C)

It has been corrected in the new version of manuscript.

lines 25/28. In this discussion it will be interesting to compare also with the FLake deep temperature.

We are not sure if we understood the comment and the term "deep temperature".

page 12 line4 "two turnover". In my opinion the first period without stratification should not be identified as a turnover. As I can imagine, the Flake were initialized with a constant temperature profile.

We believe that the constant temperature profile from the top to the bottom of the lake should be considered as mixing, if we understood the comment correctly!

line 8/9 "As a result, the water column in clear water reaches the temperature of maximum density (4°C) much faster than turbid water ..."?? is not what we can see in Figure 11 (bottom) and in Figure 14!

Thanks for catching this mistake. The sentence has been removed in the new manuscript, since turnover can happen in different temperatures as long as the water column is in the same temperature.

lines 10 / l5 The average values over the whole period does not seem to be relevant in this discussion

The Avg simulation is considered as a base for comparison between simulations with highest and lowest Kd values (clear and turbid waters).

line 10 "In the turbid" → "In the more turbid"

It has been corrected in the new version of manuscript.

line 13 "distribute to" "be absorbed in" or "distribute energy to"

Comment is not clear. Solar radiation can be distributed in a volume.

line 16: We can not say that"The MLD is influenced by the water column thermal structure". The MLD is itself a parameter used by Flake to characterize the water thermal structure...

Thanks for your comment. The sentence has been removed in the new manuscript.

Caption of figure 13. Please, improve the wording... The 4 individual figures should have the same caption.

It will be improved, thanks.

line 19: "but also warmer" is not valid for the whole period. I think it will be more correct to say something like: "warmer in spring and summer, and colder in fall"

It has been corrected in the new version of manuscript.

line 19/ 20: the sentences: "The reason is that solar radiation is mostly absorbed at the upper layers in turbid water. Thus, the radiation is used to warm up a thinner layer in dark waters leading to higher temperatures." are correct but the argument is repeated sometimes on the text. In my opinion, it will be better to explain in a more integrated way, based on physics of course, the differences between clear and turbid waters.

This discussion will be expanded in the revised version of the manuscript!

line 21: "shows that the deepening of the thermocline layer in clear waters is monotonic". I can not see this. Can you be more precise?

Figure 13 shows that in the simulation related to the clearest waters (CRCM-12.6), deepening of thermocline is faster, with a monotonic speed, as opposed to the dark waters.

line 29: before the "increased effect of cooling from the layers below" it should be noted that as the surface temperature of the turbid lakes is higher, the radiative losses to the atmosphere are greater. So, during the heating period, a turbid lake as a whole, loses more energy by radiation and therefore stores less energy..

This is mentioned in page 11 line 29.

page 13 (before Summary and Conclusions) It is difficult to analyze the discussion contained in this page without knowing the details about the initialization of the simulations. And about observations? When occurred the break-up and the freeze-up?

The timing of break-up and freeze-up for all simulations is mentioned in this paragraph. The initialization of the simulations has been added to the new version of manuscript.

line 13. "Dark waters store more heat in a shallower depth." The sentence may be misunderstood. First consider change "depth" by "layer". But if one consider the whole water column, dark waters store less heat. "Therefore, in the winter time". In my opinion the "in the winter time" should be deleted, as this is also valid in summer and autumn (and may be more important during these seasons)

"depth" has been changed "layer"; and "in the winter time" has been removed in the new of manuscript.

page 16 line 21. Arkady Terzhevik should be added to the list of co-authors.

Thanks for noticing this. It has been added to the new version of manuscript.

---

## Author Response (AR1)

[revised manuscript text omitted]
 stronger for LSWT above 12°C, which can be explained by the Kd value used. This is because, 5 sinceno matter what depth is used in simulations (either both-actual or tile depth), depths considered in both CRCM runs are as affected have larger MBE compared to Avg and Merged simulations.
- 25 However, the CRCM-20 simulation tends to produce the coldest LSWT (the most under-prediction; MBE = -1.37 °C). This can be explained by is due to the lake depth value considered for the model run which corresponds to the tile depth as opposed to the other simulations that were based on using the actual depth at station. This shows clearly that applying a realistic lake depth and Kd value will improve model results and therefore the parameterization schemes.
- Fig. 10-a and -b show that the resulting LSWT from yearly average (Ave) and monthly average (Merged) Kd are not significantly different, whereas simulations with yearly average Kd reproduces LSWT with improved RMSE and MBE values compared to monthly average (Avg: RMSE=1.54 °C, MBE=-0.08 °C; Merged: RMSE=1.57 °C, MBE=-0.14 °C). It is possible that the extent of Kd 
[revised manuscript text omitted]

|---------|-----------|------|-------|------|
|         | Avg2005   | 1.69 | -0.86 | 0.87 |
| 2005    | Merged    | 1.76 | -0.95 | 0.86 |
| May-Nov | CRCM-12.6 | 1.88 | -1.52 | 0.85 |
|         | CRCM-20   | 2.12 | -1.54 | 0.83 |
|         | Avg2006   | 1.40 | 0.59  | 0.89 |
| 2006    | Merged    | 1.42 | 0.54  | 0.89 |
| Apr-Aug | CRCM-12.6 | 1.50 | -0.98 | 0.89 |
|         | CRCM-20   | 1.47 | -1.09 | 0.89 |
|         | Avg2007   | 1.37 | 0.62  | 0.90 |
| 2007    | Merged    | 1.35 | 0.57  | 0.91 |
| Apr-Aug | CRCM-12.6 | 1.78 | -1.08 | 0.86 |
|         | CRCM-20   | 1.80 | -1.35 | 0.87 |

---

## Referee Report (RR1)

In my opinion, the authors improved the manuscript and responded satisfactorily to most of the questions that I put in my first review. However, I still have disagreements regarding 5 issues. In particular, the question 4 is to me a major issue.

The review is based on the of file hess-2016-82-author_response-version1.pdf

1. page 6, line 21/22
In my first review, I asked: depth of the water temperature measurements: why is it included here? The water temperature is not a forcing parameter...
The authors reply that:
"z_Tw_m" is one of the inputs in FLake model and is the depth of water temperature measurements."
Which is not correct.
I'm not sure about what is z_Tw_m (It is not present in the standard Flake code, see: http://www.flake.igb-berlin.de/docs.shtml), but I have no doubt that Flake don't need any "depth of water temperature measurements" as imput.

2. Page 6 line, line 25: I asked "to configure" means force initialize, or both?
The authors reply: The parameters mentioned in the bracket are constant and used to force the FLake model."
I have to insist. The authors made some confusion between configuration parameters and forcing. The height of wind measurement (5 m) and the height of air temperature sensor (4 m) are in fact constant used to configure the model, but the measured meteorological parameters and model-derived irradiance were used to **force** the FLake model (and not to configure).

3.
page 7 line 22 and equation 2
I also insist about equation 2. The authors find the following relation between kd and SDD:
$K_d = 1.64 \times SDD^{-0.76}$
I have nothing against this result, but they should not say that it is of the type of eq 2...

4. Page 10, lines 17-18:
"Dark waters in these months contribute in reproducing colder LSWT for Avg and Merged simulations in 2005."
I can not understand why water turbidity (or darkness) tends to decrease the lake surface water temperature. As the authors correctly stated along the manuscript, in dark waters the radiation is more aborved in the surface layer, and so the LSWT tends to be higher by comparison to clear waters. What I proposed in my previous review is that:
The reason lies not in the fact that during those months the water is more turbid, but because the water was more turbid before, during spring and summer, reducing the heating of deep water. This should be discussed further, in particular by analyzing the evolution of deep temperature and column mean temperature (Flake variables).
The authors reply: "According to Table 2, the Kd value for same months of year 2005-2007 are in the same range. But the difference in calculating MBE for 2005 compared to 2006 - 2007 is taking months of Sep-Nov into the calculation of MBE for 2005. Therefore, the underestimation of LSWT in 2005 cannot be related to darker waters before, in spring and summer. It is more related to the months that are taken into the calculation of MBE."
In my opinion the authors don't have reason and must examine and take into account the evolution of lake water bottom temperature (LWBT) and column mean temperature (MWCT). Alternatively they

have to provide a physically explanation about how "Dark waters in these months contribute in reproducing colder LSWT(...)".

5.
Pag 13 line 11:
I still consider that the use of the term monotonic is not appropriate

---

## Referee Report (RR2)

Reviewer comments to Satellite-Derived Light Extinction Coefficient and its Impact on Thermal Structure Simulations in a 1-D Lake Model by Zolfaghari et al. 2016

The authors have significantly improved the manuscript and replied adequately to most of the reviewers' comments. The manuscript is now structurally sound and easy to follow. There are still several things that need clarifying and corrections, but these can be considered as minor comments.

General comments:

Abstract is long and occasionally overly detailed. It should be more compact. Acronyms should be used only when necessary. Now there are e.g. MBE and I_a, which are not commonly known. Since the abstract should be shortened, I suggest to leave out these statistical parameters.

Sometimes present tense is used in the manuscript even though past tense is more appropriate.

Specific comments:

Page 1, lines 11-12. "...algorithm is applied to MERIS satellite imagery to estimate Kd and evaluated against Kd derived from Secchi disk depth...". As written, it is not clear if the algorithm is evaluated against Kd from SDD or is it the estimated Kd that is evaluated against Kd from SDD. I suggest to change it to: "... to estimate Kd, which was evaluated against...". I'm not a native English speaker but think that more attention should be paid on sentence structure throughout the manuscript.

Page 1, line 24-25. "Dark waters always produce warmer MWCT". Should it read 'colder MWCT'? At least Fig. 6b implies so.

Page 2, line 8. "Lake Surface Water Temperature...", only the first word should have capital initial letter.

Page 2, the second paragraph is really long and could be separated into two paragraphs. The second paragraph could start e.g. after "...derived from satellite imagery." in line 28.

Page 2, line 27. The reference Heiskanen et al., 2015 can be omitted because it is clear from the sentence that the authors refer to this study.

Page 2, lines 32-33. "The daily mean LSWT range increased...". It is not clear what is meant with the word 'range' because only one number (not range) is given for clear and dark waters.

Page 3, line 19. "... to investigate the improvement...". If this is the first study to use satellite-derived Kd in FLake simulations, how can it be known from beforehand that the model performance is improved? Maybe the word 'improvement' could just be omitted: "... to investigate the model performance..."

Page 3, line 20. "... and a constant value...". Be more specific, a constant value throughout the study period, or constant value of 0.2 m-1 or some other type of constant value.

Page 3, line 20. I suggest to change the word 'demonstrate' to 'evaluate' for pretty much the same reason as in the comment about 'improvement'.

Page 3, line 23. "... to Kd values based on simulated LSWT,...". Sentence structure should be changed because now it reads that the Kd values are based on simulated LSWT etc.

Chapter 2.1. There are now many references to Fig. 1 (page 4, lines 3-4, 10-11, 13, 23) even though only one would be needed. I suggest to put a sentence in the beginning of the paragraph starting "The meteorological forcing…" that says e.g. "The data for this study was collected from different stations shown in Fig. 1.". Then delete the other references to Fig. 1.

Page 4, line 4. Remove the coordinates and depth from here or from page 6, line 21.

Page 4, line 6. "Water temperature is also measured at 0.6 m below the water line.". It seems to me that water temperature is measured only at 0.6 m depth, and therefore the word 'also' should be deleted. I also suggest to use 'water surface' instead of 'water line'.

Page 4, line 32. Does the "(see Fig. 1)" refer to the corresponding tile or only to the station? If only to the station, then this ref. can be omitted. If it refers to the tile, that should be specified in Fig. 1. Now there are vertical dashed lines in the figure but they are not defined in the caption.

Page 5, line 10. At least to me it is unclear what the term 'screen height' means. If this is generally known definition, then okay, but if not, it should be specified.

Page 5, line 12. "NWRI-EC", change to "NWRI" so that same terms would be used throughout the manuscript.

Page 5, line 18. "… Earth's surface high spectral…". Is there a word missing between 'surface' and 'high'?

Page 5, line 23. The "in Lake Erie (see Fig. 1)" could be omitted, it has already been defined where the station is.

Page 5, line 25. The authors use different terms to mean the same thing, e.g. "CC Level2W", "CC MERIS L2W", "CC L2W". Make sure that the same term is always used. It is also not clear what Level2 data product means.

Page 5, line 26. "concentration of water constituents". This surely doesn't mean ALL the constituents, so it should be changed to "concentration of some water constituents" or similar.

Page 6, line 18. Are all these references really needed for this statement?

Page 7, line 1. I suppose that it should read that the MBE is calculated as the *mean of* modelled values…

Page 7, line 12. Could the authors provide a reference for the statement that resuspension is the most important cause for low water clarity in Lake Erie? Later, heavy plankton blooms are also discussed as the cause of high differences in water clarity.

Page 7, equation 2. It has been shown in later studies that the relationship between Kd and SDD is not constant, as the authors also discuss later in this chapter. It is unclear why the authors want to argue here that it is constant. Or has this relationship been the basis of relating CC-derived Kd and SDD together? If so, why that was used instead of the exponential relationship shown e.g. in Arst et al. 2008?

Page 8, lines 15-17. The authors say that, after SDD validation, the satellite-derived water clarity (as such, without any modifications) was used in the modelling. There seems to be quite good agreement between SDD and CC-derived Kd when SDD>3m (Fig. 3). However, in Discussion the authors state that in situ measurements of water clarity are a requirement for satellite-derived Kd. Is there some reason to assume that in (clear) waters in general (e.g. SDD>3m) the satellite imagery doesn't provide reasonably reliable estimates of Kd? In other words, if it worked for Lake Erie, what are the assumptions that it might not work on other lakes?

Page 8, line 16. "are deemed to be correct". There is huge scatter in Fig. 3 when SDD<2m, so it is an overstatement to say that the satellite-derived Kd are correct. I suggest to change the wording to "can be considered representative".

Page 8, lines 21-22. Very complex and vague subordinate clause (motivating the investigation of potential of integrating). Be more specific.

Page 8, line 30. "focused on", change e.g. to "shown for"

Page 9, paragraphs 1 and 2. It is quite difficult to follow the big picture what was done. This could be made easier for the reader if e.g. after "..., respectively." on line 4, a new sentence would be written: "We made four different simulation schemes which were then compared to the observed LSWT".

Page 9, line 6-7. Where does the acronym CRCM come from?

Page 9, line 11. Change sentence order from "surface temperature in spring (April-June) is modelled warmer" to "surface temperature was modelled warmer in spring (April-June)"

Page 9, lines 11-13. Looking at Fig. 4, it seems to me that 2006 and 2007 'merged', 'avg', and 'obs' are quite similar from late-June to mid-August. This is contrasting to what is now written in the text.

Page 9, line 16. Change sentence order from "to more slowly gain (lose) heat" to "to gain (lose) heat more slowly".

Page 9, line 17. Change "The performance" to "The overall performance".

Page 9, paragraph starting with Fig. 5. and Fig. 5 itself. Write the text in the same order as they are presented in the figure. Either change the order in the text or in the figure. Now the discussion starts with CRCM simulations whereas they are panels c and d in the figure.

Page 9, line 29-30. This needs a bit of rephrasing because according to the figure the CRCM simulations underestimate the observed LSWT only when LSWT is roughly >7 deg C.

Page 10, line 29. The authors argue that solar radiation is absorbed more in dark waters due to existing particles in water. If you can reliably state that this is true for Lake Erie, then the explanation is acceptable. But even in this case I suggest to change 'dark waters' e.g. to 'waters with low clarity" because particles do not always make the water dark, and the water can be dark without particulate matter.

Page 10, line 30. "(lower LBWT...)". I suggest to change it to "(which shows in lower LBWT...)" so that the reader doesn't need to guess what the authors try to imply.

Page 10, line 30. Fig. 8 can not be introduced before Fig. 7 has been introduced. Either change the text or one option would also be to combine Fig. 8 to Fig. 6 (it could e.g. be subplot d in Fig. 6).

Page 11, line 26. "lessening of the radiative absorption" implies that something has happened in the lake. I suggest to change this to "decrease in radiative forcing" which means that the incoming radiation from the atmosphere has decreased.

Page 11, lines 28-29. These are very vague sentences and not all parts true. The deepening of the thermocline is related to wind forcing. In dark waters the density gradient is sharp and forms an effective barrier for the wind-induced mixing to reach deeper depths. In clear waters the density gradient is weaker and therefore mixing can more easily deepen the thermocline. There are many processes working at the same time in lakes that affect thermal stratification. Besides heat transfer, wind currents and internal waves are important. Because the same wind forcing is used as an input for all the different model runs, it is important to explain how water clarity takes part into the development and progress of thermocline.

Page 11, line 30. "derived from isotherm". Be more specific how MLD was defined. It seems to me that the authors have identified the MLD correctly, but there are many ways to do this and no general guideline how to do this. Therefore, more specification is needed.

Page 12, paragraph starting with "In the darkest water...". Here the same oversimplifications are presented. If there is no wind, there is no mixed layer in clear or dark water because there is only stratification, no mixing. So the explanation is not only that in clear waters the solar radiation can distribute to a larger volume in the water column. Very important factor is also how much (deep) of the density stratification can be destroyed by wind-induced turbulent kinetic energy. In dark waters this layer is shallower than in clear waters and therefore dark waters have shallower MLD with the same wind forcing.

Page 12, lines 9-25. If there is some study that shows that FLake predicts well the ice phenology in Lake Erie, then this text can be as it is and that study should be cited. Otherwise, these are only simulation runs without validation, and therefore in the beginning or in the end of these paragraphs a text should be added that mentions "It must be noted that these results couldn't be verified because of lack of measurements" or similar.

Chapter 3.3. This chapter needs the most modification. Now there are three main points in the chapter: 1. spatial variation, 2. temporal variation, and 3. inter-annual variation of Kd. Currently, the authors briefly describe what was observed and show the figures, but the meaning and importance of these findings are not elaborated. From the figures 10-12 it seems evident that these are important findings but these are not discussed. For example, it seems interesting that Kd can be time-independent constant even though there are huge changes in Kd both in space and in time. If Kd influences the thermal stratification as shown in Fig. 7 and related studies, then it would be reasonable to assume that the thermal stratification is very different in the western end of Lake Erie than in the eastern end. Yet, some studies and results suggest that one lake-specific but constant Kd can be used to model the stratification. Fig. 11 seems to imply that in big lakes lake-specific Kd cannot be used.

Page 13. Kd values are presented as average value plus minus some number. Could you specify what the number is.

Page 13, line 25. It seems that the years for CC product are 2003-2012 (fig. 2), not 2002-2012. When this paragraph is written as it is now, it seems that these were the years for all the measurements and modelling. In order to not be misleading, specify at least for which years the model runs were made.

Page 14, lines 8-9, first sentence of this paragraph. It is unclear what the authors mean with the concept of 'thermal regime of lakes' in regards of this study. Only observed surface water temperatures are used to validate modelled temperatures. All the rest (water column temperature, bottom water temperature, mixed layer depth) are only simulated and thus tells more of how FLake model performs with different water clarity in this lake than how the lake thermal structure actually was influenced during the years in this study period. Also, it has already been shown in previous studies (which the authors cite) that transparency impacts physical processes, and thus this is not a new finding. I suggest to replace the first sentence of this paragraph with specific strengths of this study.

Page 14, line 22. Change "Flake" to "FLake".

All figures in general. Include tick marks to all figures, both x- and y-axes. Also minor tick marks could be useful in some cases.

Fig. 1. Write the meaning of the acronyms in the caption or describe what the different stations are. Specify what are the vertical dashed lines.

Fig. 2. Write open what 'CC-derived' means.

Fig. 4. It is difficult to understand the caption and how it is exactly linked to the figure. Discuss the lines in the caption in the same order as in the legend of the plots. Write open what 'Obs', 'AvgXXXX', 'Merged', 'CRCM-12.6', and 'CRCM-20' mean. Assign (a), (b), and (c) to the subplots. What is the time resolution of the data? The general principle is that the reader should be able to understand the figure without having the need to constantly see the main text.

Fig. 9. It seems that the green line (CRCM-12.6) is missing from the plot. The last sentence of the caption seems out of place. The proper place should be in Results/Discussion.

Figs. 10-12. Show more values in the colorbars. Now only 0; 2.5 and 5 are shown, the interval should be at least 1 m-1 (i.e. 0, 1, 2, 3, 4 ,5). Show the unit of the colorbar somewhere.

---

## Author Response (AR2)

[revised manuscript text omitted]

We would like to thank the two reviewers for their constructive comments, which greatly helped us to improve the manuscript. Our replies to the comments are listed below.

**Reviewer #2 Comments:**

5 Many major revisions were suggested by the reviewers. Most of these have not been properly addressed in this revised manuscript. For this reason, I have no choice but to recommend not to publish this manuscript.

We are sorry to hear that the reviewer is not satisfy with our answers, despite our efforts to address his/her comments. However, we understand we could do a better job to improve the quality of the paper based on the constructive suggestions provided by the reviewer. This time, we did our best to consider his/her comments more closely and cover his/her questions and concerns

10 in the text. The manuscript is significantly modified and now, we believe the revised version is reflecting his/her thoughts, and we hope this time the reviewer is happy with the new version.

General comments:
This manuscript deals with water optical properties which are acquired by remote sensing, and which are then used as input to

15 1-D lake modeling. This approach is important and needed addition to current efforts of incorporating lakes and reservoirs to weather prediction models. Water clarity is an important factor in defining lake heat budget and thermal stratification and thus is a significant parameter for processes in the air-water interface. With millions lakes of different sizes around the world, comprehensive direct measurements of water clarity are not possible, highlighting the need for indirect estimates of water optical properties. Satellite measurements show great promise in mapping light extinction coefficient, a parameter to define

20 water clarity, and to my knowledge this study is the first one to incorporate lake modeling to water clarity defined from satellite observations. This is an important study with wide interest in scientist from different fields and therefore I find this study appropriate for HESS. However, there are a few major points which prevents me from recommending this manuscript for publication as it is.
The major topic of this manuscript is that satellite-derived light extinction coefficient, Kd, represents well the in situ

25 measurements of Kd, that it can be used as an input to lake modeling, and that it enhances the performance of FLake as compared to the current approach of using constant Kd of 0.2 m-1. These points should be emphasized. Currently, the manuscript seems unbalanced, with much of the focus given to topics not strictly the main theme of this manuscript. Also some restructuring is needed so that the reader does not get distracted from the main focus. In general, the manuscript would benefit from reducing the amount of figures.

30 Thanks for your valuable comment. We have done restructuring to emphasize the main theme of the paper, focusing on the mentioned objectives. The first objective of this manuscript (satellite-derived Kd value can represent the in situ measurements of water clarity, and therefore be used as input to lake modeling) is discussed in section 3.1. Section 3.1.1 shows that the variations of water clarity at NDBC station can be derived from satellite observations. Section 3.1.2 shows that these satellite-measured water clarity is representing the in situ measurements of water clarity because there is a high agreement between the

35 field measured values of SDD and satellite observations of Kd. Therefore, the satellite-derived Kd can be used in the lake modeling, as discussed in the next section of manuscript (section 3.2).
The second objective of the paper is to show that using the satellite-derived lake-specific Kd value can improve the performance of lake modeling, compared to the current approaches of using a constant Kd value. This is covered in the next section, section 3.2.1. The range of Kd variations that brings the most sensitivity for the modelling is discussed in section 3.2.2.

40 The next section, section 3.3, shows how remote sensing observations can capture the spatial and temporal variations of Kd, which is not possible through the conventional point-wise field sampling methods. This section highlights the importance and strength of Kd satellite-derived values, which can be utilized in lake modeling to improve their performance.
Some of the figures are removed, as it will be mentioned in the following answers.
Since this is the first approach in combining satellite-derived Kd to lake modeling, it would be of value to describe the strengths

45 and the weaknesses of this approach. How easy and accurate method this is for the modeling community in general; can this

be used without in situ measurements of Kd or should there always be e.g. Secchi disk measurements for validation; what are the next steps needed for applying this method in broader context.

Thank you for your comment.

**The previous reply:** Thank you for the detailed comments. With a few exceptions (e.g. restructuring of the paper which was not a concern for Reviewers #1 and #3), we have considered all suggestions in our revised manuscript.

The strength of integrating satellite-derived Kd values in lake models is mentioned on page 14, lines 9-12 (Integrating lake specific Kd values can improve the performance of 1-D lake models. However, field measurements of Kd are not widely available. This study demonstrates that satellite observations are a reliable data source to provide lake models with global estimates of Kd with high spatial and temporal resolutions).

The weaknesses of this approach could be that the validation of satellite-derived Kd values, and therefore the results of the lake models (which are based on using satellite information), depend on in situ SDD or water clarity. Therefore, in situ data is a requirement for the validation approach. This information was provided on page 7, lines 14-15.

The globally available CC product can be easily used as a source to fill the gaps in Kd in situ observations as well as improving the performance of lake parameterization schemes and, therefore, further improve NWP and regional climate models. This is mentioned on page 14, lines 12-14.

The next steps are mentioned in the manuscript on page 14, lines 14-117: to investigate the potential of Sentinel-3 to provide lake modeling community with the water clarity information. Also, this study demonstrates improvement in a lake surface scheme (Flake) commonly used in NWP and regional climate models.

**The new reply:** The last paragraph in the conclusion section is modified to represent these points. The strength of integrating satellite-derived Kd values in lake models is mentioned on page 14, lines 8-12 (Results of this study have important implications for understanding the thermal regime of lakes and show that the transparency of lakes can impact physical processes by influencing changes in seasonal mixing regime. Integrating lake specific Kd values can improve the performance of 1-D lake models. Although field measurements of Kd are not widely available, this study demonstrates the strength of satellite observations and introduces them as a reliable data source to provide lake models with global estimates of Kd with high spatial and temporal resolutions. ).

The weaknesses of this approach is added on page 14, lines 12-15 (However, the weakness of this method is that the availability of satellite-derived Kd product can be limited due to cloud coverage or satellite overpass. Also, the in situ measurements are still required for validating satellite observations, because the in situ data collection remains the most accurate solution for water clarity measurement.).

Modifications have been made in page 14, lines 15-18 to show how easy and accurate this method can be transferred to other study areas (The accuracy of the satellite-derived Kd product has to be verified for the water body of interest, especially for the ones with complex optical properties. After validation, the on-demand globally available CC product can be simply used for the water body of interest, as a source to fill the gaps in Kd in situ observations, and improve the performance of parameterization schemes and, as a result, further improve the NWP and climate models.).

The next steps are mentioned in the manuscript on page 14, lines 18-23 (Although MERIS is no longer active, the Ocean and Land Colour Instrument (OLCI) to be operated on the ESA Sentinel-3 satellite (launched on February 16, 2016) will provide continuity of MERIS-like data. OLCI has MERIS heritages and improves upon it with an additional six spectral bands. Therefore, investigation of the Sentinel-3 potential to provide lake modelling community with the water clarity information is the next step of the current study. Also, the possible improvement in Flake output, when forcing the model with air humidity data collected directly at the station, can be examined in the future studies).

Specific comments

In regards of restructuring, I will give here an example how the figures (and related discussion) could be rearranged. After the map, I suggest to first show satellite-derived Kd at the site of FLake modeling (current Fig. 8), which is the main input parameter under focus here. For the estimated solar irradiance and incoming long-wave radiation (Figs.2 and 3), the figures add only little to the reported statistics and thus these two figures could be removed. After showing the satellite-derived Kd, it

would be logical to show their validation (current Fig. 4). Then the results of models against measurements, i.e. current Figs. 9 and 10. Current Figs. 11, 12, 13 and 14 basically show the same data in different forms. I suggest either to combine Figs. 11 and 12 and show only this, or show only Fig 13. Lastly, current Figs. 5 and 6 could be shown, which would then lead to the discussion of the strengths of satellite-derived Kd and possible future studies (see also the related comment later). **This is only**

5   **a suggestion for restructuring, it could also be done otherwise.**

**The previous reply:** Thank you for the suggestion. However, we would like to keep the current structure of the paper. This study is based on using satellite-derived shortwave radiation. Longwave radiation is also estimated. Therefore, these two parameters need to be evaluated for the study area of interest (Figures 2 and 3). We feel that the quality of shortwave and longwave radiation estimates needed to be confirmed first before being used in FLake simulations. Of the other input

10   parameters in the FLake model, the most important one is water clarity which is also derived from satellite observations. Before further discussing the potential of combining satellite observations of water attenuation into the model, first the evaluation was conducted (Figure 4). Following this, Figures 5-7 show the extent of spatial and temporal variations of water clarity (now evaluated) for the entire water body. Figure 8 shows how water clarity has changed at the station of interest (NDBC) during the study period. Based on the variations (min, max, average values of Kd at this station), different simulations were designed

15   to test the sensitivity of FLake to variations of Kd.

Figure 9 shows how observations and different simulations compare over the study period. The figure illustrates if there is any specific time when the difference between simulations and observations is more prominent; whereas Figure 10 is more for statistical evaluation purposes.

Figures 11-14 may somewhat overlap in the presentation of information. However, the figures are used in this study for

20   different purposes. Figure 11 shows LSWT and MWCT, Figure 12 MLD, Figure 13 timing, depth, and temperature of different thermal layers (epilimnion, thermocline) as well as temperature of MLD. Finally, Figure 14, shows the average temperature and depth of each thermal layer. Given the above, we find that the paper follows a logical structure.

**The new reply:** Thanks for your suggestion regarding restructuring. Figure 2 and 3 in the previous version have been removed and modifications are made in page 5 lines 1 and 12.

25   The variations of Kd at NDBC station is now moved to the beginning of section 3.1 (new section 3.1.1); and the evaluation and accuracy of the CC-derived Kd is calculated after showing Kd variations in NDBC (new section 3.1.2). Modifications are also made in page 5 lines 31-33 and page 6 line 1.

The spatial and temporal variation of Kd in the full lake is shown in section 3.3 to demonstrate the potential of remote sensing to capture the variation of this water quality parameter, whereas this is not possible using the conventional in situ

30   measurements. These information are added on page 12 lines 27-31.

Figure 14 in the old version of manuscript is removed. We agree that Figures 7-8 (in the new version of manuscript) have overlaps in the presentation of information, but they are shown for different purposes. Figure 7 shows the evolution of different thermal layers during time, as well as their temperature. However, Figure 8 is derived from Figure 7 and temperature is not anymore a factor in Figure 8, and the focus is only on MLD. This way, demonstrating the variations of MLD during time is

35   shown better and more clear compared to Figure 7. We feel that keeping figure 7 and removing Figure 8, to discuss all the points only relying on Figure 7, makes it difficult to convey the information and also to understand the discussion for the reader.

Page 4, line 9. Air humidity, which is used as FLake input, is taken from land 10 km away from the lake site. Air humidity is important for modelled latent heat fluxes. Could the authors briefly state their opinion how well the measured air humidity

40   represent that over lake, and how or if this affects the modeled results.

Thank you for this comment. Yes, we agree that air humidity is important for modeled latent heat flux and would be different over land and over lake. Since warm air holds more moisture than cold air, the percentage of humidity must change with change in air temperature. We expect that the humidity decreases as temperature increases. Large temperature and humidly differences can lead to large sensible and latent heat fluxes. Water is a good absorber of the energy but the land absorbs much

45   faster the energy from the sun. Water heats up much slowly than land and, therefore, the air above land will have higher

temperature and therefore less humidity. On the other hand, lack of in situ data over lakes and the distance of the stations from the shoreline (less than 81 km in our case, 10 km was recognized as a mistake in the manuscript, corrected in page 4 line 11 of the new manuscript) is one of the main limitation of lake studies. In this study, all model forcing variables come from the station on land such as air temperature and humidity, therefore in this way the rate of differences between air temperature and humidity was kept constant.

Page 4, lines 19-21. Sentences starting with "Available Secchi disk" and "SDD data was" are repetitive and should be merged to one sentence. Also, it could be mentioned here that the SDD data comes from Limnos cruises.

**The previous reply:** Thank you for the comment. Page 4, lines 20-21 have now been changed in the new version of the manuscript ("Available Secchi disk depth (SDD) field measurements were used to estimate lake water clarity.").

**The new reply:** Thank you for the comment. Page 4, lines 20-22 have now been changed in the new version of the manuscript ("Available Secchi disk depth (SDD) field measurements were collected by EC research cruises on board the Canadian Coast Guard Ship Limnos and utilized in this study to evaluate the satellite-derived water clarity. The cruise visited Lake Erie at a total of 89 distributed stations in five different years").

Page 6, lines 1-2. Please clarify this sentence. Does this basically mean that only the pixels which were not rejected according to the criteria in Table 1 were used?

Yes, you are correct, this is what we meant.

Page 7, Chapter 3.1.1. A lot of space is dedicated for this, and therefore a justification could be given in the first sentence. E.g. "Validating the satellite-derived Kd with in situ observations is important because. . ." And in the end of the chapter the outcome of the evaluation, e.g. "For these reasons, we deem the satellite-derived Kd correct and thus were confident in using them in the modelling." Also, in Chapter 3.1.1. or later, the authors could discuss whether this kind of validation is always needed with satellite observations, what are the implications, etc.

**The previous reply:** The justification of having section 3.1.1 is that the reliability of satellite-derived Kd values, to integrate them in lake models, is highly dependent on their comparison and evaluation against independent in situ SDD measurements. This highlights the importance of this section as mentioned on page 7, lines 14-15.

Also, at the end of this section, the reason and motivation of using satellite-derived water clarity measurements is mentioned on page 8 lines 9-13 (in situ SDD data are not always describing Kd values. Only small values of Kd are described using SDD). On page 7 lines 14-15, it is mentioned that the reliability and validation of satellite-derived Kd values highly depends on comparison with in situ measurements.

**The new reply:** Thanks for your constructive comment. We applied your suggestion with slight changes to fit in the manuscript. The justification of having section 3.1.2 (in the new version) is added in the new manuscript page 7 line 19-20 ("The validation of satellite observations against in situ data is important, because the in situ data are still considered as the most accurate measurement of water clarity.")

Also, at the end of this section, the reason and motivation of using satellite-derived water clarity measurements is added on page 8 lines 12-22 ("Results from the above procedure show that Kd can be derived from SDD, using the equation 〖 K〗_d=1.64× 〖SDD〗^(-0.76), with a strong determination of coefficient value (R2 = 0.78). Arst et al. (2008) obtained a similar regression formula between SDD and Kd for the boreal lakes in Finland and Estonia representing different types of water, expanding from oligotrophic to hypertrophic. Because there is a good agreement between Kd and the corresponding ones estimated from in situ measured SDD (N = 49, RMSE = 0.63 m-1, MBE = -0.09 m-1, I_a = 0.65; Fig. 3), the satellite-derived water clarity are deemed to be correct and were used in the modelling for this study.

However, SDD is not always describing Kd values. SDD is a suitable characteristic to describe water transparency for small values of Kd. For high values of Kd (ranging above 4 m-1), Arst et al. (2008) and Heiskanen et al. (2015) suggest that SDD is unable to describe any changes in Kd. Fig. 3 also shows that SDD cannot describe the scatter of Kd for values above 4 m-1. Therefore, the estimation of Kd from SDD should be used with caution, motivating the investigation on the potential of integrating satellite-based estimations of Kd into lake models.")

On page 7 lines 19-21, and page 14 lines 13-15, it is mentioned that in situ data are still considered the most accurate measurement method and therefore the reliability and validation of satellite-derived Kd values highly depends on comparison with in situ measurements.

Page 8, Chapter 3.1.2. This chapter seems interesting but out of place. These results are not further elaborated, and therefore I suggest to move them to the end of the Discussion. This way the authors could show what benefits remote sensing of Kd would bring (spatial and temporal variability, which is not achieved well with manual sampling; perhaps good input for 2D and 3D modeling), which would lead to the discussion of possible next studies. This way also the key input parameter, Kd at the NDBC station, would be shown earlier.

**The previous reply:** Thank you for the suggestion. This section describes how Kd values vary spatially and temporally over the full lake, and it ends with the variations of Kd at the location of the NDBC station. This section aims to demonstrate how variable Kd can be across the lake and over a period of time, demonstrating the lack of in situ observations to cover these variations temporally and spatially, and highlighting the motivation of using remote sensing observations to overcome these concerns. After highlighting the important role that remote sensing observations can play within lake models, the paper continues by showing the results of integrating satellite-derived water clarity within FLake. Therefore, we would prefer to keep the current structure of the manuscript since, in our minds, it follows a logical sequence.

**The new reply:** Thank you for the suggestion. The section related to the spatial and temporal variations of satellite-derived Kd is moved to the end of discussion in the new version of manuscript (section 3.3). This section starts with explaining the important role of remote sensing measurements to fill the temporal and spatial gaps in the in situ data collection ("As it was described in the previous section, variations in water clarity plays an important role in defining lake heat budget and thermal stratification and thus is a significant parameter for processes in the air-water interface. However, the long term spatial and temporal trends of water clarity cannot be achieved through discontinuous conventional point-wise in situ sampling. These observations can be provided from satellite measurements. This section demonstrates the strength of satellite observations to detect the spatial and temporal variations of Kd in Lake Erie")

Page 9, Chapter 3.2.1. If the satellite-derived Kd has been validated sufficiently well and it produces better simulations, what would be needed for the simulations to match the measured LSWT more accurately? This could lead to suggestions for future research.

**The previous reply:** The comment is not clear for us. Section 3.2.1 discusses the improvement in modeling results using the satellite-derived Kd values. The next section (3.2.2) examines the sensitivity of FLake to the variations of Kd, and if it is necessary to consider the temporal variations (monthly basis) of Kd in simulations or simply a constant-lake specific value in the modeling of Lake Erie.

Therefore, if this comment is suggesting to consider the temporal variations of Kd in simulations for further improvement, this has already been considered and tested for the range of Kd values in Lake Erie.

Perhaps having met. station forcing data at the NDBC station directly could slightly improve the LSWT. However, here we would only be speculating. But the land station being 81 km away could be a factor.

**The new reply:** Thank you for your comment. Section 3.2.1 discusses the improvement in modeling results using the satellite-derived Kd values. The next section (3.2.2) examines the sensitivity of FLake to the variations of Kd, and if it is necessary to consider the temporal variations (monthly basis) of Kd in simulations or simply a constant-lake specific value in the modeling of Lake Erie.

Therefore, if this comment is suggesting to consider the temporal variations of Kd in simulations for further improvement, this has already been considered and tested for the range of Kd values in Lake Erie.

Air humidity data in the current study was provided from a station located 81 km from NDBC station. Therefore, perhaps having met. station forcing data (air humidity in our study) at the NDBC station directly could slightly improve the LSWT. However, here we would only be speculating. But the land station being 81 km away could be a factor. This is added on page 14 line 22-23 to show the potential for future researches.

Page 9, Paragraph starting 'Fig. 9 shows the results. . .'. I suggest to first describe the observed behavior in the temperatures and then discuss how the modelled behaviors compare to these.

**The previous reply:** The authors did not find it necessary to add the observed temperature behavior to the manuscript. This is because the temperature behaviour in three years, 2005-2007, have a normal fluctuation, increasing from spring to summer and decreasing toward winter. This is a basic knowledge.

**The new reply:** Thank you for your suggestion. The figure is now Fig. 4 in the revised manuscript. Modifications are made on page 9 line 1-4 ("Fig. 4 compares the results of different LSWT FLake simulations with observations at the NDBC station. LSWT observations have maximum values of 27.53 ℃, 26.48 ℃, and 25.46 ℃ in August during 2005, 2006 and 2007. The minimum values of 2.71 ℃, 7.3 ℃, and 3.42 ℃ were observed in December 2005, and April in 2006 and 2007. The average LSWT observations in 2005, 2006, and 2007 have values of 18.45 ℃, 17.12 ℃, and 17.75 ℃, respectively. The simulated LSWT values in Fig. 4 are produced by first applying Kd = 0.2 m-1 from Martynov et al. (2012) using both the real lake depth at the station (12.6 m: CRCM-12.6) and also a tile depth corresponding to the station in their study (20 m: CRCM-20). Then, simulations using the yearly average CC-derived Kd for each year of study are plotted (Avg).")

Page 10, lines 17-18. This is quite strong statement and probably not true for all lakes. Lakes are very heterogeneous, be more specific which type of lakes is meant here.

**The previous reply:** These sentences were meant to explain why results of two simulations of Avg and Merged are comparable, while Avg simulation are producing lower MBE. The statement starts with "it is possible". Therefore, it is only a potential reason for such results in Lake Erie, and not a generalized rule for all lakes. However, modification to the sentence has been applied to clearly make this point (Page 10 line 32 also page 1 lines 19-20).

**The new reply:** Thanks for your comment. Sorry for the confusion. These sentences were meant to explain why results of two simulations of Avg and Merged are comparable, while Avg simulation are producing lower MBE. The statement starts with "it is possible". Therefore, it is only a potential reason for such results in Lake Erie, and not a generalized rule for all lakes. However, modification to the sentence has been applied to clearly make this point (Page 10 line 3-6 also page 1 lines 19-20). "It is possible that the extent of Kd variations is best represented by the yearly average value. Therefore, using a constant annual open water season value for Kd could be potentially sufficient to simulate LSWT in 1-D lake models with relatively high accuracy (the range of Kd variations that brings the most sensitivity for the modelling is discussed in Sect. 3.2.2)." This statement is more clarified in the next section when the sensitivity of Flake to the variations of Kd is investigated.

Page 10, Chapter 3.2.2. This chapter needs the most restructuring. E.g. the paragraph on page 11, lines 25-28, could be removed. The two first sentences are basic limnological knowledge and the last sentence does not really lead the story further. In this chapter, the theme of light penetration and absorption is discussed in many places, e.g. on page 11, lines 8-9, lines 11-12 and lines 29-34, page 12, lines 11-13 and lines 19-20. Remove excess repetition. The last paragraph on page 12 (starting Fig. 14 depicts. . .) repeats what is said earlier and is not the main focus of this study, therefore I suggest to remove that paragraph. The last paragraph of Chapter 3.2.2. discusses about modeled ice cover. This seems a bit out of scope and there really seems to be no ice measurements against which to validate modeling. For this reason, I suggest to either remove this paragraph or significantly shorten it.

**The previous reply:** Thank you for the comment. Indeed, in situ measurements of ice are not available at the station. However, we performed a sensitivity analysis of FLake to variations in Kd in reproducing ice phenology and thickness. We feel that this it is useful to see the possible impact of Kd on ice conditions even if no in situ observations are available. This type of sensitivity analysis is commonly performed by the modeling community.

The paragraph starting with "Fig. 14 depicts", discusses how different values of Kd are affecting simulations of different layers of the thermal structure which is one of the objectives of our study. Fig. 14 elaborates more on temperature changes with depth. The timing factor has been removed in this figure compared to Fig. 13, to simplify making this point.

Page 11 lines 25-28 of the old version of manuscript have been removed. To avoid repetition, changes are made on page 11 lines 18-20 of the new manuscript. Also, removed are: page 11 lines 10-12, page 11 lines 31-33, page 12 lines 11-12 and lines 19-20 in the old version of manuscript.

**The new reply:** Thank you for the detailed comment. Page 11 lines 25-28 of the old version of manuscript have been removed. To avoid repetition, changes are made on page 10 lines 28-30 of the new manuscript. Also, removed are: page 11 lines 10-12, page 11 lines 31-33, page 12 lines 11-12 and lines 19-20 in the old version of manuscript.

The paragraph starting with "Fig. 14 depicts" in the previous version of manuscript is removed and changes are made in page 11 lines 20-30.

Indeed, in situ measurements of ice are not available at the station. However, we performed a sensitivity analysis of FLake to variations in Kd in reproducing ice phenology and thickness. We feel that this is useful to see the possible impact of Kd on ice conditions even if no in situ observations are available. This type of sensitivity analysis is commonly performed by the modeling community.

Page 11, lines 11-12. The authors seem to mix two concepts here. Darker water color is related to dissolved substances, such as colored dissolved organic matter, not to particulate matter.

Thank you for the comment. We agree that the two concepts have been mixed in the manuscript. However, we believe that light attenuation can be described using the terms "dark" and "clear" waters.

Clarity describes concentrations of both dissolved and suspended matters and can be related to attenuation of light. Therefore, the term of "clear water" (as opposed to "dark water") is used in this manuscript to explain waters with low (high) light attenuation coefficients. Light attenuation in clear (dark) waters is low (high) and this could be because of the existence of dissolved (e.g. absorption) or suspended matters (e.g. scattering).

On the other hand, turbidity is an indirect measure of scattering by particles (Bukata et al.,1995). It does not include dissolved matter in its definition. Therefore, the term "turbid water" is related to high concentrations of suspended matters. Turbidity of water could be low but still with high light attenuation due to high dissolved matters concentrations. Therefore, the term "turbid water" has been changed to "dark water" in the manuscript (page 2 lines 31, 33 – page 10 line 28, 29 – page 11 lines 1, 11, 12, 29, page 12 lines 15, 23).

Page 11, lines 13-14. The authors over-simplify the underlying mechanisms for LSWT behavior. The loss of energy to the atmosphere is related to the surface water temperature (and wind), not only in fall but throughout the open-water season. However, the mechanism how mixed layer depth affects the rate of heat loss needs more explaining.

**The previous reply:** We agree with the comment and have therefore modified page 11 lines 19-28 to reflect this. Considering MLD to explain the reason is basically combining the effect of both temperature and wind. This is because mixing is related to both wind forcing and convection. The mixed layer depth (MLD) affects the speed at which energy is lost to the atmosphere throughout the year.

**The new reply:** We agree with the comment and have applied modifications. To add explanation about the effect of MLD, page 10 line 31-33 and page 11 line 1-2 are modified. Considering MLD to explain the reason is basically combining the effect of both temperature and wind. This is because mixing is related to both wind forcing and convection. The mixed layer depth (MLD) affects the speed at which energy is lost to the atmosphere throughout the year.

Page 11, lines 18-23. Tie these results from the literature more tightly to the findings in this study, e.g. by writing whether this study supports or opposes previous findings. Also, the sentence on lines 21-23 (starting 'Heiskanen et al. . .') could be removed either from here or from the Summary.

**The previous reply:** The results of our study is already tied to other studies found in the literature. In page 11 lines 29-31, the result of our study is that the sensitivity of the model increases from Min to CRCM-12.6 simulation (Kd decreasing from 0.58 to 0.2). The statements after these lines (page 11 lines 31-33 and page 12 lines 1-2) are discussing other studies which support the finding of our study. This finding is that FLake is more sensitive to Kd values less than 0.5.

To prevent repetition, the statement starting at "Heiskanen et al. (2015) recommend to use a value of Kd that is too high rather….." has been removed from the summary on page 13 line 31 and page 14 line 1 of the old manuscript.

**The new reply:** Thanks for your comment. Page 11 line 6 is modified to show that other studies also support the results of our study.

To prevent repetition, the statement starting at "Heiskanen et al. (2015) recommend to use a value of Kd that is too high rather....." has been removed from the summary on page 13 line 31 and page 14 line 1 of the old manuscript.

Page 12, paragraph starting with 'Fig. 12 shows'. Here full mixing is described in very atypical way on several occasions, e.g. by 'highest depth of mixing' and 'reaches maximum MLD'. I suggest to describe these occasions either by discussing of overturn, of full mixing or similar.

**The previous reply:** Lake turnover is the process of lake's water turning over from top to the bottom, which is full mixing. The maximum/highest possible depth of mixing at NDBC station is 12.4 m, so when MLD reaches this depth turnover happens. This is the reason for describing turnovers using terms such as 'highest depth of mixing' and 'reaches maximum MLD'. However terms of "maximum MLD" and "highest depth of mixing "have been changed to "full mixing "or ""overturn" on page 12, lines 13-17.

**The new reply:** Thanks for your comment. Terms of "maximum MLD" and "highest depth of mixing "have been changed to "full mixing "or ""overturn" on page 11 line 34, page 12 line 1.

Page 12, lines 8-9. If this is the reason for earlier overturn in simulations with clearer water, how the authors then explain the results shown in Figs. 11 and 13 where it is evident that there is full mixing in the beginning of September in CRCM-12.6 simulation with temperatures of about 20 deg C? Fig. 14 also shows that the clearer the water, the higher the water temperatures in Oct and Nov. Note that in addition to convection, mixing is related to wind forcing and density gradient in the water column.

**The previous reply:** CRCM-12.6 has the clearest water compared to other simulations, therefore the water column reaches the same temperature in its layers earlier than other simulations, leading to earlier turnover. Figures 11 and 13 and 14 support this statement.

**The new reply:** Thank you for catching this mistake. The sentence has been removed in the new manuscript since turnover can happen at different temperatures as long as the water column is at the same temperature.

Page 12, lines 16-17. MLD is not influenced by the thermal structure, but it is part of the thermal structure. I would remove this sentence.

Point well taken. The sentence on page 12 lines 16-17 of the previous manuscript has been removed.

Page 12, lines 13-14. Fig. 13 is essentially the same data as in Fig. 12 and therefore one cannot be used to confirm the results of the other.

**The previous reply:** confirms" has been changed to "also demonstrates" on page 12 line 24.

**The new reply:** Thank you. This sentence is removed in the new version of manuscript.

Page 12, lines 20-22. Deepening of the thermocline is related e.g. to wind forcing and thus it cannot be suggested that thermocline deepening in clear waters is monotonic. Also, it is not clear what is meant with 'stabilize the temperatures'. I suggest to remove this sentence.

**The previous reply:** Figure 13 shows that in the simulation related to the clearest waters (CRCM-12.6), deepening of thermocline is faster, with a constant speed (monotonically increasing), as opposed to the dark waters.

On page 12 line 22 of the previous manuscript, "stabilize" has been removed.

**The new reply:** These sentences are removed on page 12 lines 21-22 of the previous manuscript. Thanks.

Technical corrections

Page 6, line 23. The same result 'was' found for. . .

It has been corrected on page 6 line 24. Thanks.

Page 8, line 32. 'leads to higher water clarity'. The authors must mean lower water clarity.

Thank you for catching this mistake. It has been corrected on page 7 line 15.

Page 9, line 18. The sentence starting 'The Kd values' and the sentence after that could be merged and rephrased. E.g. 'The monthly-averaged Kd were used to simulate the surface water temperature and produce a merged LSWT (Merged)."

Thanks. It has been rephrased on page 9 line 8.

Page 9, line 20. Comparing LSWT in situ observations (Obs) with. . .

It has been added on page 9 line 10. Thank you.

Page 9, line 21. How can the authors compare measured and modelled surface temperatures in April when there seems to be little or no measured LSWT during April, at least according to Fig 9?

**The previous reply:** Observations for 2006 and 2007 start on 19 and 18 April, respectively, so there is data available for comparison.

**The new reply:** Unfortunately, there was no in situ measurements available for April 2005. Also, in situ measurements in April 2006 and 2007 starts from 19 and 18 April, respectively. Therefore, these available data were used in calculation of MBE for Spring (April-June) in 2006 and 2007.

Page 10, lines 2-3. Rephrase. Do the authors mean that the annual average of Kd can occasionally be closer to the actual Kd than the monthly-averaged Kd? This same topic is also mentioned in lines 16-17, and at least to me it is unclear how yearly average value (i.e. one single number) can represent the extent of Kd variations (i.e. how big is the range).

Thanks for your comment. Monthly averages are calculated based on satellite-derived Kd values, which might not be available due to cloud coverage in MERIS images. However, there are more MERIS images available in the longer period of one year that can potentially catch the actual variations of Kd value, rather than only a few images (or even none) in a month. Therefore, a yearly-average Kd could potentially be closer to the actual Kd value. The statement has been rephrased on page 9 lines 24-26 and page 10 lines 3-4.

Page 10, line 10. Please clarify what is specifically meant with 'are as affected'.

It means that no matter which depth we used, the actual depth at station or a tile depth, the large under-prediction happened for both simulations of CRCM-12.6 and CRCM-20 (MBE for both is above 1ºC compared to Avg and Merged simulations), especially for temperatures above 12 ºC.

This point is clarified on page 9 line 30-32: "The under-prediction of these model runs is stronger, particularly for LSWT above 12ºC, which can be explained by the Kd value used. This is because no matter what depth is used in simulations (either actual or tile depth), both CRCM runs have larger MBE compared to Avg and Merged simulations."

Page 10, lines 11-12. 'This can be explained by. . .'. Be more specific in telling how lake depth explains this.

Thanks for your comment. CRCM-12.6 and CRCM-20 only differ in the depth used as input in the simulations. Therefore, if CRCM-20 has the most under-prediction compared to all other simulations (including CRCM-12.6), it is related to the input depth. Clarification has been added to the manuscript on page 9 lines 33-34.

Page 10, lines 12-13. This should be self-evident if the model is any good, and therefore I suggest to remove this sentence.

**The previous reply:** The authors would prefer to keep the statement to emphasize on this and results from other studies which are also mentioned in page 9 lines 18-19.

**The new reply:** Thanks for your comment. These sentences are removed in the new version of manuscript.

Page 11, line 9. '. . . causing thinner mixing depth (Fig. 12)'

It has been added on page 10 line 30 (Fig. 8 in the new manuscript). Thanks.

Page 11, line 35. Change 'when Kd changes. . .' e.g. to 'when maximum (minimum) Kd is used instead of its average value. . .'

Thanks for your comment. It has been rephrased on page 11 lines 14-16.

Page 12, line 1. Similar comment as previous. This is a bit misleading wording since it gives the idea that Kd changes naturally, whereas what is meant that different Kd is used as an input.

Thank you. It has been rephrased accordingly on page 11 line 14-16.

Page 13, line 27. Write open the abbreviation 'LSWT' here. It is not typical abbreviation and not clear for those who only read Summary and conclusions.

Thank you for the comment. It has been added on page 14 line 1 and also for other abbreviation on page 14 lines 6-7.

Page 14, line 3. Change 'has' to 'have'.

Thanks for catching the mistake. It has been corrected on page 14 line 8.

Comments to figures

Fig. 1. It would be of interest to see the main river inlets and outlets. This way it would be easier to assess how much river inflow possibly affects modeling results.

Thank you for the suggestion. Assessing the impact of river inflows and outflows on the simulation results is outside the scope of this paper as these are 1-D simulations. However, inflows/outflows have been added in Fig.1 of the revised manuscript.

Fig. 5. Remove 'Lake Erie boundary' from the legend, it is not needed. Also make the color bar much larger. Same comments for other similar figures.

Thank you. This has been corrected in Figs 10-12 in the revised manuscript.

Fig. 8. It would be of interest to see the SDD at this location (or from the nearest location where those exist) together with these CC-derived Kd. These could be marked to the same graph with secondary y-axis.

Thank you for the suggestion. We agree that adding SDD measurements makes the graph more informative and interesting. However, there are no in situ SDD measurements available for NDBC station. According to Fig. 1, the nearest locations with SDD observations are within about a 20 km distance from NDBC station. On the other hand, water optical properties change in spatial scales much smaller than this distance. Therefore, showing SDD values for those stations are not a good approximation of SDD for the NDBC station, and comparing those to CC-derived Kd is not rationale.

Fig. 10. It would be interesting to see the performance for each year separately. This could shown by plotting each year with different color. Also, it is more standard to show these kind of scatter plots as box plots (both axes of same length).

**The previous reply:** The performance of each year is shown separately in Table 3. Also, we preferred to add color to Figure 10 to show the seasonal pattern of the three years of LSWT simulations (based on comment from reviewer #3).

**The new reply:** Thank you for your suggestion. This is a valuable comment that also suggested by another reviewer. Therefore, plots are redrawn using different colors; however, colors are defined based on different seasons rather than different years as our focus here is on the open water season. In addition, the performance of simulations based on each year separately (2005-2007) is already mentioned in Table 3.

The main purpose of these graphs is to show over/underestimation of each type of simulations against observations, for each season. We believe that scatterplot make these points more clear as we are not interested to show the range of the variable and finding the outliers to exclude them from further processing.

Fig. 11. The measured LSWT should be shown. Otherwise, it is impossible to say which simulation performs the best. Use a) and b) for these two graphs. Also in the legend, the Kd values could be shown for each model run.

**The previous reply:** a) and (b) are now used in the new manuscript for Fig. 11. Details of simulations (Kd value and depth) are given in the manuscript on page 11 lines 15-18. Therefore, to avoid repetition and save space, Kd values are not added to the legends of the figures.

However, because this figure is related to a sensitivity analysis, there is no need to show the observations. Section 3.2.1, which is more related to the accuracy assessment and improvement of simulation results, shows observations.

**The new reply:** Thank you for your comment. (a), (b), and (c) are now used in the new manuscript for Fig. 6. Details of simulations (Kd value) are added to the caption of Figs 6-9.

This figure is showing a sensitivity analysis, comparing the model response to the Kd change. We are not interested to see which one is closer to the observations. We are interested to see how the model responses to the change of Kd values. For this purpose of sensitivity analysis, we considered the Avg simulation as the base result to compare the results of other simulations and show how different the simulations results are when we change Kd values.

Fig. 15. Model run CRCM-12.6. is not visible. If the resolution can not be increased, describe in the caption where the line is.

Thank you for your comment. Description of the figure is given in the body of manuscript (page 12 lines 11-12). It has now been added as well in the caption of Fig. 9.

**Reviewer #3 Comments:**

In my opinion, the authors improved the manuscript and responded satisfactorily to most of the questions that I put in my first review. However, I still have disagreements regarding 5 issues. In particular, the question 4 is to me a major issue.

The review is based on the of file hess-2016-82-author_response-version1.pdf

5    1. page 6, line 21/22

In my first review, I asked: depth of the water temperature measurements: why is it included here? The water temperature is not a forcing parameter...

The authors reply that:

"z_Tw_m" is one of the inputs in FLake model and is the depth of water temperature measurements."

10    Which is not correct.

I'm not sure about what is z_Tw_m (It is not present in the standard Flake code, see: http://www.flake.igb-berlin.de/docs.shtml), but I have no doubt that Flake don't need any "depth of water temperature measurements" as imput.

Thanks for your comment, we agree with the reviewer. Therefore, modifications have been made in page 6 line 20. This parameter is removed in the revised manuscript.

15    2. Page 6 line, line 25: I asked "to configure" means force initialize, or both?

The authors reply: The parameters mentioned in the bracket are constant and used to force the FLake model."

I have to insist. The authors made some confusion between configuration parameters and forcing. The height of wind measurement (5 m) and the height of air temperature sensor (4 m) are in fact constant used to configure the model, but the measured meteorological parameters and model-derived irradiance were used to force the FLake model (and not to configure).

20    We appreciate your comment. The point is now well taken; and changes are made in the manuscript page 6 line 21-22.

3. page 7 line 22 and equation 2

I also insist about equation 2. The authors find the following relation between kd and SDD:

Kd = 1.64 × SDD−0.76

I have nothing against this result, but they should not say that it is of the type of eq 2...

25    Thanks for your comment. To avoid confusion, the term "based on Eq. (2)" has been removed in the new manuscript, page 7 line 28.

4. Page 10, lines 17-18:

"Dark waters in these months contribute in reproducing colder LSWT for Avg and Merged simulations in 2005."

I cannot understand why water turbidity (or darkness) tends to decrease the lake surface water temperature. As the authors

30    correctly stated along the manuscript, in dark waters the radiation is more absorbed in the surface layer, and so the LSWT tends to be higher by comparison to clear waters. What I proposed in my previous review is that:

The reason lies not in the fact that during those months the water is more turbid, but because the water was more turbid before, during spring and summer, reducing the heating of deep water. This should be discussed further, in particular by analyzing the evolution of deep temperature and column mean temperature (Flake variables).

35    The authors reply: "According to Table 2, the Kd value for same months of year 2005-2007 are in the same range. But the difference in calculating MBE for 2005 compared to 2006 - 2007 is taking months of Sep-Nov into the calculation of MBE for 2005. Therefore, the underestimation of LSWT in 2005 cannot be related to darker waters before, in spring and summer. It is more related to the months that are taken into the calculation of MBE."

In my opinion the authors don't have reason and must examine and take into account the evolution of lake water bottom

40    temperature (LWBT) and column mean temperature (MWCT). Alternatively they have to provide a physically explanation about how "Dark waters in these months contribute in reproducing colder LSWT(...)".

Thanks for your comment and pointing this out. We examined calculating MBE for 2005-2007, focusing only on the months that are common between three years (May-Aug). The calculated MBE did not resulted in the same conclusion (either overestimation or underestimation) for three years. So our conclusion (Considering the months of September-November into

the calculations of MBE for 2005 can be the reason of underestimating LSWT in this year for both Avg and Merged simulations compared to two other years) was not correct. So we removed these sentences in the revised manuscript.

If the in situ data for LBWT and MWCT were available, the performance of simulations for over/underestimation of them could be examined and related to over/underestimation of LSWT in different simulations. However, these in situ data are not available in our study.

Instead, LBWT and MWCT are produced in the Flake simulations. The attached excel file compares LSWT produced from different simulations for each year separately (Page 50-52). In 2005, the underestimation of LSWT increases from Avg to Merged, CRCM12.6, and CRCM20 simulations, respectively. However, the highest LBWT is estimated for CRCM12.6. Therefore, there is no relationship between under/overestimation of LSWT and producing LBWT.

The excel file also compares LSWT produced in each simulation for different years (Page 53-56). In Avg simulation, the LSWT is underestimated in 2005; and overestimation of LSWT in Avg simulation increases from 2006 to 2007, respectively. The graph for LBWT shows that temperature for 2005 was the highest and decreased in 2006 and then 2007, respectively. For CRCM12.6, LSWT is underestimated and the underestimation increases from 2006 to 2007, and 2005, respectively. However, LBWT in CRCM12.6 is highest for 2006, 2005, and 2007, respectively. Therefore, there is not any pattern for relationship between over/underestimation of LSWT in each simulation for different years and the produced LBWT.

If we only compare simulations for 2006 and 2007, which cover the exact same time of year: in CRCM12.6 and CRCM20 underestimation increases from 2006 to 2007. But, LBWT in CRCM12.6 is higher for 2006 compared to 2007; whereas in CRCM20, LBWT is higher for 2007 compared to 2006.

Therefore, we can conclude that LBWT does not have a relationship with under/overestimation of LSWT in different simulations and different years. Similar explanation is correct for MWCT. Therefore, we don't have any other explanation for underestimation in Avg and Merged simulations for 2005, contrary to overestimation in Avg and Merged simulations for 2006 and 2007. Having more in situ data, especially for LBWT and MWCT, could be helpful to clarify the reason.

5. Page 13 line 11:

I still consider that the use of the term monotonic is not appropriate

Thanks for your comment. This term has now been removed in the new manuscript.

---

## Author Response (AR3)

[revised manuscript text omitted]

We appreciate the reviewer's detailed and constructive comments, which helped improve the manuscript. Our replies to comments are covered below.

5   Reviewer comments to Satellite-Derived Light Extinction Coefficient and its Impact on Thermal Structure Simulations in a 1-D Lake Model by Zolfaghari et al. 2016

The authors have significantly improved the manuscript and replied adequately to most of the reviewers' comments. The manuscript is now structurally sound and easy to follow. There are still several things that need clarifying and corrections, but these can be considered as minor comments.

10   General comments:

Abstract is long and occasionally overly detailed. It should be more compact. Acronyms should be used only when necessary. Now there are e.g. MBE and I_a, which are not commonly known. Since the abstract should be shortened, I suggest to leave out these statistical parameters.

Thank you for this comment. The abstract has been shorten in the new version. Lines 7-9, 11-14, 15-16, 16-17 and 21-23 of old version have been modified accordingly. "NDBC station" in line 20 of old manuscript is mentioned generally as "the Lake Erie station" in line 14 of the new version to avoid the acronym.

Sometimes present tense is used in the manuscript even though past tense is more appropriate.

Thank you for your comment. The use of the past tense has now been made whenever appropriate across the entire manuscript. We do not provide specific line numbers and pages here, since it was a long list.

Specific comments:

Page 1, lines 11-12. "…algorithm is applied to MERIS satellite imagery to estimate Kd and evaluated against Kd derived from Secchi disk depth…". As written, it is not clear if the algorithm is evaluated against Kd from SDD or is it the estimated Kd that is evaluated against Kd from SDD. I suggest to change it to: "… to estimate Kd, which was evaluated against…". I'm not a native English speaker but think that more attention should be paid on sentence structure throughout the manuscript.

Thank you for your comment. This sentence has been removed in the new version to improve clarity and to shorten the abstract.

Page 1, line 24-25. "Dark waters always produce warmer MWCT". Should it read 'colder MWCT'? At least Fig. 6b implies so.

Thanks for catching this mistake. Correction is made on page 1 line 17.

Page 2, line 8. "Lake Surface Water Temperature…", only the first word should have capital initial letter.

5  Thank you. Page 2 line 8 is now corrected.

Page 2, the second paragraph is really long and could be separated into two paragraphs. The second paragraph could start e.g. after "…derived from satellite imagery." in line 28.

As suggested, the second paragraph on page 2 of the old manuscript is now separated into two paragraphs, starting at line 28 on page 2 of the revised manuscript.

10  Page 2, line 27. The reference Heiskanen et al., 2015 can be omitted because it is clear from the sentence that the authors refer to this study.

The reference has been removed from page 2 line 27.

Page 2, lines 32-33. "The daily mean LSWT range increased…". It is not clear what is meant with the word 'range' because only one number (not range) is given for clear and dark waters.

15  You are correct. The word "range" has now been removed from the sentence on page 2 line 33 to avoid confusion.

Page 3, line 19. "… to investigate the improvement…". If this is the first study to use satellite-derived Kd in FLake simulations, how can it be known from beforehand that the model performance is improved? Maybe the word 'improvement' could just be omitted: "… to investigate the model performance…"

20  Thank you for the comment. The improvement of the model is assessed versus previous studies that were using only a "generic" constant value in lake modeling. The value in Martynov et al. (2012) is not derived using satellite observations, but it is a constant value used to apply FLake on Lake Erie. Our study compares the results with the approach in Martynov et al. (2012) to show the improvement when satellite-derived Kd is used versus using a constant value which is not lake-specific. The sentence has been

25  modified accordingly on page 3 line 19-20 to clearly make this point. It now reads: "The objectives of this study were to: 1) evaluate satellite-derived $K_d$ values for a large lake in the Great Lakes region; 2) apply the evaluated satellite-derived $K_d$ in FLake model to investigate the improvement of model performance to reproduce LSWTs, compared to previous studies using a constant $K_d$ value of 0.2 m$^{-1}$."

Page 3, line 20. "… and a constant value…". Be more specific, a constant value throughout the study period, or constant value of 0.2 m-1 or some other type of constant value.

The constant value is now specified on page 3 line 21: "…compared to previous studies using a constant $K_d$ value of 0.2 m$^{-1}$."

Page 3, line 20. I suggest to change the word 'demonstrate' to 'evaluate' for pretty much the same reason as in the comment about 'improvement'.

Correction applied.

Page 3, line 23. "… to Kd values based on simulated LSWT,…". Sentence structure should be changed because now it reads that the Kd values are based on simulated LSWT etc.

Thank you for the comment. The sentence has been changed. It now reads: "…3) understand the sensitivity of the FLake model to variations in $K_d$, based on the analysis of simulated LSWT,…"

Chapter 2.1. There are now many references to Fig. 1 (page 4, lines 3-4, 10-11, 13, 23) even though only one would be needed. I suggest to put a sentence in the beginning of the paragraph starting "The meteorological forcing…" that says e.g. "The data for this study was collected from different stations shown in Fig. 1.". Then delete the other references to Fig. 1.

Thank you for your comment. Modifications have been made on page 4 lines 2, 3, 4, 10, 13, 22.

Page 4, line 4. Remove the coordinates and depth from here or from page 6, line 21.

The coordinates and depth are now removed from page 6 line 21 of the old version.

Page 4, line 6. "Water temperature is also measured at 0.6 m below the water line.". It seems to me that water temperature is measured only at 0.6 m depth, and therefore the word 'also' should be deleted. I also suggest to use 'water surface' instead of 'water line'.

"Also" has been removed, page 4 line 6.

Page 4, line 32. Does the "(see Fig. 1)" refer to the corresponding tile or only to the station? If only to the station, then this ref. can be omitted. If it refers to the tile, that should be specified in Fig. 1. Now there are vertical dashed lines in the figure but they are not defined in the caption.

Page 4 line 31 is modified (it refers to the station). Explanation of the dashed lines has been added in the caption of Fig. 1.

Page 5, line 10. At least to me it is unclear what the term 'screen height' means. If this is generally known definition, then okay, but if not, it should be specified.

Thank you for your comment. The term is defined in the reference that is mentioned in the manuscript (Duguay et al. (2003), page 5 line 5) and also the references therein.

Page 5, line 12. "NWRI-EC", change to "NWRI" so that same terms would be used throughout the manuscript.

Thank you for the comment. This is corrected on page 4 line 15 and page 5 line 11.

Page 5, line 18. "… Earth's surface high spectral…". Is there a word missing between 'surface' and 'high'?

Thank you for catching this mistake. It is corrected on page 5 line 17. The word "at" was missing. Now reads: "…Earth's surface at high spectral…"

Page 5, line 23. The "in Lake Erie (see Fig. 1)" could be omitted, it has already been defined where the station is.

Correction applied. Removed from page 5 line 22.

Page 5, line 25. The authors use different terms to mean the same thing, e.g. "CC Level2W", "CC MERIS L2W", "CC L2W". Make sure that the same term is always used. It is also not clear what Level2 data product means.

Page 5 line 23-24 now explains what level 2 product is. Page 5 line 30, page 8 line 1 are changed to "CC Level2W" to be consistent throughout the manuscript.

Page 5, line 26. "Concentration of water constituents". This surely doesn't mean ALL the constituents, so it should be changed to "concentration of some water constituents" or similar.

Thank you for catching this. Now corrected on Page 5 line 26. The word "some" has been added.

Page 6, line 18. Are all these references really needed for this statement?

Only one main reference is now mentioned on page 6 line 18.

Page 7, line 1. I suppose that it should read that the MBE is calculated as the mean of modelled values…

Thanks for catching this mistake. It is now corrected on page 7 line 1.

 Could the authors provide a reference for the statement that resuspension is the most important cause for low water clarity in Lake Erie? Later, heavy plankton blooms are also discussed as the cause of high differences in water clarity.

Resuspension is mostly the major factor in low water clarity of shallow sections of the lake, whereas algal bloom is in general a factor threatening water quality of the whole lake. Clarifications are now applied on page 7 lines 11-14. A reference is also added. The sentence now reads as "Lake Erie (specifically its shallow regions) is more susceptible to re-suspension of bottom sediments compared to the other Great Lakes, which leads to lower water clarity (Binding et al., 2010)."

Page 7, equation 2. It has been shown in later studies that the relationship between Kd and SDD is not constant, as the authors also discuss later in this chapter. It is unclear why the authors want to argue here that it is constant. Or has this relationship been the basis of relating CC-derived Kd and SDD together? If so, why that was used instead of the exponential relationship shown e.g. in Arst et al. 2008?

Equation 2 shows the general form of relationship between SDD and Kd, which demonstrates these two variables are negatively correlated (page 7 lines 22-23). As it is discussed in the manuscript, this relationship was established by a pioneer study of Poole and Atkins (1929), but later on, other studies such as Koenings and Edmundson (1991) showed that different values of "k" than 1.7 (introduced in the pioneer study) can be derived, depending on the type of water bodies (page 7, lines 26-27).

In the other studies, there different relationships were introduced, although their general form is still showing a negative correlation between the two variables of SDD and Kd. Armengol et al. (2003) (page 7, lines 27-28), Arst et al. (2008) (page 8, lines 14-15), and the relationship in our study (page 8, line 13), all resulted in a power relationship with the form of $y = ax^{-k}$ , where "a" and "k" are constant values in each study, but different from each other. The "constant" term used in our manuscript is not referring to a constant value for the relationships developed for all different water bodies. Instead, it is implying that this is a constant value in one developed relationship, which could be a different value from one study to another (lines 26-27).

This is clarified on page 7 lines 26-28.

Page 8, lines 15-17. The authors say that, after SDD validation, the satellite-derived water clarity (as such, without any modifications) was used in the modelling. There seems to be quite good agreement between

SDD and CC-derived Kd when SDD>3m (Fig. 3). However, in Discussion the authors state that in situ measurements of water clarity are a requirement for satellite-derived Kd. Is there some reason to assume that in (clear) waters in general (e.g. SDD>3m) the satellite imagery doesn't provide reasonably reliable estimates of Kd? In other words, if it worked for Lake Erie, what are the assumptions that it might not work on other lakes?

Thank you for this interesting question and comment. Lake optical properties vary significantly in different water bodies. The optical properties of Lake Erie are specifically complex as they are affected by different water constituents including suspended and dissolved matters, as well as algal bloom, which does not covary with the concentrations of the other two. Therefore, if the CoastColour algorithm is providing Kd with a high accuracy in this paper for the Lake Erie NDBC station, it still has to be verified in other water bodies using collected in situ data. But in general, for clear waters (e.g. SDD>3 m) with less complex optical properties, such as ocean, the performance of algorithms is expected to be even higher. But we are only speculating here, and further research and experiment is required to test this.

Page 8, line 16. "are deemed to be correct". There is huge scatter in Fig. 3 when SDD when SDD<2m, so it is an overstatement to say that the satellite-derived Kd are correct. I suggest to change the wording to "can be considered representative".

We applied your suggested correction on page 8 lines 17-18. Thank you.

Page 8, lines 21-22. Very complex and vague subordinate clause (motivating the investigation of potential of integrating). Be more specific.

Thank you for the comment. Modifications have been made on page 8 lines 22-24. It now reads: "Direct measurements of $K_d$ in the field is not widely available. These limitations motivate the investigation on the potential of integrating satellite-based estimations of $K_d$ into lake models. "

Page 8, line 30. "focused on", change e.g. to "shown for"

Change applied on page 9 line 1.

Page 9, paragraphs 1 and 2. It is quite difficult to follow the big picture what was done. This could be made easier for the reader if e.g. after "…, respectively." on line 4, a new sentence would be written: "We made four different simulation schemes which were then compared to the observed LSWT".

Thank you. This sentence has been added on page 9 lines 7-8.

Page 9, line 6-7. Where does the acronym CRCM come from?

The acronym comes from the study by Martynov et al. (2012). Our study was built upon their study to improve FLake simulations results. FLake results in Martynov study are coupled with the Canadian Regional Climate Model (CRCM), as a lake representing scheme. This information was mentioned in page 2 line 20 already.

Page 9, line 11. Change sentence order from "surface temperature in spring (April-June) is modelled warmer" to "surface temperature was modelled warmer in spring (April-June)"

Change applied on page 9 line 15.

Page 9, lines 11-13. Looking at Fig. 4, it seems to me that 2006 and 2007 'merged', 'avg', and 'obs' are quite similar from late-June to mid-August. This is contrasting to what is now written in the text.

Thank you for your comment. The calculations on page 9 lines 15-17 are based on LSWT values in all three years of 2005-2007 together. Therefore, these years together show underestimation in fall and summer, overestimation in spring. The seasonal-based performance of each simulations for all three years together is also shown in Figure 5.

If we only focus on summer (which is considered as months of July-Sep) 2006, Avg and Merged simulations have MBE values of -0.38 and -0.46. For summer 2007, Avg and Merged simulations have MBE values of -0.40 and -0.41, respectively.

Page 9, line 16. Change sentence order from "to more slowly gain (lose) heat" to "to gain (lose) heat more slowly".

Corrected. Page 9 line 20 is now modified.

Page 9, line 17. Change "The performance" to "The overall performance".

Correction applied on page 9 line 21.

Page 9, paragraph starting with Fig. 5. and Fig. 5 itself. Write the text in the same order as they are presented in the figure. Either change the order in the text or in the figure. Now the discussion starts with CRCM simulations whereas they are panels c and d in the figure.

Thank you for this comment. We have made the suggested change on page 9 line 34 and page 10 lines 5-12.

Page 9, line 29-30. This needs a bit of rephrasing because according to the figure the CRCM simulations underestimate the observed LSWT only when LSWT is roughly >7 deg C.

The sentence on page 10 line 6 has been modified.

Page 10, line 29. The authors argue that solar radiation is absorbed more in dark waters due to existing particles in water. If you can reliably state that this is true for Lake Erie, then the explanation is acceptable. But even in this case I suggest to change 'dark waters' e.g. to 'waters with low clarity" because particles do not always make the water dark, and the water can be dark without particulate matter.

Changed from "dark waters" to "waters with low clarity" on page 11 line 3.

Page 10, line 30. "(lower LBWT…)". I suggest to change it to "(which shows in lower LBWT…)" so that the reader doesn't need to guess what the authors try to imply.

Changed as suggested on page 11 line 4.

Page 10, line 30. Fig. 8 can not be introduced before Fig. 7 has been introduced. Either change the text or one option would also be to combine Fig. 8 to Fig. 6 (it could e.g. be subplot d in Fig. 6).

Thank you for the suggestion. Figure 8 is now in Figure 6 as a subplot (d). Changes are made accordingly on page 11 line 4-5, and page 12 line 6.

Page 11, line 26. "lessening of the radiative absorption" implies that something has happened in the lake. I suggest to change this to "decrease in radiative forcing" which means that the incoming radiation from the atmosphere has decreased.

Changed as suggested on page 11 line 34.

Page 11, lines 28-29. These are very vague sentences and not all parts true. The deepening of the thermocline is related to wind forcing. In dark waters the density gradient is sharp and forms an effective barrier for the wind-induced mixing to reach deeper depths. In clear waters the density gradient is weaker and therefore mixing can more easily deepen the thermocline. There are many processes working at the same time in lakes that affect thermal stratification. Besides heat transfer, wind currents and internal waves are important. Because the same wind forcing is used as an input for all the different model runs, it is important to explain how water clarity takes part into the development and progress of thermocline.

Thank you for this comment. The heat transfer is happening due to different factors including convective, and mechanical forcing (wind driving mixings, and internal waves), as it is mentioned by the respectful

reviewer. Therefore, all these factors control the heat transfer and consequently the deepening of the thermocline. This information is now added on page 12 line 3-5.

Page 11, line 30. "derived from isotherm". Be more specific how MLD was defined. It seems to me that the authors have identified the MLD correctly, but there are many ways to do this and no general guideline how to do this. Therefore, more specification is needed.

MLD and isotherms are both direct outputs of FLake. Therefore, "derived from isotherms" might be misleading here since the authors did not apply any calculation to actually derive MLD values. Instead, the output of FLake for MLD was used. What we were trying to imply here was that, the two figures of MLD and isotherms might have overlap in what they are showing. More specifically, MLD can be a subsequence of isotherm plot. But here we show them separately for different purposes and to make different points.

The sentence on page 12 line 4 is now modified to avoid confusion.

Page 12, paragraph starting with "In the darkest water…". Here the same oversimplifications are presented. If there is no wind, there is no mixed layer in clear or dark water because there is only stratification, no mixing. So the explanation is not only that in clear waters the solar radiation can distribute to a larger volume in the water column. Very important factor is also how much (deep) of the density stratification can be destroyed by wind-induced turbulent kinetic energy. In dark waters this layer is shallower than in clear waters and therefore dark waters have shallower MLD with the same wind forcing.

Thank you for mentioning this point. This is now added on page 12 lines 14-16.

Page 12, lines 9-25. If there is some study that shows that FLake predicts well the ice phenology in Lake Erie, then this text can be as it is and that study should be cited. Otherwise, these are only simulation runs without validation, and therefore in the beginning or in the end of these paragraphs a text should be added that mentions "It must be noted that these results couldn't be verified because of lack of measurements" or similar.

A sentence has now been added to this effect on page 13 lines 1-2.

Chapter 3.3. This chapter needs the most modification. Now there are three main points in the chapter: 1. spatial variation, 2. temporal variation, and 3. inter-annual variation of Kd. Currently, the authors briefly

describe what was observed and show the figures, but the meaning and importance of these findings are not elaborated. From the figures 10-12 it seems evident that these are important findings but these are not discussed. For example, it seems interesting that Kd can be time-independent constant even though there are huge changes in Kd both in space and in time. If Kd influences the thermal stratification as shown in
5  Fig. 7 and related studies, then it would be reasonable to assume that the thermal stratification is very different in the western end of Lake Erie than in the eastern end. Yet, some studies and results suggest that one lake specific but constant Kd can be used to model the stratification. Fig. 11 seems to imply that in big lakes lake-specific Kd cannot be used.

As the reviewer mentions, this section has three main points: 1- spatial variations of Kd which is covered
10  in lines 9-18 page 13; 2- temporal variation of Kd, covered in lines 19-25 page 13; the inter-annual changes of Kd, covered in line 26 page 13 to line 2 page 14. These lines explain the observation of Kd variations derived from satellite imagery, demonstrating that satellite measurements are capable of capturing these variations, while the conventional in situ measurements cannot. This is mentioned on page 13, lines 5-9, which is the main focus of this section, to imply the strength of remote sensing versus in
15  situ measurements.

A **lake-specific time-invariant** Kd value is sufficient for simulating thermal structure of NDBC station. However, this comment and conclusion has to be confirmed for other locations on the lake, since depth is another major factor influencing simulation results.

**Lake-specific constant for the full Lake:** Other locations of the lake, especially eastern basin, are
20  potentially significantly sensitive to the changes in Kd value. Since most of the lake has Kd values in the critical threshold, a lake-specific constant Kd value cannot be used in 3-D lake models on Lake Erie.

**Time-invariant:** Using a time-invariant Kd value for other locations depend on the influence of depth on the simulation results. The importance of these results has been added at the end of the section 3.3, lines 4-9 of page 14.

25  Page 13. Kd values are presented as average value plus minus some number. Could you specify what the number is.

Thank you for your comment. The numbers are the average value ± standard deviation. This is now clarified on page 13 line 16.

Page 13, line 25. It seems that the years for CC product are 2003-2012 (fig. 2), not 2002-2012. When this paragraph is written as it is now, it seems that these were the years for all the measurements and modelling. In order to not be misleading, specify at least for which years the model runs were made.

Thank you for catching this mistake. Page 14 line 12 is now corrected. All simulation results are studied for years 2003-2012. However, the simulations were run from 2002-2012, and then the first year of results (2002) omitted from further processing. This is recommended for the 'spin up' period to stabilize the model output in lake modeling community.

Page 14, lines 8-9, first sentence of this paragraph. It is unclear what the authors mean with the concept of 'thermal regime of lakes' in regards of this study. Only observed surface water temperatures are used to validate modelled temperatures. All the rest (water column temperature, bottom water temperature, mixed layer depth) are only simulated and thus tells more of how FLake model performs with different water clarity in this lake than how the lake thermal structure actually was influenced during the years in this study period. Also, it has already been shown in previous studies (which the authors cite) that transparency impacts physical processes, and thus this is not a new finding. I suggest to replace the first sentence of this paragraph with specific strengths of this study.

Thank you. Modifications are now applied on page 14 lines 27-29.

Page 14, line 22. Change "Flake" to "FLake".

Thank you for the comment. It is corrected on page 15 line 9.

All figures in general. Include tick marks to all figures, both x- and y-axes. Also minor tick marks could be useful in some cases.

Thank you for this comment. All figures x- and y-axis have major ticks now. Minor ticks are added for numeric axes.

Fig. 1. Write the meaning of the acronyms in the caption or describe what the different stations are. Specify what are the vertical dashed lines.

Thanks for the comment. The meaning of the abbreviations is now provided in the caption of Fig. 1. Also, the vertical dashed lines are specified.

Fig. 2. Write open what 'CC-derived' means.

The abbreviation is "opened" in the caption of Fig. 2. Thank you for the comment.

Fig. 4. It is difficult to understand the caption and how it is exactly linked to the figure. Discuss the lines in the caption in the same order as in the legend of the plots. Write open what 'Obs', 'AvgXXXX', 'Merged', 'CRCM-12.6', and 'CRCM-20' mean. Assign (a), (b), and (c) to the subplots. What is the time resolution of the data? The general principle is that the reader should be able to understand the figure without having the need to constantly see the main text.

The caption of Fig. 4 has been modified. And (a), (b), (c) are assigned to plots.

Fig. 9. It seems that the green line (CRCM-12.6) is missing from the plot. The last sentence of the caption seems out of place. The proper place should be in Results/Discussion.

Thank you for your comment. In the Result and Discussion section, page 12 lines 13-14, it is mentioned that CRCM-12.6 and min (Avg and Max) simulations are reproducing similar ice phenology. This explains the reason for not seeing the green and black lines as obvious as red and orange ones. This explanation is also mentioned in the caption to avoid confusion for the reader. We feel that the last sentence in caption should remain to elaborate the reason for not seeing two of the lines. More explanation is added in the caption of Fig. 9.

Figs. 10-12. Show more values in the colorbars. Now only 0; 2.5 and 5 are shown, the interval should be at least 1 m-1 (i.e. 0, 1, 2, 3, 4 ,5). Show the unit of the colorbar somewhere

Thank you for the comment. The colorbar values in the figures have been changed and unit is also added, Fig. 10-12.